# Pan-cortical 2-photon mesoscopic imaging and neurobehavioral alignment in awake, behaving mice

Evan D Vickers[1], David A McCormick[1,2]*

[1]Institute of Neuroscience, University of Oregon, Eugene, United States;
[2]Department of Biology, University of Oregon, Eugene, United States

**Abstract** The flow of neural activity across the neocortex during active sensory discrimination is constrained by task-specific cognitive demands, movements, and internal states. During behavior, the brain appears to sample from a broad repertoire of activation motifs. Understanding how these patterns of local and global activity are selected in relation to both spontaneous and task-dependent behavior requires in-depth study of densely sampled activity at single neuron resolution across large regions of cortex. In a significant advance toward this goal, we developed procedures to record mesoscale 2-photon $Ca^{2+}$ imaging data from two novel *in vivo* preparations that, between them, allow for simultaneous access to nearly all 0f the mouse dorsal and lateral neocortex. As a proof of principle, we aligned neural activity with both behavioral primitives and high-level motifs to reveal the existence of large populations of neurons that coordinated their activity across cortical areas with spontaneous changes in movement and/or arousal. The methods we detail here facilitate the identification and exploration of widespread, spatially heterogeneous neural ensembles whose activity is related to diverse aspects of behavior.

*For correspondence:
davidmc@uoregon.edu

**Competing interest:** The authors declare that no competing interests exist.

## eLife assessment

This **important** paper presents a thoroughly detailed methodology for mesoscale-imaging of extensive areas of the cortex, either from a top or lateral perspective, in behaving mice. The examples of scientific results to be derived with this method offer promising and stimulating insights. Overall, the method and results presented are **convincing** and will be of interest to neuroscientists focused on cortical processing in rodents and beyond.

## Introduction

Recent advances in large-scale neural recording technology, such as widefield imaging (*Musall et al., 2019*; *Gallero-Salas et al., 2021*; *Esmaeili et al., 2021*), large field-of-view (FOV) 2-photon (2p) imaging (*Sofroniew et al., 2016*; *Stringer et al., 2019*; *Yu et al., 2022*), and Neuropixels high-density extracellular electrophysiology recordings (*Jun et al., 2017*; *Steinmetz et al., 2019*) have allowed for rapid advancement in our understanding of the relationships between brain-wide neural activity and both spontaneous and task-engaged behavior in mice. For example, these techniques can now be deployed in recently developed behavioral paradigms that allow for the temporal separation of periods during which cortical activity is dominated by activity related to stimulus representation, choice/decision, maintenance of choice, and response or implementation of choice during different intra-trial epochs of 2-alternative forced choice (2-AFC) discrimination tasks (*Guo et al., 2014*), the standardization of training and performance analysis across laboratories (*Aguillon-Rodriguez et al.,*

2021; eLife), and the separation of context-dependent rule representation and choice in a working memory task (*Wu et al., 2020*).

One of the striking features of neocortical neuronal activity is how strongly changes in behavioral state, such as task engagement, movement, or arousal, affect the spontaneous and evoked activity of neurons within visual, auditory, somatosensory, motor and other cortical regions (*Niell and Stryker, 2010*; *McGinley et al., 2015*; *Musall et al., 2019*; *Stringer et al., 2019*). However, this is not to say that these effects are uniform across the cerebral cortex (*Morandell et al., 2023*; *Wang et al., 2023*). Neurons exhibit diversity in the dependence of their neural activity on arousal and behavioral state both between and within cortical areas (*Niell and Stryker, 2010*; *McGinley et al., 2015*; *Shimaoka et al., 2018*), and these areas are active at different times during a rewarded task (*Salkoff et al., 2020*; *Esmaeili et al., 2021*; *Gallero-Salas et al., 2021*). Furthermore, the arousal dependence of membrane potential across cortical areas has been shown to be diverse and predictable by a temporally filtered readout of pupil diameter and walking speed (*Shimaoka et al., 2018*).

For this reason, directly combining and/or comparing the correlations between behavior and neural activity across regions imaged in separate sessions may not reveal the true differences in the relationship between behavior and neural activity across cortical areas, due to a 'temporal filtering effect'. In other words, the correlations between behavior and neural activity in each region appear to depend on the exact time since the behavior began (*Shimaoka et al., 2018*). In our view, this makes the simultaneous recording of multiple cortical areas essential for proper comparison of the dependencies of their neural activities on arousal/movement, because only then are the distributions of behavioral state dwell times the same across cortical areas.

Areas involved in sensory decision making are often far from each other (*Gallero-Salas et al., 2021*) and can exhibit coordinated state-dependent changes in functional coupling (*Clancy et al., 2019*). Also, multimodal sensory information is multiplexed and combined as it ascends across the cortical hierarchy (*Coen et al., 2023*). For these reasons, understanding the brain activity underlying optimal performance during multimodal, task-engaged behavior will require dense sampling of many brain areas at single neuron resolution across lateral, dorsal, and frontal cortices simultaneously at a temporal resolution high enough to describe both spontaneous behavioral state transitions and the neural dynamics relevant for a given task.

Dense intra-cortical sampling at a fixed depth/cortical layer across many areas is not possible with current Neuropixels probes (*Steinmetz et al., 2021*), 1-photon (1p) widefield imaging can be contaminated by neuropil and hemodynamic signal (*Waters, 2020*; *Valley et al., 2020*) and typically does not achieve single cell resolution (but see *Yoshida et al., 2018*; *Kauvar et al., 2020*), and standard 2p imaging is limited by scanning speed (see *Gong et al., 2015*) and field of view spatial extent (FOV; i.e. simultaneously imageable, either contiguous or non-contiguous, regions; *Allen et al., 2017*; *Hattori et al., 2019*; but see *Yu et al., 2021*). Note that although some recent advances in 1p widefield imaging have allowed for the imaging of individual cells, both in head-fixed and freely moving mice, they do not achieve true single cell resolution in practice (they get close to ~10 µm xy or 'lateral' resolution, with undefined z or 'depth' resolution) and rely in part on the sparse labeling of neurons with $Ca^{2+}$ indicators either within or across cortical layers (*Cai et al., 2016*; *Hope et al., 2023*, bioRxiv; *Xie et al., 2023*).

Recent advancements in cranial window preparations have enabled imaging over large portions of the dorsal cortex (*Kim et al., 2016*; *Stringer et al., 2023*, bioRxiv). However, simultaneous 2p imaging over a large cortical area including both dorsal and lateral regions, particularly in a preparation that allows for simultaneous imaging of the major primary sensory cortices (auditory, visual, and somatosensory) and frontal motor/choice areas (M1, M2) in awake, behaving mice has not been previously shown.

Here, we developed a set of new techniques and integrated them with existing technologies in order to overcome the limitations of current state-of-the-art methods and provide the first pan-cortical 2p assays at single-cell resolution in awake, behaving mice. To achieve this, we designed custom 3D-printed titanium headposts, mounting devices, adapters, and cranial windows, and modified ('Crystal Skull'; *Kim et al., 2016*; *Figure 1a, d*, *Figure 1—figure supplement 1c, upper, e*) or developed ('A1/V1/M2' or 'temporo-parietal'; *Figure 1b, e*, *Figure 1—figure supplement 1a, f*) two *in vivo* surgical preparations, which we will henceforth refer to as the 'dorsal mount' and 'side mount', respectively.

The dorsal mount, which was based on the earlier 'Crystal Skull' preparation (*Kim et al., 2016*; see also *Ghanbari et al., 2019* for a similar preparation), included modifications such as a novel head-post and support arms, along with other hardware and novel surgical, data acquisition, and analysis methods, and enabled simultaneous imaging across nearly all of bilateral dorsal cortex. Our novel side mount preparation, on the other hand, allowed for simultaneous imaging of much of the dorsal and lateral cortex across the right hemisphere (although our design can easily be 'flipped' across a mirror-image plane to allow for imaging of the left hemisphere, if desired).

These two novel preparations enabled mesoscale 2p imaging (here, we used the 2p-RAM meso-scope, Thorlabs; *Sofroniew et al., 2016*) of up to ~7,500 individual neurons simultaneously at ~3 Hz across a 5x5 mm FOV (*Figure 2*, *Figure 3*, *Figure 1—figure supplement 2*; *Videos 1, 2*, 8 and 9; Protocol 1), or up to 800 neurons combined across four 660x660 µm FOVs at ~10 Hz (*Figure 1—figure supplement 2d, e*; Protocol 1; see Supplementary methods and materials). Although these recording speeds are not fast enough to capture the full dynamics of rapid neural processing involved in, for example, initial cortical sensory encoding and decision making, they are faster than earlier similar mesoscale recordings in visual cortex (*Stringer et al., 2019*), and they are likely fast enough to capture important components of spontaneous arousal and movement related fluctuations in neural dynamics across dorsal and lateral cortex. The total number of imaged neurons can be significantly increased in our preparations by using other mouse lines or different imaging or analysis parameters.

Finally, we designed a custom, LabView-controlled behavioral setup with up to 3 high-speed body and face cameras (*Figure 1a, b*; *Videos 1 and 2*). This behavioral monitoring allowed us to compare widespread, densely sampled, high-resolution neural activity to movement and behavioral arousal state variation in head-fixed, awake, ambulatory (i.e. running atop a cylindrical wheel) mice under conditions of spontaneous behavior (*Figure 4*, *Figure 5*, *Figure 6*, *Figure 4—figure supplements 1 and 2*, *Figure 5—figure supplements 1 and 2*, *Figure 6—figure supplement 1*), passive sensory stimulation (*Figures 1–3*, *Figure 1—figure supplement 2*), or 2-alternative forced choice (2-AFC) task engagement (*Figure 1c*, Protocol 1).

We present here detailed methods, along with example recording sessions during spontaneous behavior from both our dorsal and side mount preparations, to demonstrate the feasibility of wide-spread 2p cortical neuronal imaging simultaneously with behavioral monitoring. For a graphical sche-matic overview of our methods, please see the following supplementary material: https://github.com/vickerse1/mesoscope_spontaneous/blob/main/pancortical_workflow_diagrams.pdf; *Vickers, 2024b*.

## Results

Recent studies employing large-scale imaging (e.g. widefield 1p and mesoscale 2p) and/or electro-physiology (e.g. Neuropixels) recording technologies have suggested that a significant percentage of the variance of neural activity across neocortex can be accounted for by rapid spontaneous fluctua-tions in arousal and self-directed movement during both spontaneous behavior and task performance (*Musall et al., 2019*; *Steinmetz et al., 2019*; *Stringer et al., 2019*; *Jacobs et al., 2020*; *Salkoff et al., 2020*; *Stringer et al., 2023*, bioRxiv). Given that arousal state- and movement-related activity appears to be a ubiquitous feature of cortical activity across all regions, it would be advantageous to develop methodologies that allow for simultaneous, single neuron resolution, contiguous monitoring of neuronal activity during second-to-second movements and changes in arousal, during both spon-taneous and trained behaviors. Here, we harness new imaging and analysis technologies in order to both address this methodological gap and provide a proof of principle test of these methods by examining the relationship of behavioral arousal and movement to detailed spatial patterns of neural activity across the dorsal and lateral neocortex.

The overall workflow for our mouse preparations, data acquisition, and data analysis can be found in the following supplementary document: https://github.com/vickerse1/mesoscope_spontaneous/blob/main/pancortical_workflow_diagrams.pdf; *Vickers, 2024b*. The remainder of our supplemen-tary materials are also hosted on the main /mesoscope_spontaneous folder of our GitHub repository, including documented analysis code, design and 3D-component printable files, grayscale versions of all figures, and supplementary figures and movies. Related data for example recording sessions shown in main and supplementary figures are publicly available on FigShare+ at: https://doi.org/10.25452/figshare.plus.c.7052513.

# Large field of view 2-photon imaging in behaving mice

In order to simultaneously monitor the activity of cortical neurons across a large area (~25 mm², or up to 36 mm² in some cases) of either the bilateral dorsal cortex (dorsal mount), or of both dorsal and lateral cortex across the right hemisphere (side mount) in awake, behaving mice, we chose to utilize 2p imaging of GCaMP6s fluorescence using an existing and commercially available 2p random access mesoscope (Thorlabs; 2p-RAM; *Sofroniew et al., 2016*). We chose this form of data collection for use in head-fixed mice because it allows for: (1) rapid scanning (i.e. it uses a resonant scanner coupled to a virtually conjugated galvo pair) over a large (5x5 mm fully corrected, or up to 6x6 mm imageable), field-curvature corrected field of view (FOV); (2) subcellular resolution in the z-axis to avoid region-of-interest (ROI) contamination by neuropil and neighboring cells (0.61 x 0.61 x 4.25 µm xyz point-spread function at 970 nm laser excitation wavelength); (3) correction for aberrations at all wavelengths between 900–1070 nm, to allow for combined imaging of multiple fluorophores. We achieved these three objectives in head-fixed mice that were able to walk freely on a running wheel (See *Figure 1a, b*, *Video 1*, and Video 6), to clearly view visual monitors on both sides, and to lick either of two water spouts (left, right) for 2-alternative forced choice responses, while we performed left, right, and rear videography to monitor each mouse's face, pupil, and body.

The neuronal imaging methodology we chose (see below) is not the only one available. For example, another promising and recently developed alternative implementation, Diesel 2p, allows for similarly flexible and large FOV imaging at single-cell resolution, but with a significantly higher z-axis point-spread function (~8–10 µm; *Yu et al., 2021*). Despite this drawback, its use of dual scan engines allows for true simultaneous imaging of different cortical areas, while the 2p RAM mesoscope (Thorlabs) accomplishes this by rapid jumping (~1 ms) between FOVs. Both systems offer excellent corrected field curvatures on the order of +/-25 µm over ~5 mm.

Several primary design constraints needed to be met to achieve our goal of monitoring neuronal activity over a large portion of the mouse lateral and/or dorsal cortices. These ranged from aspects of the surgical preparation, mounting, and imaging, to neural recording (*Figure 1—figure supplement 1*, *Videos 1 and 2*, Protocols II-III). To increase the extent of the neocortex over which we could monitor neural activity, we developed two distinct headpost/cranial window preparations. In the first, the dorsal mount, the mouse is head-fixed upright on a cylindrical running wheel and the objective is vertical (*Figure 1a*). Here, both wings of the headpost are attached to support arms, and each support arm is mounted on two vertical 1" diameter posts secured with flexing feet for maximum stability. This preparation allowed for monitoring of neural activity over large regions of both cerebral hemispheres, from the posterior aspects of visual cortex to the anterior aspects of motor cortex, and laterally to the dorsal-most aspects of auditory cortex, depending on the placement of the 2p mesoscope objective (*Figure 1d*).

The second preparation, the side mount, required the head of the mouse to be rotated 22.5° to the left (or right, not shown), with the microscope objective vertical or tilted 1–5 degrees to the right, so as to extend the 5x5 mm imaging FOV to include the auditory cortex and other neighboring ventral/lateral cortical areas (*Figure 1b and e*). For the side mount, the headpost had only a single, left wing

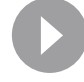

Dorsal mount

**Video 1.** Dorsal mount. Upper left, 3D printed titanium headpost for dorsal mount (left; i.materialise.com and sculpteo.com) and accompanying cranial window for dorsal mount (labmaker.org, or TLCInternational.com and glaswerk.com), shown in top and side orthogonal projection, followed by isometric projection (AutoDesk Inventor, Adobe Acrobat Pro 3D viewer). Upper right, 3D printed plastic (PLA) light-shields or 'woks', same views as upper left. Horizontal light shield (left) fits onto perimeter of dorsal mount headpost and is attached with Sylgard 170 Fast Cure silicone elastomer, and vertical light shield fits onto vertical perimeter ridge of horizontal light shield and is held by gravity. Bottom, simultaneous and temporally aligned high-resolution videography from three points-of-view of a mouse under spontaneously behaving conditions (shown at 3x speed or 90 Hz; left and right camera are GigE Teledyne Dalsa M2050 cameras, and posterior camera is FLIR grasshopper USB3 a camera). Example pose tracking labeling by DeepLabCut (*Mathis et al., 2018*) is shown (right camera).

https://elifesciences.org/articles/94167/figures#video1

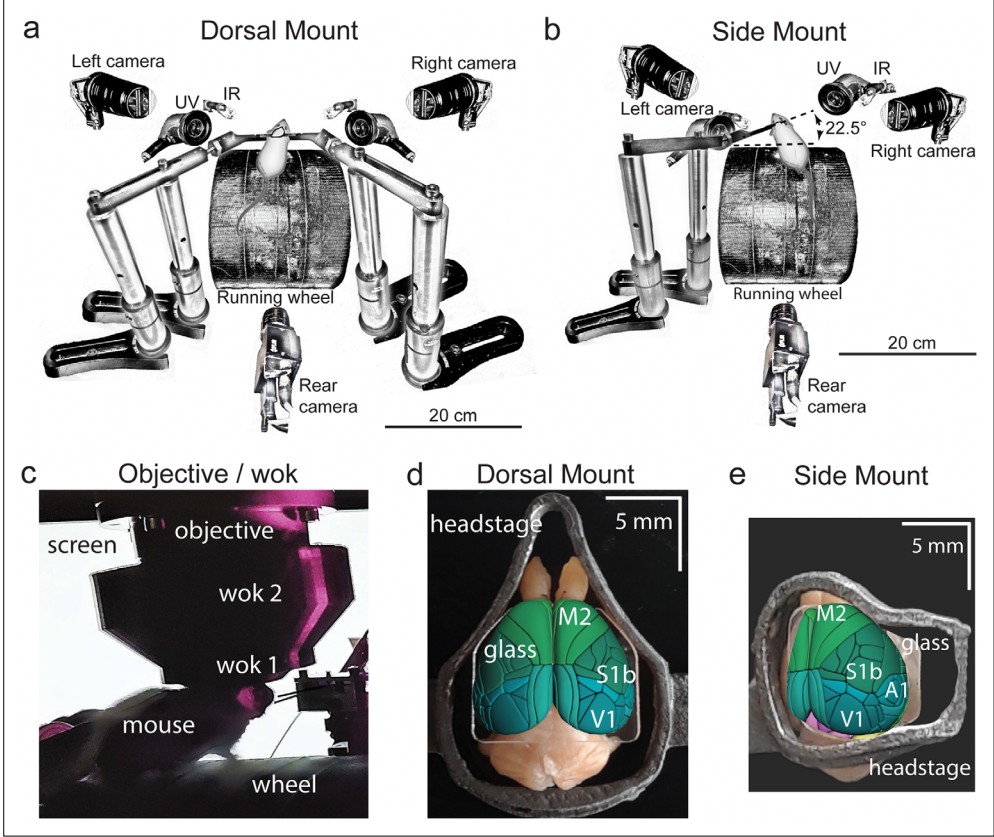

**Figure 1.** Technical adaptations for dual-mount *in vivo* pan-cortical imaging with the Thorlabs 2p-RAM mesoscope. (**a**) Mesoscope behavioral apparatus for the dorsal mount preparation. Mouse is mounted upright on a running wheel with headpost fixed to dual adjustable support arms mounted on vertical posts (1″ diameter). Behavior cameras are fixed at front-left, front-right, and center-posterior, with ultraviolet and infrared light-emitting diodes aligned with goose-neck supports parallel to right and left cameras. (**b**) Mesoscope behavioral apparatus for side mount preparation. Same as in (**a**), except that the mouse is rotated 22.5 degrees to its left so that the objective lens (at angle 0) is orthogonal to the center-of-mass of the preparation across the right cortical hemisphere. The objective can rotate +/-20 degrees medio-laterally, if needed, to optimize imaging of any portion of the cortex under the cranial window. The right behavior camera is positioned more posterior and lower than in (**a**), to allow for imaging of the eye under the acute angle formed with the horizontal light shield, shown in (**c**). (**c**) Mouse running on wheel with side mount preparation receiving visual stimulation from an LED screen positioned on the left side, with linear motor-positioned dual lick-spouts in place and 3D printed vertical light shield (wok 2) attached to rim of flat shield (wok 1) to block extraneous light from entering the objective. (**d**) Overhead view of dorsal mount preparation with 3D printed titanium headpost, custom cranial window, and Allen CCF aligned to paraformaldehyde-fixed mouse brain. Motor region = light green, somatosensory = dark green, visual = dark blue, retrosplenial = light blue. Olfactory bulbs (anterior) at top, cerebellum (posterior) at bottom of image. Note ridge along perimeter of headpost for fitted horizontal light shield (wok 1) attachment. (**e**) Rotated dorsal view (22.5 degrees right) of side mount preparation with 3D printed titanium headpost, custom cranial window, and Allen CCF aligned to paraformaldehyde-fixed mouse brain. Auditory region shown in light blue, ventral and anterior to visual areas, and ventral and posterior to somatosensory areas (right side of image is lateral/ventral, and left side of headpost perimeter is medial/dorsal). Other regions shown with the same color scheme as in (**d**). UV = ultraviolet, IR = infrared, M2 = secondary motor cortex, S1b = primary somatosensory barrel cortex, V1 = primary visual cortex, A1 = primary auditory cortex.

The online version of this article includes the following figure supplement(s) for figure 1:

**Figure supplement 1.** Detailed technical adaptations for dual-mount *in vivo* pan-cortical imaging with 2p-RAM mesoscope.

**Figure supplement 2.** Widefield-imaging based dual-mount multimodal CCF alignment, ROI detection, and areal assignment of GCaMP6s$^+$ excitatory neurons.

so that the right side of the mouse's face would not be occluded. The deep lateral/ventral extent of the side mount headpost prevented easy addition of a right headpost wing. Attaching the left wing to a single support arm supported by two 1" diameter vertical posts mounted with flexing feet, as in the dorsal mount preparation, was sufficient to minimize movement artifacts. A key additional difference to note here is that two small screws were used to attach the left wing of the side mount headpost to the support arm, thus giving it additional stability. The dorsal mount preparation used only one screw (the outermost hole) on each of the left and right side headpost wings. Thus, both preparations were secured by a total of two mounting screws, each countersunk into headpost arms. An important design feature is that the headpost wings fit into recessed rectangular slots machined into the support arms, facilitating stability.

We observed that mice adapt well to head tilt in the side mount preparation, and will readily whisk, walk or run, and learn to perform lick response tasks in this configuration. Keeping the microscope objective in a vertical or near-vertical orientation, and rotating the mouse's head (instead of rotating the objective), significantly improved the manageability of the water meniscus between the objective and the cranial window. However, if needed, the objective of the Thorlabs mesoscope may be rotated laterally up to +/- 20° for direct access to more ventral cortical areas, for example if one wants to use a smaller, flat cortical window that requires the objective to be positioned orthogonally to the target region. In general, setups such as Sutter's MOM system and Thorlabs' Bergamo microscope, which offer even greater degrees of microscope objective rotation, would allow enhanced versatility with our preparations.

Preliminary comparisons across mice indicated that side and dorsal mount mice showed a similar degree of behavioral variability both within and across sessions. An example of the relative ease with which mice adapted to the ~22.5 degree neck rotation can be seen in *Video 2* and Video 10. It was in general important to make sure that the distance between the wheel and all four limbs was similar when using either preparation. In particular, careful attention must be paid to the positioning of the front limbs in the side mount mice so that they are not too high off the wheel. This can be accomplished by a slight forward angling of the left support arm. With dorsal mount mice, it was important to not place them too close to the wheel, in which case they would exert pressure on the headpost and support arms, leading

Side mount

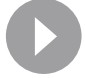

**Video 2.** Side mount: same as in dorsal mount (*Video 1*), but for side mount hardware. Note that the mouse's headpost is retained by a single fixed support arm, rotated 22.5 degrees to the left, whereas the mouse in the dorsal mount example is held by dual orthogonally positioned fixed support arms.
https://elifesciences.org/articles/94167/figures#video2

Visual multimodal mapping (Widefield, dorsal mount)

Mean dF/F (%)

Time (s)

**Video 3.** Dorsal mount multimodal mapping: example widefield (1p) imaging of GCaMP6s fluorescence responses during a visual multimodal mapping session. Top right, full field, left-side isoluminant Gaussian noise stationary grating patches (vertical and horizontal stationary grating patches; 0.16 cpd, 30 deg; *Michaiel et al., 2019*) presented to elicit a visual response in right cortex, with small upper left-corner alternating white/black box positioned under photodiode to record precise stimulus presentation times. Top left, pixelwise dF/F response of the entire image for a single trial, recorded at 50 Hz and shown at 0.5 x speed. The baseline was calculated as the median of a 1 s period leading up to the stimulus onset. The dashed white circle indicates the putative primary visual cortex (right V1). A = anterior, L = left, R = right, P = posterior, dF/F = change in fluorescence divided by baseline fluorescence; midline extends vertically near the center of frame from bottom to top edge roughly between the 'A' and 'P': labels. Bottom, trace of mean dF/F for all pixels inside dashed white circle (mask), expressed as percent change. Vertical black line indicates stimulus onset. The visual stimulus is present through the end of the epoch shown. Upper left, overlay: mean of 33 dF/F responses (mean of 1 s after stimulus onset minus mean of 1 s leading up to stimulus onset) in a single dorsal mount session under 2–3% isoflurane anesthesia. ITI = inter-trial interval, measured from beginning of one stimulus to beginning of the next stimulus.
https://elifesciences.org/articles/94167/figures#video3

Whisker multimodal mapping (Widefield, dorsal mount)

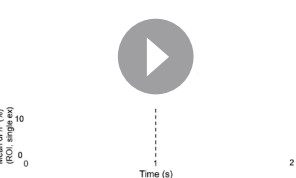

Auditory multimodal mapping (Widefield, side mount)

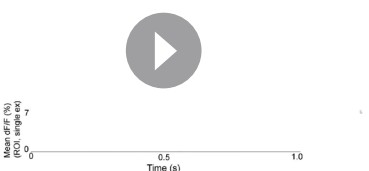

**Video 4.** Whisker stimulation: same as in dorsal mount multimodal mapping visual stimulation example video (*Video 3*), with same mouse on same day, except with 5 Hz, 100 ms duration forward swipes with custom 3D printed plastic (PLA) whisker-deflector, as indicated by vertical deflections in stimulus trace, mid-right. Example video of mouse shown from a different session than dF/F data, because mouse face video was typically not recorded during multimodal alignment sessions. Red S1b (and dashed line) = right primary whisker barrel cortex. A = anterior, P = posterior, L = left, R = right, V1 = primary visual cortex.

https://elifesciences.org/articles/94167/figures#video4

**Video 5.** Side mount multimodal mapping: Same as in *Video 3* but for side mount preparation. 1.5–3% isoflurane anesthesia was used in all 3 sessions. Auditory: 1 s tone cloud with tones between 2 and 40 kHz presented for 0.5 s starting at black vertical dashed line (bottom). Sonogram display from Spike2 (CED) shows individual tones as horizontal green lines, where y-axis is sound frequency (~0–25 kHz) and x-axis is time (0–1 s). Movies shown at 0.25x speed. As in other example videos, the mouse shown is from a different session, but is exposed to the same stimulus at the indicated time. The mouse shown is from a different session type when videography was enabled, to show the normal response of the mouse to the stimulus. ml = midline, A1 = primary auditory cortex, A = anterior, L = left, R = right, P = posterior, dF/F = change in fluorescence divided by baseline fluorescence.

https://elifesciences.org/articles/94167/figures#video5

to increased movement artifacts, or too far above the wheel, which can significantly reduce walking bout frequency.

Although it was in principle possible to image the side mount preparation in the same optical configuration without rotating the mouse (by rotating the objective to 20 degrees to the right), we found that the last 2–3 degrees of unavailable, yet necessary, rotation (our preparation is rotated 22.5 degrees left, which is more than the full available 20 degrees rotation of the objective), along with several other factors, made this undesirable. First, it was difficult or impractical to attach the horizontal light shield and to establish a water meniscus with the objective fully rotated. One could use gel instead of water (although, in our hands, gel was optically inferior), but without the horizontal light shield, light from the UV and IR LEDs can reach the photomultiplier tubes (PMTs) via the objective and contaminate the image or cause tripping of the PMT. Second, imaging the right pupil and face of the mouse was difficult under these conditions because the camera would need the same optical access angle as the objective, or would need to be moved down toward the air table and rotated up

Visual multimodal mapping (Widefield, side mount)

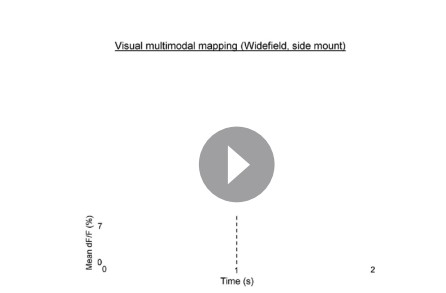

Whisker multimodal mapping (Widefield, side mount)

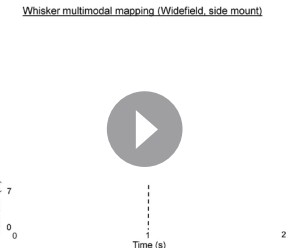

**Video 6.** Same as in *Video 5* but for visual stimulation, with different side mount example mouse. ml = midline, A1 = primary auditory cortex, A = anterior, L = left, R = right, P = posterior, dF/F = change in fluorescence divided by baseline fluorescence.

https://elifesciences.org/articles/94167/figures#video6

**Video 7.** Same as in *Videos 5 and 6* but for whisker stimulation, with same side mount example mouse as in *Video 5*. ml = midline, A1 = primary auditory cortex, A = anterior, L = left, R = right, P = posterior, dF/F = change in fluorescence divided by baseline fluorescence.

https://elifesciences.org/articles/94167/figures#video7

20 degrees, in which case its view would be blocked by the running wheel and other objects mounted on the air table. A system of mirrors for imaging the animal could be used to overcome this problem, but would further complicate the current setup.

Cortical imaging in the novel side mount preparation was restricted to one hemisphere and a thin medial strip (~1 mm wide) of the contralateral hemisphere. This allowed for imaging from roughly the anterio-medial areas of V1 to medial and medio-lateral secondary regions of motor cortex (or slightly farther, if the anterior-posterior axis of the brain is oriented along the diagonal of the 5x5 mm FOV, or if one images to the full allowable but partially uncorrected 6x6 mm extent with ScanImage). This anterior-posterior reach was similar to that observed in the dorsal mount preparation. However, the side mount windowing and the accompanying mounting procedure significantly increased our ability to image neural activity in lateral (ventral) cortical aspects, including auditory, somatosensory, and association cortical regions (*Figure 1e*), notably without the need for substantial rotation of the objective.

To achieve 2p neuronal imaging from broad cortical regions, we created large and stable, custom (designed in AutoDesk Inventor) 3D-printed titanium (laser sintered powder with active cooling and shot-peened post-processing) headposts (*Figure 1d, e* and *Figure 1—figure supplement 1e, f*; Suppl Design Files; i.materialise.com and sculpteo.com) and glass (0.21 mm thick, Schott D263T) cranial windows (*Figure 1*, *Figure 1—figure supplement 1a, c, d, e, and f* ; *Videos 1 and 2*; Suppl Design files: https://github.com/vickerse1/mesoscope_spontaneous (copy archived at *Vickers, 2024a*); Labmaker.org for dorsal mount; TLC International for cutting, and GlasWerk for bending, for the side mount and early proto-types of the dorsal mount). We incorporated these custom made headposts and cranial windows into the dorsal mount (*Figure 1a and d*, *Figure 1—figure supplement 1c* upper, d upper left, e), and side mount (*Figure 1b, e*, *Figure 1—figure supplement 1a, c, lower, d, lower left and right, and f, upper right*) preparations. The dorsal mount preparation allowed for imaging over roughly the same extent of bilateral dorsal cortex as in *Kim et al., 2016*, although we were able to image up to ~25 mm$^2$ at a time, compared to the more typical maximum 2p FOV of ~1 mm$^2$. In addition, the area of bilateral cortex imageable with this preparation is comparable to that of many recent studies employing widefield 1-photon imaging, such as *Musall et al., 2019*. The side mount preparation allowed for imaging of an extent of right hemisphere comparable to that of *Esmaeili et al., 2021*, although they used through-the-skull widefield 1-photon imaging, while our cranial window preparation allows for either widefield 1-photon imaging, or single neuron resolution 2p imaging, in the same mouse.

To mimic the curvature of the brain and reduce tissue compression, which is a problem with flat covers-lips with a diameter larger than ~3 mm, cranial windows were curved by heating pre-cut (TLC International) glass pieces over 9 or 10 mm bend-radius molds (GlasWerk). The bend radius (i.e. radius of half cylinder mold over which the melted glass is bent to achieve its curved shape) was fixed at 10 mm for Labmaker.org dorsal mount windows (*Figure 1—figure supplement 1e*, right; *Video 1*; early attempts with 11 or 12 mm bend radii failed for all but the largest adult male mice). For windows custom designed in collaboration with GlasWerk (*Figure 1—figure supplement 1f*, right; *Video 2*), the bend radius was set at either 9 mm, for a tight fit to ventral auditory areas, or at 10 mm, to enable simultaneous imaging of the entire prepara-tion at a single focal depth. In our experience, successful cranial windows last between 100 and 150 days, or in some rare cases up to 300 days: https://github.com/vickerse1/mesoscope_spontaneous/blob/main/window_preparation_stability.pdf; *Vickers, 2024b*. The keys to this stability were even pressure across the surface of the window, complete enclosure of the glass-skull interface with flow-it, causing minimal damage to the dura during the craniotomy, and constant attention to the minimization of infection either across the surface of the skull, between the headpost and window surgeries, or around the external perim-eter of the headpost, after the window surgery (See Protocol III and *Figure 1—figure supplement 1*).

The success rate for window implantation, for both dorsal mount and side mount preparations, was around ~65% (*Figure 1—figure supplement 1g*; see also Protocol III). However, with extensive practice this can be raised to be closer to 75–80%, especially if care is taken to both maximize the area of the crani-otomy and to leave the dental cement attaching the headpost to the skull undamaged during drilling, so that the perimeter of the window fits fully inside the craniotomy, the window 'floats' or sits freely directly on the surface of the brain, and the headpost remains firmly attached to the skull. This was accomplished in later iterations of our designs and methods by making the side mount window slightly smaller at the anterior edge of M2, and by using an adjustable support arm (*Figure 1—figure supplement 1b*, lower) mounted on a small breadboard attached underneath the base of the stereotax in order to fix the mouse in its 22.5 degree left-rotated position during the window surgery so that it would not move or vibrate

during drilling of the craniotomy. This also acted to minimize 'break-throughs' of the drill bit tip through the skull and into the brain.

The main causes of death during cranial window surgery were exsanguination during the surgery following a ruptured major blood vessel on the surface of the brain (either sagittal or lateral, usually), damage to the dura caused by the skull during the removal step of the craniotomy or by the sharp tip of a tool used to remove the skull, failure to properly pressurize the window across the entire cortical surface, or failure to completely seal the window to the skull fragment around its entire perimeter with flow-it (here, rapid UV light application during flow-it application can help). In some cases where the mouse survived the window surgery it was still considered a failure if a bleed re-erupted and occluded at least 30% of the window, or if glue occluded at least 30% of the window either above or below the surface of the window. In general, most bleeds cleared, on their own, by cerebrospinal fluid within a week if there was no direct clotting on the surface of the brain.

Changes in vasculature following cranial window surgery were usually minimal but could involve the following: (i) sometimes a vessel is displaced or moved during the window surgery, (ii) sometimes a vessel, in particular the sagittal sinus, will enlarge or increase its apparent diameter over time if it is not properly pressured by the cranial window, and (iii) sometimes an area experiencing window pressure that is too low will, over time, show outgrowth of fine vascular endings. The most common of these was (i), and (iii) is perhaps the least common. In general the vasculature is quite stable, and window preparations that we observed to be of high quality at ~1 week post-surgery showed minimal changes over the first ~150 days (see supplementary materials: https://github.com/vickerse1/mesoscope_spontaneous/blob/main/window_preparation_stability.pdf; *Vickers, 2024b*).

Within a field of view of 5x5 mm, curvature of the brain, especially in the mediolateral direction, is a significant problem. To partially compensate for brain curvature, online field-curvature correction (FCC) was applied in ScanImage (Vidrio Technologies, MBF Biosciences) during mesoscope image acquisition. In addition, complementary fast-z 'sawtooth' or 'step' corrections were applied when all ROIs were acquired at the same or different z-planes, respectively (see *Sofroniew et al., 2016*). Despite this, small uncorrected discrepancies in the depth of the imaging plane below the pial surface may have remained, especially along the long-axis of acquisition ROIs in the side mount preparation (i.e. mediolateral). It is likely that optical effects of the curvature of the glass partially compensated for these differences in targeted imaging depth (i.e. by refraction to normalize approach of the laser to the pial surface), although we did not quantify this effect. In cases where we imaged multiple small ROIs, nominal imaging depth was adjusted in an attempt to maintain a constant relative cortical layer depth (i.e. depth below the pial surface).

We estimate that we experienced ~200 µm of depth offset across 2.5 mm. If the objective is orthogonal to our 10 mm bend window and centered at the apex of its convexity, a small ROI located at the lateral edge of the side mount preparation would need to be positioned around 200 µm below that of an equivalent ROI placed near the apex, and would be at close to the same depth as an ROI placed at or near the midline, at the medial edge of the window. We determined this by examining the geometry of our cranial windows in CAD drawings (available at https://github.com/vickerse1/mesoscope_spontaneous/tree/main/cranial_windows), and by comparing z-depth information from adjacent sessions in the same mouse, the first of which used a large FOV and the second of which used multiple small FOVs optimized so that they sampled from the same cortical layers across areas.

The mouse was restrained by fixation of the headpost with two countersunk screws to either a fixed (*Figure 1—figure supplement 1b*, top) or adjustable aluminum support arm (custom; *Figure 1a, b, Figure 1—figure supplement 1b*, bottom) and mounting apparatus (*Figure 1a and b*). For the dorsal mount preparation, two headpost support arms, one on each side of the head, were used (*Figure 1a, Figure 1—figure supplement 1c*, top; outer screw of both headpost wings used) while for the side mount preparation, a single, left side support arm was sufficient (*Figure 1—figure supplement 1*, bottom; inner and outer screws in the left headpost wing were both used). It was important to fix the headpost mounting arms to the top of the airtable with 1-inch diameter vertical mounting posts (Thorlabs RS6P8, RSH4, PF175), making sure all joints were clean of debris and tightened securely, to reduce movement artifacts.

To further minimize movement artifacts, it was also important to mount the mouse at a distance from the surface of the running wheel that was not too low, which would allow the mouse to push up with its legs, or too high, so that part of the mouse's body weight would be supported by the headpost. A final consideration to minimize movement artifacts was to ensure that proper, even pressure was maintained

across the entire interface between the brain and the cranial window during window implantation, both by making the craniotomy large enough so that the entire window could be set freely on the brain surface, and by using the 3D-printed window stabilizer properly during the application of flow-it glue to attach the edge of the window to the skull (see Protocols II, III).

Both preparations were found to be relatively free of vibration and movement artifacts and allowed for micro-adjustments (i.e. pitch, yaw, and roll) of the mouse relative to both the objective and the running wheel (*Figure 1a, b and c*). Increased positioning flexibility of the support-arm assembly was achieved through either of two ways: (1) by positioning it on a distal base-attached ball-joint mount (Thorlabs, SL20 articulating base; not shown); (2) through the use of a custom adjustable headpost support arm that was proximally adjusted around a ball-and-joint assembly (*Figure 1—figure supplement 1b*, bottom). This adjustable support arm operated through the use of a custom wrench and reverse-threaded (left-handed) nut for initial coarse tightening, and a micro-clamp for rotational stabilization during tightening. Four micro-hex wrench driven set-screws, two on either side of the ball-and-joint fitting, were used to achieve the final, fully locked state for imaging. The adjustable support arm was found to be superior to the base-attached ball joint mount in allowing for iterative micro-positioning of the animal in relation to the running wheel, objective, and lick spouts after initial fixation (mounting) of the headstage to the support bar.

In order to perform 2p neuronal imaging, the large 2p-RAM water-immersion objective (~10x net magnification, 0.6 NA, ~1.3 kg, 12 mm diameter tip, 25.6 mm shaft) must be positioned between 2.2 and 2.8 mm (i.e. ~working distance minus window thickness) from the curved glass surface of the cranial window while maintaining a stable water meniscus. Providing a base for the water meniscus while also blocking incident light entry was achieved by creating a custom 3D printed plastic light shield (AutoDesk Inventor, MakerGear M3 printer, PLA) that was attached to the protruding, fitted rim of the headpost with 170 FAST CURE Sylgard ('wok one'; *Figure 1c* and *Figure 1—figure supplement 1c, d*). A second light shield ('wok two') served to further block extraneous light entry into the objective and was attached to the base of the lower light shield ('wok one') by fitting of a U-shaped profile over a vertically protruding single edge profile along the perimeter of the first, lower light shield ('wok one'; *Figure 1c*).

Together, these two custom-printed light shields prevented incident light (from the video stimulus monitor, the ultraviolet LEDs used for controlling the baseline and dynamic range of pupil diameter, and the infrared LEDs used to illuminate the mouse) from entering the imaging objective and thereby either contaminating the image or tripping the photomultiplier tubes (PMTs). Line-of-sight for the animal to the video stimulus monitor was retained, and this design allowed for free vertical and rotational (over a limited range) movement of the objective lens, which was contacted directly only by the water meniscus.

Direct left, right, and posterior camera angles, or 'lines of sight', were preserved to enable reliable recording and proper illumination of pupil diameter, whisker pad movement, and the movement of other body parts such as the nose, mouth, ear, paws, and tail (*Figure 1a and b*; see *Videos 1 and 2*). Placement of the rotated mouse in the side mount preparation near the left side of the running wheel allowed for the animal's left and right fields of view to not be significantly obstructed. Adjustable positioning of two vertically and laterally offset conductance-based lick spouts for 2-alternative forced choice (2-AFC) task performance (lick left, lick right) and reward delivery was achieved with a rapidly translatable motorized linear stage (*Figure 1c*; 63.7ms total travel time over 7 mm; Zaber Technologies). This system allowed for rapid withdrawal of lick spouts between trials, and presentation of the lick spouts during the response period of each trial.

## Pseudo-widefield imaging and optogenetic stimulation hardware modifications of the Thorlabs 2-photon mesoscope

Examining neuronal activity at the single cell level and relating it to stimulus-evoked activity observed at the widefield level (*Figures 2 and 3*), as well as manipulating regions of the cortex through localized light delivery and optogenetics (see Video 11), required precise optical alignment across all three of these methods. First, a standardized cortical map needed to be fitted onto images of skull landmarks and the cortical vasculature by widefield multimodal sensory mapping (MMM; *Figures 2 and 3a, b, and c, top*). Then, a method was needed for transferring this cortical map onto the field of view of the mesoscope to both allow for online FOV targeting and for post-hoc assignment of each imaged neuron to a designated cortical area on the Allen common-coordinate framework (CCF v3.0) map for subsequent analyses (*Figures 2 and 3c, bottom, d*). Finally, the spatial coordinate system of the 1p opto-stimulation laser routed in through an auxiliary light-path of the mesoscope needed to be aligned to that of the 2p laser

so that specific cortical subregions could be targeted for optogenetic inhibition by light activated, ChR2-mediated excitation of parvalbumin interneurons in Thy1-RGECO x PV-Cre x Ai32 mice (Video 11).

In order to align 2p maps of single neuron activity with functional 1p cortical maps determined through pseudo-widefield imaging (i.e. combined reflected and fluorescence light imaging with a standard 'non-widefield' format CCD camera), we designed a method for reliably aligning a standardized cortical map (i.e. the Allen CCF v3.0; we used either a 0 degree, https://github.com/vickerse1/mesoscope_spontaneous/blob/main/python_code/CCF_map_rotation/0deg/CCF_MMM.png; *Vickers, 2024b*, or 22.5 degree, https://github.com/vickerse1/mesoscope_spontaneous/blob/main/python_code/CCF_map_rotation/22deg/CCF_MMM.png; *Vickers, 2024b*, rotated CCF map outline created with the following code: https://github.com/vickerse1/mesoscope_spontaneous/blob/main/python_code/CCF_map_rotation/Rotate_CCF.py; *Vickers, 2024b*), to an image of the cortical vasculature pattern, which can be directly visualized with both widefield and 2p imaging techniques for each brain (see Protocol V). The epifluorescence light source (Excelitas) for the pseudo-widefield imaging, which was parfocal with the 2p imaging coordinate system, was controllable by a dial, shutter, and remote foot-pedal for optimal ease of positioning the objective in the water meniscus at a distance of ~2.2 mm from the headpost and cranial window without crashing into the preparation (*Figure 1c* and *Figure 1—figure supplement 1d*, top and bottom right). This allowed us to 'drive' the position of the 2p mesoscope FOV to the desired Allen CCF cortical area by moving the objective to a location where the observed vasculature pattern matched that of a processed image we created ahead of time on our 1-photon widefield imaging rig that combined vasculature, sensory responses, and skull landmarks from each mouse.

This technique required several modifications of the auxiliary light-paths of the Thorlabs mesoscope, and would also likely involve similar modifications in other comparable microscope setups, such as Diesel 2p (*Yu et al., 2021*). For switchable blue/green widefield imaging and 2p imaging in our original configuration (see (a) in https://github.com/vickerse1/mesoscope_spontaneous/blob/main/mesoscope_optical_path_opto_switching.pdf; *Vickers, 2024b*), we used a 469/35 Semrock excitation filter, 466/40 Semrock dichroic (1st, top cube), and mirror (2nd, bottom cube). For the combination of pseudo-widefield imaging and rapid, targetable optogenetic 1-photon (1p) stimulation with concurrent 2p imaging (see (b) in https://github.com/vickerse1/mesoscope_spontaneous/blob/main/mesoscope_optical_path_opto_switching.pdf), which we used with PV-Cre x Ai32 x Thy1-RGECO mice (Video 11), we established dual coupling of the broadband fluorescence light-source (Excelitas) used in the original configuration and a 473 nm laser driver (SF4C 473, Thorlabs), coupled via liquid light guide, to converging, dual input auxiliary light paths of the Thorlabs mesoscope via two in-series, magnetically secured, switchable filter cubes (DFM1T1 cube, Thorlabs; see https://github.com/vickerse1/mesoscope_spontaneous/blob/main/mesoscope_filterCube_schematic_dual_opto_Jan0320.jpg; *Vickers, 2024b*; the top cube can be switched to allow pseudo-widefield imaging of either GCaMP6s or Thy1-RGECO, and the bottom cube can be switched to allow either pseudo-widefield imaging or optogenetic stimulation).

The 473 nm (blue) laser driver was connected to a single open loop, high speed buffer (50 LD, Thorlabs), and targeted to the coordinate system of the 2p laser with a grid-calibrated, auxiliary galvo-galvo scanner (GVSM002, Thorlabs). This allowed for 2p imaging and blue 1p laser optogenetic stimulation to occur pseudo-simultaneously, because rapid, electronic control of the PMT1 and PMT2 shutters was able to limit interruption of image acquisition to a brief period during presentation of the optogenetic stimulus (~100 ms; see Video 11; see https://github.com/vickerse1/mesoscope_spontaneous/blob/main/mesoscope_optical_path_opto_switching.pdf). Here, replacement of the switchable mirror behind the objective, which blocked 2p imaging during pseudo-widefield imaging in our original configuration, with a dichroic that, when positioned in the light path, allowed both 920 nm (2p excitation) and 473 nm (opto-excitation) light to reach the mouse brain without requiring any additional slow switching of optical components.

## Reduction of resonant scanner noise

Resonant scanners in 2p microscopes emit intense sound at the resonant mirror frequency, which is well within the hearing range of mice (~12.5 kHz emitted in the Thorlabs mesoscope). Because scanning precision requires the scanner to remain at a stable, elevated temperature, the scanner must remain on during the entire experimental session (i.e. up to ~2 hr per mouse). The unattenuated or native high-frequency background noise generated by the resonant scanner causes stress to both mice (*Sadananda et al., 2008*) and experimenters (*Fletcher et al., 2018*), and likely acts to prevent mice from achieving maximum

performance in auditory mapping, spontaneous activity sessions, auditory stimulus detection, and auditory discrimination sessions/tasks.

To reduce this acoustic noise, we encased the resonant scanner and attached light path tubes with a custom 3-dimensional (3D)-printed assembly containing dense interior insulating foam. See Supplementary methods and materials for diagrams: https://github.com/vickerse1/mesoscope_spontaneous/blob/main/resonant_scanner_baffle/closed_cell_honeycomb_baffle_for_noise_reduction_on_resonant_scanner_devices.pdf; *Vickers, 2024b*, and for a text description: https://github.com/vickerse1/mesoscope_spontaneous/blob/main/resonant_scanner_baffle/closed_cell_honeycomb_baffle_methodology_summary.pdf; *Vickers, 2024b*; '3D scanning and honeycomb patterned nylon print' (University of Oregon Innovation Disclosure #DIS-23–001, US provisional patent application UOR-145-PROV). It was critical to use 3D scanning of encased components in the design of the noise reduction shield so that the sound-reduction assembly would closely follow all of the surface contours and fit accurately in spaces with tight tolerances.

The result of this encasement was a large reduction in resonance scanner sound, from ~60 dB to ~5 dB measured at the head of the mouse (Bruel and Kjaer ¼" pressure filled microphone 4938 A-011; i.e. below the mouse hearing threshold at 12.5 kHz of roughly 15 dB; *Zheng et al., 1999*). By comparison, encasements designed using standard 3D-design and printing techniques (i.e. not based on a 3D scan) were, in our hands, only able to achieve a noise reduction of ~30 dB, to a level still audible to both mice and humans. This difference was due to the enhanced precision of encasement fit enabled by using the 3D-scanned map of the microscope's surface contours.

## Cortical alignment to the common coordinate framework (CCF) v3.0 map

Interpretation of the diversity of activity of thousands of neurons simultaneously identifiable with the mesoscope requires proper cortical areal localization, including that of areas anatomically or functionally distant from primary sensory cortices. Such mapping is not routinely possible directly on the 2p mesoscope when performing neuronal-level imaging, due to both the heterogeneity of responses within primary sensory cortices at the single neuron level, and to animal-to-animal variations in the spatial extent of cortical regions (*de Vries et al., 2020*; *Bimbard et al., 2023*). To facilitate the assignment of neurons to cortical areas, we sought to align the Allen Institute Common Coordinate Framework (CCF v3.0; *Wang et al., 2020*) to our 2p neuronal imaging results, using blood vessels, skull landmarks, and widefield imaging responses as intermediaries. We used an overlaid image of these features that we refer to as the 'multimodal map' (MMM) as on-the-fly guidance for FOV placement during 2p mesoscope imaging sessions, and then performed a precise post-hoc CCF alignment based on the MMM and vasculature patterns observed during 2p imaging to assign a unique CCF area identifier to each Suite2p-identified ROI/neuron during preprocessing stages of our data analysis (see https://github.com/vickerse1/mesoscope_spontaneous/tree/main/matlab_code for creation of the MMM, and https://github.com/vickerse1/mesoscope_spontaneous/tree/main/python_code for generation of rotated CCF map outlines and precise neuron assignment to CCF areas, which takes place in the following jupyter notebook: https://github.com/vickerse1/mesoscope_spontaneous/blob/main/python_code/mesoscope_pre_proc/meso_pre_proc_1.ipynb).

Alignment of the CCF to the dorsal surface of cortex and subsequent image-stack registration based on widefield imaging Ca$^{2+}$ fluorescence data has been performed elsewhere with a variety of techniques. These include the use of skull landmarks (e.g. bregma and lambda; *Musall et al., 2019*), responses to unimodal sensory stimuli (*Gallero-Salas et al., 2021*), visual field mapping (*Zhuang et al., 2017*), and/or autocorrelation maps of spontaneous activity (*Peters et al., 2021*). Few studies, however, have performed such alignments with rotated cortex (i.e. imaging lateral portions of the cortex at an angle; but, see *Esmaeili et al., 2021*).

To summarize our approach, we used a technique where, for both the dorsal mount (*Figure 2*, *Figure 1—figure supplement 2a*) and side mount (*Figure 3*, *Figure 1—figure supplement 2b*) preparations, we first imaged 1p widefield GCaMP6s responses to unimodal passive sensory stimulation (e.g. visual, auditory, and somatosensory) to create a MMM consisting of multiple sensory area masks on top of the cortical vasculature. We then overlaid skull landmarks (e.g. bregma and lambda), imaged with reflected green light, onto the MMM. This intermediate overlay image, created using custom MatLab code to extract mean widefield dF/F sensory responses (https://github.com/vickerse1/mesoscope_spontaneous/blob/main/matlab_code/SensoryMapping_Vickers_Jun2520.

m; *Vickers, 2024b*), followed by z-projection and selection masking techniques in Fiji/ImageJ, was used to guide selection of the FOV during 2p mesoscope imaging sessions. The Allen CCF was then warped onto the vasculature/skull/MMM overlay, and the mean 2p image was warped onto the MMM by vasculature alignment using custom code (see:https://github.com/vickerse1/mesoscope_ spontaneous/blob/main/python_code/mesoscope_pre_proc/meso_pre_proc_1.ipynb for full initial processing steps of 2p data, or https://github.com/vickerse1/mesoscope_spontaneous/blob/main/ python_code/mesoscope_preprocess_MMM_creation.ipynb; *Vickers, 2024b* for a short notebook containing only the relevant CCF alignment steps - as with all of our jupyter notebooks, these should be run in the 'uobrainflex' python environment: https://github.com/sjara/uobrainflex; *Jaramillo, 2020*). Finally, each neural ROI in the 2p image was assigned, post-hoc, to a position in the MMM x-y coordinate system and a corresponding CCF area based on the final overall alignment of the MMM, CCF, and 2p image. The following sections describe these steps in more detail.

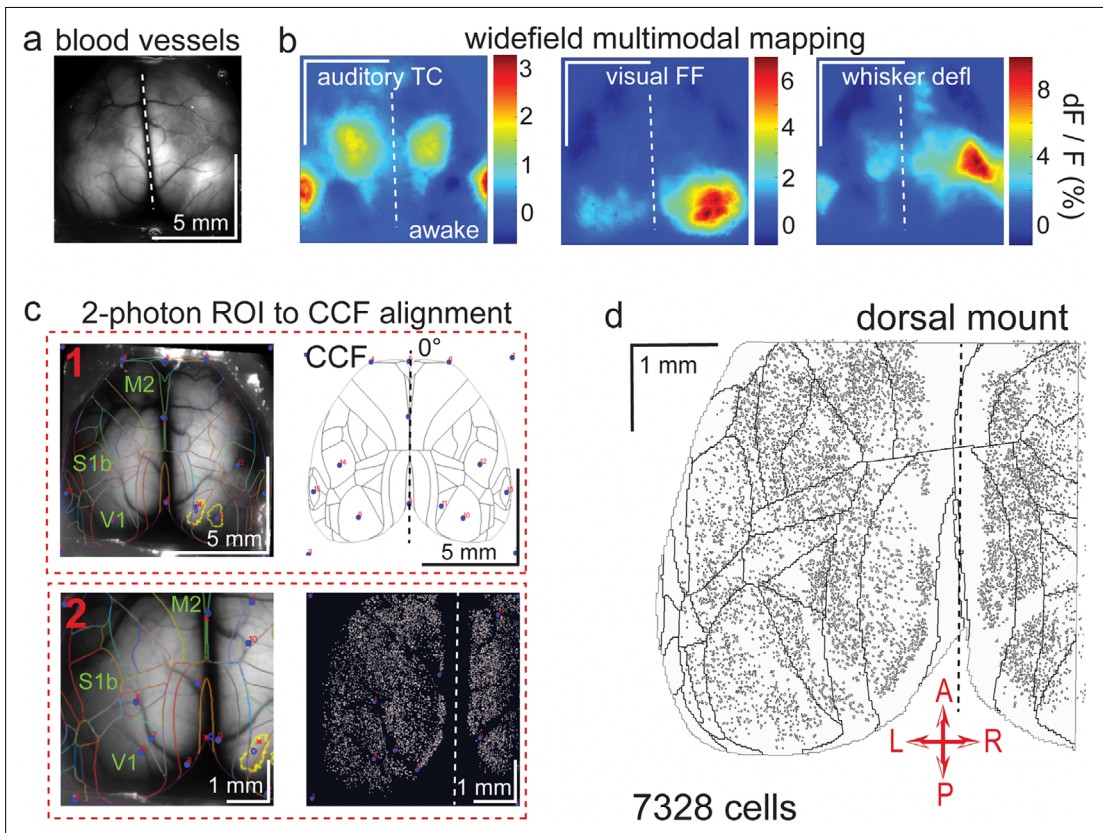

**Figure 2.** Widefield-imaging-based multimodal mapping (MMM) and CCF alignment, ROI detection, and areal assignment of GCaMP6s[+] excitatory neurons for the dorsal mount preparation. (**a**) Mean projection of 30 s, 470 nm excitation epifluorescence widefield movie of vasculature performed through a dorsal mount cranial window. Midline indicated by a dashed white line. (**b**) Mean 1 s baseline subtracted dF/F images of 1 s responses to ~20–30 repetitions of left-side presentations of a 5–40 kHz auditory tone cloud (auditory TC; left), visual full field (FF) isoluminant Gaussian noise stimulation (center), and 100 ms x 5 Hz burst whisker deflection (right) in an awake dorsal mount preparation mouse. (**c**) Example two-step CCF alignment to dorsal mount preparation, performed in Python on MMM masked blood-vessel image (upper left), rotated outline CCF (upper right), and Suite2p region of interest (ROI) output image containing exact position of all 2p imaged neurons (lower right). Yellow outlines show the area of masks created from the thresholded mean dF/F image for, in this example, repeated full-field visual stimulation. In step 1 (top row), the CCF is transformed and aligned to the MMM image using a series of user-selected points (blue points with red numbered labels, set to matching locations on both left and right images) defining a bounding-box and known anatomical and functional locations. In step 2 (bottom row), the same process is applied to transformation and alignment of Suite2p ROIs onto the MMM with user-selected points defining a bounding box and corresponding to unique, identifiable blood-vessel positions and/or intersections. Finally, a unique CCF area name and number are assigned to each Suite2p ROI (i.e. neuron) by applying the double reverse-transformation from Suite2p cell-center location coordinates, to MMM, to CCF. (**d**) CCF-aligned Suite2p ROIs from an example dorsal mount preparation with 7328 neurons identified in a single 30-min session from a spontaneously behaving mouse. TC = tone cloud, FF = full-field, defl = deflection, dF/F = change in fluorescence over baseline fluorescence, CCF = Allen common coordinate framework version 3.0, M2 = secondary motor cortex, S1b = primary somatosensory barrel cortex, V1 = primary visual cortex, A = anterior, P = posterior, R = right, L = left.

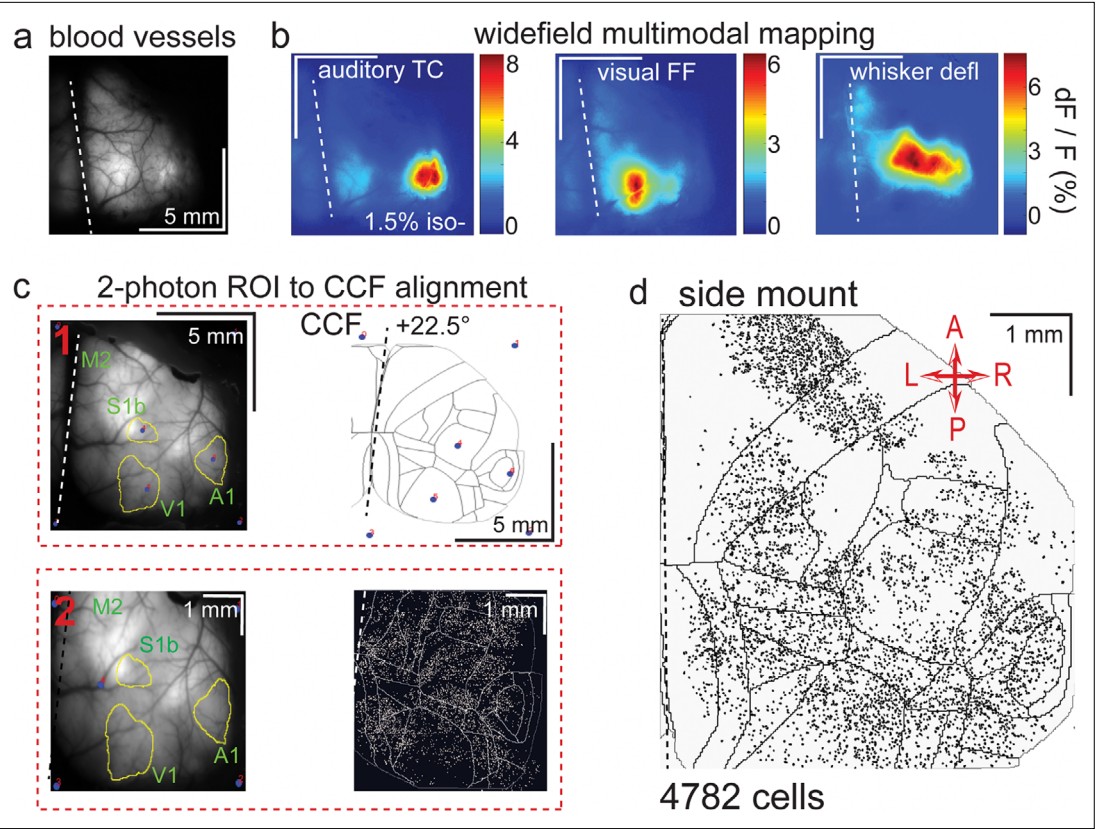

**Figure 3.** Widefield-imaging-based multimodal mapping (MMM) and CCF alignment, ROI detection, and areal assignment of GCaMP6s⁺ excitatory neurons for the side mount preparation. (**a**) Mean projection of 30 s, 470 nm excitation fluorescence widefield movie of vasculature performed through a side mount cranial window. Midline indicated by a dashed white line. (**b**) Mean 1 s baseline subtracted dF/F images of 1 s responses to ~20–30 repetitions of left-side presentations of a 5–40 kHz auditory tone cloud (auditory TC; left), visual full field (FF) isoluminant Gaussian noise stimulation (center), and 100 ms x 5 Hz burst whisker deflection (right) in an example side mount preparation mouse under 1.5% isoflurane anesthesia. (**c**) Example two-step CCF alignment to side mount preparation, performed in Python on MMM masked blood-vessel image (upper left), rotated outline CCF (upper right), and Suite2p region of interest (ROI) output image containing exact position of all 2p imaged neurons (bottom right). Yellow outlines show the area of masks from thresholded mean dF/F image for repeated auditory, full-field visual, and/or whisker stimulation. In step 1 (top row), the CCF is transformed and aligned to the MMM image using a series of user-selected points (blue points with red numbered labels, set to matching locations on both left and right images) defining a bounding-box and known anatomical and functional locations. In step 2 (bottom row), the same process is applied to transformation and alignment of Suite2p ROIs onto the MMM with user-selected points defining a bounding box and corresponding to unique, identifiable blood-vessel positions and/or intersections. Finally, a unique CCF area name and number are assigned to each Suite2p ROI (i.e. neuron) by applying the double reverse-transformation from Suite2p cell-center location coordinates, to MMM, to CCF. (**d**) CCF-aligned Suite2p ROIs from an example side mount preparation with 4782 neurons identified in a single 70 min session from a mouse performing our 2-alternative forced choice (2-AFC) auditory discrimination task. TC = tone cloud, FF = full-field, defl = deflection, dF/F = change in fluorescence over baseline fluorescence, CCF = Allen common coordinate framework version 3.0, M2 = secondary motor cortex, S1b = primary somatosensory barrel cortex, A1 = primary auditory cortex, V1 = primary visual cortex, A = anterior, P = posterior, R = right, L = left.

## Widefield multimodal mapping

To create the multimodal map, each mouse was put under light isoflurane anesthesia (~1.5%) after head-post surgery, but before implantation of the cranial window, and exposed to 5 min each of full-field visual (vertical and horizontal stationary grating patches; 0.16 cpd, 30 deg; *Michaiel et al., 2019*), tone-cloud auditory (a series of overlapping 30 ms duration tones randomly selected from a frequency range of 5–40 kHz and presented at 100 Hz; *Xiong et al., 2015*), and piezo-driven whisker deflection (or, in some cases, forelimb and trunk stimulation; see *Gallero-Salas et al., 2021*). For whisker deflection, a 1 s, 5 Hz burst of five 100 ms duration forward sweeps was used (i.e. each sweep consists of 100ms of forward movement and 100 ms of backward movement), consisting of posterior to anterior sweeps. The whisker deflector was a custom 3D-printed triangular polylactic acid (PLA) piece mounted on a 21-gauge needle and attached to a PL140.11 piezo-actuator (PI Ceramic) with epoxy glue. It was actuated with a Physik-Instrumente controller driven by a custom pulse sequence

generated in Spike2 and delivered from an analog output of a CED Power 1401, with an inter-stimulus interval of ~10 s (*Figure 2b*, right, and *Figure 3b*, right; see also *Videos 3–7*).

Light anesthesia was used to minimize unwanted cortical activity due to spontaneous movements and arousal fluctuations, and to prevent the spread of cortical sensory responses to areas downstream of primary sensory areas. Imaging through the skull between the headpost and cranial window implantation surgeries was done to allow for coregistration of skull landmarks with vasculature and sensory responses, and in general yielded more contiguous, easily interpretable multimodal maps than the same widefield mapping done through the cranial window. Although the resulting overlay of vasculature and the multi-modal sensory map was useful for cranial window placement, it was not strictly necessary. Additional MMM sessions performed through the cranial window were performed every 30–60 days or as necessary due to slight changes in vasculature. In general, vasculature was stable throughout the entire ~150–200 day lifespan of a successful cranial window preparation (see https://github.com/vickerse1/mesoscope_spon-taneous/blob/main/window_preparation_stability.pdf; *Vickers, 2024b*), although in some cases slight increases in the diameter of major blood vessels (e.g. sagittal sinus), or outgrowth of fine arteriole endings, were observed.

Averaged, baseline-subtracted dF/F responses (*Figures 2b and 3b*) were thresholded, masked, outlined, and layered onto the blood vessel image (*Figure 2a, c*, *Figure 3a, c*, *Figure 1—figure supple-ment 2a, b*) along with bregma and lambda skull landmarks (*Figure 1—figure supplement 1a*) iden-tified with 530 nm (green) reflected-light skull-imaging using a custom protocol/macro in Fiji (ImageJ; *Figure 2c*, *Figure 3c*, *Figure 1—figure supplement 2a, b*; *Videos 3–7*). This multimodal mapping proce-dure and alignment to the CCF and skull landmarks was repeated if significant changes in vasculature pattern occurred over the days/weeks of the experiment. More precise maps of visual cortical areas can be achieved through visual field mapping using a topographic stimulus consisting of a bar sweeping in azimuth or elevation (*Garrett et al., 2014*). This technique was not applied here, because our goal was global alignment of the Allen Institute cortex-wide CCF to our dorsal or side mount views, and not precise alignment to visual subareas per se.

## Co-alignment of 2-photon image to vasculature and CCF

The MMM was aligned to an overlay of the rotated CCF edges and region of interest (ROI) masks using custom Python code and the built-in function 'PiecewiseAffineTransform', given a user-supplied series of bounding-box and alignment points common to both images (*Figure 2c* (upper, part 1), 3 c (upper, part 1)). A second, similar alignment was then performed between the multimodal map and all neural ROI locations (i.e. Suite2p-identified neurons) relative to the 2p image plane; here, vasculature is inferred from the pattern of gaps (e.g. vascular 'shadows') in the spatial distribution of neurons (*Figure 2c* (lower, part 2), 3 c (lower, part 2)), and can be confirmed by parfocal pseudo-widefield (i.e. combined reflected and epiflu-orescence light) imaging directly on the 2p Thorlabs mesoscope. Note that in 2p imaging below the pial surface, the effective/apparent width of the vasculature, or its "shadow", is significantly larger than that of the actual blood vessel, and its width increases with depth. The transforms resulting from each of these two alignments were applied in serial (i.e. multimodal map to CCF, then multimodal map to 2p neural image; *Figures 2c and 3c*) to overlay the neural image directly onto the cortical map, using outlines, and to assign a unique CCF area identifier (i.e. name and ID number) to each Suite2p-extracted, 2p-imaged neuron using CCF masks (*Figures 2d and 3d*). Note that, while the final assignment of CCF area identifiers to each Suite2p-identified neuron (*Figure 2c* (2), right) was performed post-hoc, the intermediate blood-vessel/MMM overlay image (*Figure 2* (1) and (2), left) was used for online guidance of 2p mesoscope FOV selection during imaging sessions.

## 2-photon imaging across broad regions of the dorsal and lateral cortex in behaving mice

Previous 2p imaging studies have demonstrated that arousal and/or orofacial/body movement can explain a significant proportion of the variance in spontaneous and sensory evoked neural responses in visual and other restricted dorsal cortical regions (*Musall et al., 2019*; *Stringer et al., 2019*). Here, as a proof of principle, we examined the generality of this finding by simultaneously monitoring neuronal activity across broad regions of bilateral dorsal cortex, with our dorsal mount preparation, and both dorsal and lateral cortex across the right hemisphere, in our side mount preparation. Previous studies suggest that there may be both commonalities as well as heterogeneity in the effects of changes in arousal/movement on

spontaneous and sensory-evoked responses across different cortical areas (*McGinley et al., 2015*; *Musall et al., 2019*; *Stringer et al., 2019*). For example, running typically enhances the gain of visually evoked responses in the mouse primary visual cortex (*Niell and Stryker, 2010*), while it significantly decreases that of evoked auditory responses in primary auditory cortex (*Zhou et al., 2014*; *McGinley et al., 2015*).

Assessing such potential differences between functionally distinct, and potentially distant, cortical areas required simultaneous imaging across multiple cortical regions while maintaining adequate imaging speed, quality, and resolution. To achieve this level of 2p imaging across several millimeters of cerebral cortex (e.g. from visual to motor or auditory cortical areas), we first sought to optimize the parameters of our mesoscope imaging methods and protocols.

Conventional 2p imaging using a preparation similar to our dorsal mount ("Crystal Ckull"; *Kim et al., 2016*) previously employed serial acquisition of ~1 x 1 mm FOVs (*Kim et al., 2016*). Other 2p Thorlabs mesoscope imaging studies have used acquisition protocols targeting z-stacks of 600x600 µm FOVs in barrel cortex (3 planes at 7 Hz, 1.17 µm/pixel; *Peron et al., 2015*), 900x935 µm FOVs in visual cortex (11 planes at ~1 Hz; *Stringer et al., 2019*), and three adjacent 600x1800 µm FOVs in CA1 hippocampus (*Sun et al., 2023*; bioRxiv). Here, with our dorsal and side mount preparations, we were able to routinely image between 2000 and 7600 neurons per session over up to ~25 mm$^2$ of dorsal cortex simultaneously at ~3 Hz with a resolution of 5 µm/pixel in GCaMP6s mice (combined total in both preparations for all large-FOV sessions: N=17 mice, n=91 sessions, ~350,000 neurons; *Figure 1—figure supplement 2e*, *Video 8*).

In order to achieve spatially broad and dense sampling, while monitoring changing activity in each neuron as frequently as possible, we typically tiled seven mediolaterally aligned 5000x660 µm FOVs (i.e. mechanical scanning along a long-axis oriented mediolaterally, combined with fast resonant-scanning across multiple short-axes oriented anterior-posterior) over either posterior-medial, posterior-lateral, or anterior dorsal, cortex at a pial depth of ~200–300 µm (cortical layers 2/3). Bidirectional scanning (with a scan-phase correction of between -0.9 and -1.0) and field-curvature correction were typically enabled, with a 1-5 ms 'flyback' and 'frame' time, which are defined in ScanImage as the time delay for retargeting of the laser from the end of one line or frame acquisition to the beginning of the next, respectively. Here, there is a trade-off between longer flyback times, which reduce imaging speed but minimize positioning errors at the beginning of each scan line or ROI, and shorter values, which accelerate overall imaging acquisition speed but increase these positioning errors.

We typically imaged GCaMP6s fluorescence in awake, behaving mice at an excitation wavelength of 920 nm with 60–95 mW power calibrated at the pia over 20–90 min sessions, with little to no bleaching or change in baseline fluorescence. Suite2p reliably extracted neurons with >0.9 classifier probabilities, and nearly completely (i.e. to less than 0.1 pixels) removed x-y movement artifacts with uncorrected, raw principal components of up to ~1.3 pixels (i.e. up to ~15% of a 20–40 µm, or 4–8 pixels, diameter neuron at an imaging resolution of 0.2 µm/pixel, reduced to less than 0.5 µm or 1–2% of the diameter of a neuron after combined rigid and non-rigid motion correction; see Video 11).

To further control for z-movement artifacts (whose correction is intractable post-acquisition under normal scanning conditions) we also occasionally used GPU-enabled online fast-z motion correction (42/52 large FOV side mount sessions, and 9/39 large FOV dorsal mount sessions, used online z-correction) with ScanImage (Vidrio Technologies, LLC). This was only possible, in our hands, during multi-frame imaging when all frames were acquired at the same z-level. Quality of post-hoc motion-corrected dF/F traces and correlations with behavioral variables did not appear, by eye, to be different under these conditions (*Figure 4—figure supplement 1a vs b*; see https://github.com/vickerse1/mesoscope_spontaneous/tree/main/online_fast_z_correction; *Vickers, 2024b*), as both types of sessions contained similar mixtures of neurons with both walk-bout dependent and independent activity, for example. Because the z-motion correction did not lead to large qualitative differences in neural activity and its relationship to behavioral primitives, both types of session are used interchangeably here.

In order to confirm neural region-of-interest (ROI) selection (i.e. each neuron as identified by a spatiotemporal footprint in Suite2p), increase scan speed, and correct our cortical layer 2/3 targeting for the effects of brain curvature, we also routinely imaged in random access mode with 4–6 small FOVs (660–1320 x 660 µm) positioned over visual (V1), somatosensory (S1), retrosplenial (RSpl), posterior parietal (PPC), and/or motor (MOs_pm, MOs_am, and/or MOs_al - ALM) cortices at independently controlled z-levels, with an imaging speed of 4–10 Hz and a resolution of 1.0–2.0 µm/pixel (see Protocol I; *Figure 1—figure supplement 2d, e*). These multi-FOV sessions yielded hundreds

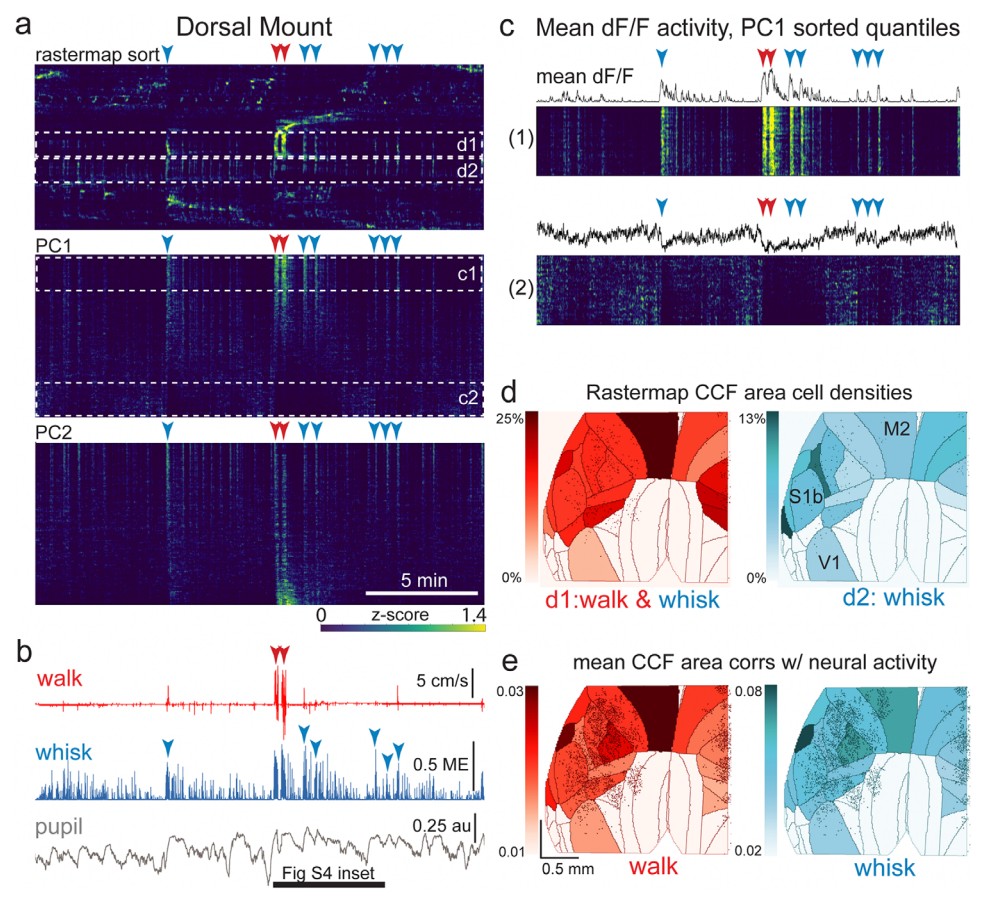

**Figure 4.** The relationship between spontaneous behavioral measures and neural activity across the dorsal cortex is globally heterogeneous. (**a**) Rastermap (top), first principal component (PC1; middle), and second principal component (PC2, bottom) sorting of normalized, rasterized, neuropil subtracted dF/F neural activity for 3096 cells from a single 20-min duration example dorsal mount 2p imaging session. Each row in the display corresponds to a 'superneuron', or average activity of 50 adjacent neurons in the Rastermap sort (***Stringer et al., 2023***, bioRxiv). Activity for each superneuron is separately z-scored, then displayed as low activity in blue, intermediate activity in green, and maximum activity level in yellow (standard viridis color look-up table). Red and blue arrowheads show alignment of walking and whisking bouts, respectively, to neural activity. White dashed boxes indicate dual selections highlighted in panels (**c**) and (**d**). Scale bar shows color look-up map for each separately z-scored, individually displayed superneuron in the raster displays. (**b**) Behavioral arousal primitives of walk speed, whisker motion energy, and pupil diameter shown temporally aligned to the rasterized neural activity traces in (**a**), directly above. Horizontal black bar indicates time of expanded inset in ***Figure 4—figure supplement 2***. (**c**) Expanded insets of top and bottom fifths of rasterized PC1 sorting from the middle segment of panel (**a**), with mean dF/F activity traces shown above each. Red and blue arrowheads indicate the same walking and whisking bouts, respectively, as in (**a**) and (**b**). (**d**) Normalized density of neurons in each CCF area belonging to two example Rastermap sorted groups (d1, left, red: MIN = 0%, MAX = 25%; d2, right, blue: MIN = 0%, MAX = 13%), with rasterized activities shown in corresponding labeled white dashed boxes in the top segment of panel (**a**). The neurons present in selected Rastermap groups are shown as Suite2p footprint outlines. The type of behavioral arousal primitive activity typically concurrently active with high neural activity for each Rastermap group is indicated below each Rastermap group's CCF density map (i.e. walk and whisk for group d1, left (red), and whisk for group d2, right (blue)). (**e**) Normalized mean correlations of neural activity and walk speed (left, red; mean: MIN = 0.01, MAX = 0.03) and whisker motion energy (right, blue; mean: MIN = 0.02, MAX = 0.08) per CCF area (i.e. average correlation of all neurons in each area) are shown for this example session. Mean walk speed correlations with neural activity (dF/F) were significantly greater than zero (p<0.001, median t(3095)=3.7, single-sample t-test; python: scipy.stats.ttest_1samp) for 15 of the 19 CCF areas with at least 20 neurons present. The areas with mean correlations not significantly larger than zero were right VISam, right SSp_ll, left VIS_rl, and left AUDd. Mean whisker motion energy correlations with neural activity (dF/F) were significantly more than zero (p<0.001, median t(3095)=4.4, single-sample t-test) for 16 of the 19 CCF areas with at least 20 neurons present. The areas with

*Figure 4 continued on next page*

*Figure 4 continued*

mean correlations not significantly larger than zero were right VISam, left VISam, and left VISrl. PC = principal component, ME = motion energy, au = arbitrary units, M2=secondary motor cortex, S1b = primary somatosensory barrel cortex, V1 = primary visual cortex.

The online version of this article includes the following figure supplement(s) for figure 4:

**Figure supplement 1.** Comparison of neurobehavioral alignments with and without online z-motion correction.

**Figure supplement 2.** Alignment of neural activity sorted Rastermap groups and spontaneous behavior in an example dorsal mount session.

---

of high quality neurons per FOV, at a density roughly twice that of the lower resolution sessions (*Figure 1—figure supplement 2c vs d*).

Uncontrolled image motion due to the movement of behaving animals is a significant impediment to image quality (*Pachitariu et al., 2016*). We have taken significant care in our headpost design and fixation (see *Figure 1a, b*, *Figure 1—figure supplement 1a–f*, and Supplementary methods and materials) so that our preparations are highly stable, such that motion-corrected or 'registered' movies showed very little to no residual movement in either large FOV, low-resolution, or multiple small FOV, high-resolution sessions (*Video 9*; Video 11). Furthermore, movement-related neural activities across neighboring detected ROIs in large FOV, low-resolution sessions were correlated but non-identical upon detailed examination, and differed from dF/F $Ca^{2+}$ fluorescence fluctuations detected in nearby non-ROI and blood-vessel regions of similar cross-sectional area (see *Video 9*; Video 11), thus strongly suggesting that it was non-artifactual. In theory, movement artifact signals should be relatively time-locked or nearly synchronous across cortical regions during vigorous walking or movement. However, we did not observe this in our preparations.

These findings, therefore, gave us confidence that neural activity changes detected during spontaneous movements of the mouse were due to real changes in $Ca^{2+}$ fluorescence within each neuron, and not to movements of the brain causing the neuron to shift into or out of each Suite2p-defined ROI and leading to artifactual fluorescence transients (see *Video 9*; Video 11, which use blood vessels and inactive fluorophore/neuropil, respectively, as examples of the lack of significant movement-driven artifactual transients in GCaMP6s(-) structures in our preparations). In other studies, performed using the same setup and methods as those applied here, we have demonstrated that non-activity dependent fluorescent markers, such as the expression of a non-activity dependent fluorescent marker mCherry in cholinergic or noradrenergic axons, or autofluorescent blebs, exhibit no significant movement-related changes in fluorescence, indicating that movement artifacts are minimal in our preparations (*Collins et al., 2023*). It should be noted that axonal diameters are on the order of ~1 µm or less - this is significantly smaller than the diameter of the neuronal cell bodies (~15–30 µm) imaged here. Axons, therefore, are a more stringent test of the stability of our imaging preparations.

## Modulation of neural activity by spontaneous movements and arousal as observed with the dorsal mount preparation

Recent work has shown that sorting fluorescence based 2p GCaMP cortical neuronal activity with an algorithm called Rastermap is a powerful method to reveal clusters of cells that exhibit similar patterns of activity (*Stringer et al., 2023*, bioRxiv). An example of such sorting of our data for a dorsal mount session of 3096 neurons is shown in *Figure 4a* (top). This method for one-dimensional nonlinear embedding works by sorting and clustering of ROIs (i.e. neurons) by similarity of neural activity patterns across multiple timescales, using k-means clustering, sorting by asymmetric similarity as defined by peak cross-correlations at non-negative timelags, and upsampling of individual cluster activities in principal components space to allow sorting within clusters (*Stringer et al., 2023*, bioRxiv). Here, sorting with Rastermap revealed clustering of neurons with clearly distinguishable activity patterns that exhibited heterogeneous relationships to the behavioral primitive arousal/movement variables of walk speed, whisker motion energy, and pupil diameter (*Figure 4a, top, and b*; *Video 8*).

The activity generated by these thousands of neurons was clearly not random, with sub-groups of cells exhibiting similar activity patterns. Many neurons exhibited activity that appeared to be coupled to walking/whisking (*Figure 4a* top, b). It has previously been shown that the first principal component

of cortical neuronal activity is highly correlated with movement (*Stringer et al., 2019*). Indeed, sorting our neurons based on degree of activity loading onto their first two principal components (i.e. neurons at the top of the sort have the largest amount of activity accounted for by the first principal component of activity across the entire population) revealed several distinct patterns of activity. The first principal component, PC1 (*Figure 4a*, middle) demonstrated a clear relationship to walking/whisking, such that some neurons were strongly activated in association with movement (e.g. *Figure 4a and c1 in PC1, middle, and c, top*), while other neurons appeared to be deactivated during movement bouts (e.g. *Figure 4a and c2 in PC1, middle, and c, bottom*).

Closer examination revealed that some manually selected Rastermap and PC1 clusters were active primarily during walking (*Figure 4a*, top, d1, *Figure 4a*, middle, c1, and *Figure 4*, d1) and/or whisking, with either transient or sustained activity, while others were either active during low arousal periods (*Figure 4a*, middle, c2) or appeared to display a combination of ON/OFF transient whisking-related activity (*Figure 4a*, top, d2, and *Figure 4d*, right; see *Figure 4—figure supplement 2*). In the illustrated example, alignment of marked increases and decreases in bulk population neural activity to eight brief walking bouts and whisking bouts in the absence of walking (red and blue arrowheads, respectively; *Figure 4a–c*), both of which were accompanied by pupil dilations, was more clearly seen following ROI sorting by first and second principal components (*Figure 4a* middle and bottom; PC1 and PC2, respectively, *Figure 4c*) than in Rastermap sorting (*Figure 4a*, top). In particular, PC1 seemed to sort neurons by transient walk and whisk activity, while PC2 appeared to sort neurons according to a more sustained component of whisking activity.

Interestingly, separation of the top and bottom fifth of neurons in the example data set sorted by PC1 (*Figure 4a* middle, c1 and c2, respectively; *Figure 4c* top and bottom, respectively) showed large groups of neurons with clear positive (*Figure 4c1*, top) and negative (*Figure 4c2*, bottom) correlations with arousal/movement, as demonstrated by alignment of rasterized neural activity with walk and whisker transients, and local maxima in pupil diameter (see *Figure 4b*). These separate groups of 'ON' and 'OFF' neurons were not as well identified when sorting by PC2. Specifically, neurons with high PC1 values (*Figure 4c1*) showed elevated activity during periods of increased arousal/movement, while neurons with low PC1 values (*Figure 4c2*) showed depressed activity levels during these periods. Subsets of these neurons showed either elevated basal activity with a suppression of activity during periods of enhanced arousal and/or movement (*Figure 4c2*, top of Rastermap), or transient increased activity leading up to movement/arousal (*Figure 4c2*, bottom of Rastermap), followed by transient suppression (see *Video 8*).

Examination of individual 'raw' dF/F traces confirmed that these differences were not due to artifacts of z-scoring or superneuron averaging (*Stringer et al., 2023*, bioRxiv) in the PC and Rastermap sorted displays (not shown). The term 'superneuron' here refers to the fact that when Rastermap displays more than ~1000 neurons it averages the activity of each group of adjacent 50 neurons in the sort to create a single display row, to avoid exceeding pixel limitations of the display. Each single row representing the average activity of 50 neurons is referred to as a 'superneuron' (*Stringer et al., 2023*; bioRxiv).

## Spatial distribution of neurons related to movement/arousal

Performing simple correlations between dF/F in each neuron with either whisker movements, walking, or pupil diameter revealed a broad range of values (*Figure 5—figure supplement 2b*; whisk –0.4 to +0.6, pupil –0.6 to +0.6, and walk –0.2 to +0.4; same mouse as *Figure 4*, but a different session). The population distributions of these correlations were highly skewed (positive skew for whisk and walk, negative skew for pupil diameter) with modes at small positive values (*Figure 5—figure supplement 2a, b*). Plotting the spatial location of neurons by correlation value within the cortical map area revealed a broad distribution across the dorsal cortex of neurons whose activity was either strongly negatively or positively (or in between) correlated with walking, whisking, and/or pupil diameter, inclusively across every CCF subregion imaged (*Figure 5—figure supplement 2a–c*).

At the local spatial scale (*Figure 5—figure supplement 2a*, colored dots indicate individual neurons), neurons whose GCaMP6s activity was strongly positively correlated with movement or pupil diameter could often be found near neurons whose activity exhibited a wide variety of correlations, from strongly positive to strongly negative. Plotting the average correlation value for neurons within a CCF region revealed modest yet statistically significant (see *Figure 5—figure supplement 2c*, legend)

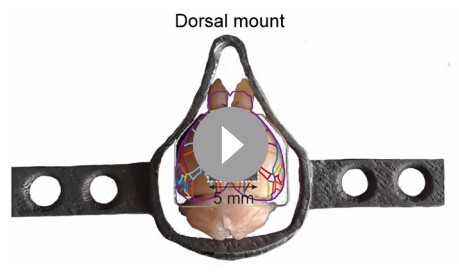

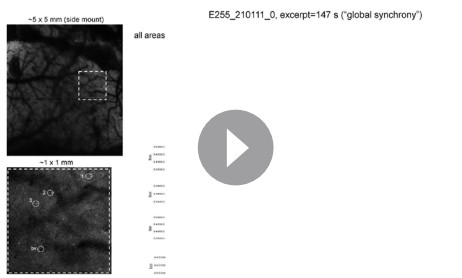

**Video 8.** Dorsal mount. Titanium 3D printed headpost shown from above, with paraformaldehyde-fixed mouse brain shown beneath custom cranial window, colored Allen CCF outlines, and an example 5 x 4.62 mm mesoscope FOV corresponding to data in this video shown inside an inset box indicated by a white dashed line. Next segment, left, full 5 x 4.62 mm ScanImage (Vidrio) rendered mesoscope 2-photon field of view, rotated and flipped so that the top corresponds to the front of the mouse and the left corresponds to the left hemisphere of the cerebral cortex. Black horizontal joining lines indicate the 'seams' between adjacent ROIs where they are joined by rendering. The resonant scanner moves along the short axis of each rectangle, perpendicular to the 'seams', and mechanical galvanometers move along the long axis, parallel to the 'seams'. Midline vertically oriented (i.e. from posterior, at bottom, to anterior, at top) blood vessel (sinus) prominent at center of frame. A movie (100 s duration shown) acquired with unidirectional scanning at 1.62 Hz, smoothed with a running average of three frames, is shown at 3x playback rate. Center top, expanded inset from white dashed box (left), shown synchronized with larger video. Bottom center, high-resolution left face videography of mouse from this session, synchronized with 2p video, rasterized neural data (upper right), and Spike2-recorded (CED) behavioral data (whisking motion energy in blue, pupil diameter in gray, and walk speed in red). Neural data was sorted in Suite2p (*Stringer et al., 2019*) with Rastermap (top), or by its first (middle) or second (bottom) principal component of activity over the entire session, and is displayed as z-scored (normalized) dF/F neuropil subtracted activity indexed to a single color-map lookup table with yellow as maximum, green as intermediate, and blue/purple as minimum activity level (i.e. GCaMP6s fluorescence; same lookup table scale as in *Figure 4a*). Red arrowheads indicate co-alignment of transient arousal increases accompanied by walking, whisking, and pupil dilation, with diverse changes in neural activity across rastermap, PC1-, and PC2-sorted ensembles.

https://elifesciences.org/articles/94167/figures#video8

**Video 9.** Example two-dimensional field of view (FOV) rendered movie and insert from an example session where factorial hidden Markov modeling showed near global synchrony of PC1 loading during walking bouts in a side mount mouse. A 5 x 5 mm full FOV movie (~10x real-time) is shown at top left, with a 1 x 1 mm inset (white dashed box) expanded and shown at bottom left. Three example Suite2p-extracted neuron ROIs (1, 2, and 3) and a blood vessel ROI (bv) are indicated with white dashed circles. Gray traces at right indicate mean pixel fluorescence intensity of all pixels within each of these ROIs over ~450 2p movie frames. Rasterized rendering at top right indicates PC1 loading (blue/green/yellow = low/medium/high) across all CCF areas contained in the 2p movie (one per row). The red horizontal lines at the end of the shown epoch are a graphical rendering artifact that was not removed. White/gray/black rows indicate the current state of each of four hidden factors (i.e. state 0 = white, state 1 = gray, and state 2 = black) in a factorial hidden Markov model (fHMM) fit to the first 15 PCs of global neural data, the blue row indicates walk speed, the next, split blue row indicates left and right whisker pad motion energy, and the red, bottom row indicates pupil diameter. Actual behavioral arousal and movement primitives are shown as traces below (walk = red, whisk = blue, pupil = gray). Note that walk-related activity changes are correlated with independent fluctuations across the three example neurons, and furthermore that blood vessel fluctuations are small and not movement-locked, consistent with the idea that neurobehavioral activity alignment in this case, even with detected transient global synchrony across the first PC of activity, is not driven by brain movement artifact.

https://elifesciences.org/articles/94167/figures#video9

spatial heterogeneities (*Figure 4e*, *Figure 5—figure supplement 1*) that were evident both in each of two different single sessions (*Figure 4e*; *Figure 5—figure supplement 2a, b*) and across sessions (*Figure 5—figure supplement 2c*, mean of 8 sessions from the same mouse).

To examine the spatial distribution of different Rastermap groups, we created heatmaps of the density of cells in each CCF area that belong to a particular Rastermap group (i.e. percent of cells in each region belonging to that Rastermap group; *Figure 4d*; only areas with at least 20 cells were considered). A Rastermap group with increased activity in relationship to walking and whisking bouts (Rastermap cell group d1 in *Figure 4a*, top, and *Figure 4—figure supplement 2a*) had a CCF density

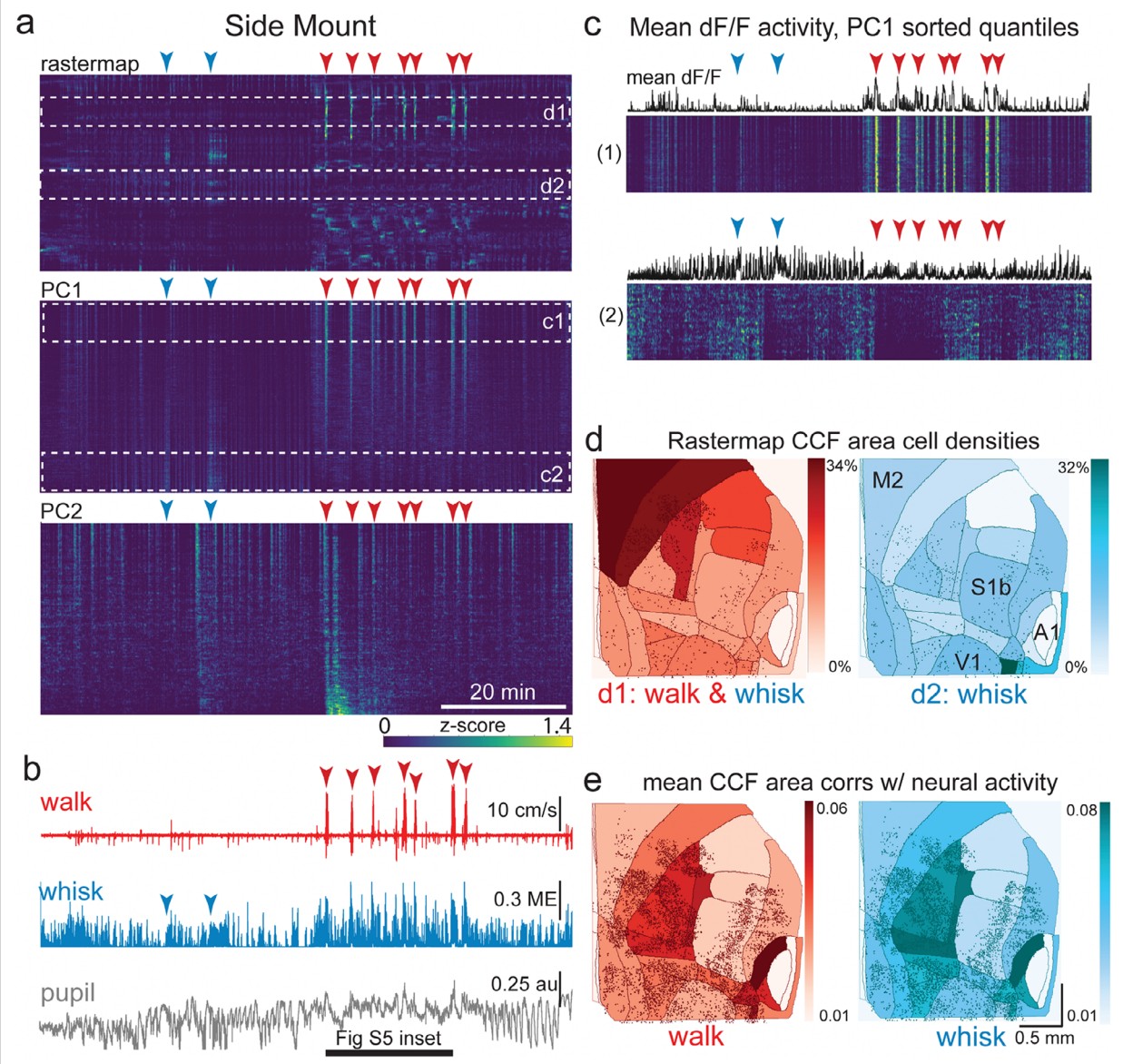

**Figure 5.** The relationship between spontaneous behavioral measures and neural activity across the lateral cortex is regionally patterned. (**a**) Rastermap (top), first principal component (PC1; middle), and second principal component (PC2, bottom) sortings of normalized, rasterized, neuropil subtracted dF/F neural activity for 5678 cells from a single 90 min duration example side mount 2p imaging session. Each row in the display corresponds to a 'superneuron', or average of 50 adjacent neurons in the Rastermap sort (*Stringer et al., 2023*, bioRxiv). Low activity (z-scored) in blue, intermediate activity in green, maximum activity level in yellow. Red and blue arrowheads show alignment of walking and whisking bouts, respectively, to neural activity. White dashed boxes indicate selections highlighted in panels (**c**) and (**d**). Scale bar shows color look-up map for each separately z-scored, individually displayed superneuron in the raster displays. (**b**) Behavioral arousal primitives of walk speed, whisker motion energy, and pupil diameter shown temporally aligned to the rasterized neural activity traces in (**a**), directly above. (**c**) Expanded insets of top and bottom fifths of rasterized PC1 sorting from the middle segment of panel (**a**), with mean activity traces shown above each. Red and blue arrowheads indicate the same walking and whisking bouts, respectively, as in (**a**) and (**b**). Horizontal black bar indicates time of expanded inset shown in *Figure 5—figure supplement 1*. (**d**) Normalized density of neurons in each CCF area belonging to two example Rastermap sorted groups (d1, left, red: MIN = 0%, MAX = 34%; d2, right, blue: MIN = 0%, MAX = 32%), with rasterized activities shown in corresponding labeled white dashed boxes in the top segment of panel (**a**). Only cells in selected Rastermap groups are shown. The type of behavioral arousal primitive (i.e. walk and whisk, left, in red; whisk, right, in blue) that was typically concurrently active with high neural activity is indicated below each Rastermap group's CCF density map. (**e**) Normalized mean correlations of neural activity and walk speed (left, red; mean: MIN = 0.01, MAX = 0.06; standard deviation: MIN = 0.000, MAX = 0.029) and whisker motion energy (right, blue; mean: MIN = 0.01, MAX = 0.08; standard deviation: MIN = 0.000, MAX = 0.043) per CCF area for this example session. Mean walk speed correlations with neural activity (dF/F) were significantly more than zero (p<0.001, median t(5677)=4.4, single-sample t-test; python: scipy.stats.ttest_1samp) for 20 of the 24 CCF areas with at least 20 neurons present. The areas with mean correlations not significantly larger than zero were left VISp, right SSpn,

*Figure 5 continued on next page*

*Figure 5 continued*

right AUDpo, and right TEa. Mean whisker motion energy correlations with neural activity (dF/F) were significantly more than zero (p<0.001, median t(5677)=7.0, single-sample t-test) for 21 of the 24 CCF areas with at least 20 neurons present. The areas with mean correlations not significantly larger than zero were right SSpn, right AUDpo, and right TEa. PC = principal component, ME = motion energy, au = arbitrary units, M2 = secondary motor cortex, S1b = primary somatosensory barrel cortex, V1 = primary visual cortex, A1 = primary auditory cortex.

The online version of this article includes the following figure supplement(s) for figure 5:

**Figure supplement 1.** Alignment of neural activity sorted Rastermap groups and spontaneous behavior in example side mount session.

**Figure supplement 2.** Heterogeneity of arousal and movement correlations with neural activity in the dorsal and side mount preparations.

distribution ranging from 0 to 25% and was concentrated in select motor, somatosensory and visual subareas (MOs, SSn, VISrl, and VISa; CCF neural density map; *Figure 4d*, left). Here, CCF density distribution refers to the full set of neural density percentages (i.e. percentage of neurons in each CCF area that belong to a particular Rastermap group) across all CCF areas present in a given preparation. A second example Rastermap group, d2 (*Figure 4a*, top, and *Figure 4—figure supplement 2c*) exhibited increases in fluorescence in relationship to whisking, but not as strikingly during bouts of walking. This cell group had a CCF density distribution ranging from 0 to 13% and was concentrated in select somatosensory and auditory subareas (SSun, SSb, and AUDd; *Figure 4d*, right).

Next, we compared, across all CCF areas, the proportion of neurons within each CCF area that exhibited large positive correlations with walking speed and whisker motion energy. In this example dorsal mount session, walk and whisk related neurons formed the largest relative proportion of cells in select CCF areas such as motor and somatosensory subareas (MOs, SSll, and SSn; *Figure 4e* left and right, respectively). Mean correlations for each CCF area were generally statistically significantly greater than zero (see *Figure 4e*, legend).

## Modulation of neural activity by spontaneous movements and arousal as observed with the side mount preparation

To extend the potential diversity of neural activity patterns during spontaneous behavior, and to examine the generality of the observation of movement/arousal related neuronal activity to the lateral cortex, we monitored neuronal activity in spontaneously behaving mice with our lateral window preparation. We imaged ~85 sessions in the side mount preparation (20–90 minutes each, including both large- and multi-FOV sessions) across 11 GCaMP6s mice, for a total of ~320,000 neurons (*Figure 1—figure supplement 2e*; *Video 10*). In an example imaging session (*Figure 5*) of a mouse implanted with our custom 3D-printed titanium side mount headpost (*Figure 1e*, *Figure 1—figure supplement 1f*) and 10 mm radius bend cranial window (*Figure 1e*, *Figure 1—figure supplement 1a, d, f*; *Video 1*), we were able to image 5678 neurons at 5x5 µm/pixel resolution over a combined 5.0x4.62 mm FOV (*Figure 5*). For the ScanImage online fast-z motion correction documentation corresponding to this example session, see the .csv files for 3056_200924_E235_1: https://github.com/vickerse1/mesoscope_spontaneous/tree/main/online_fast_z_correction; *Vickers, 2024b*. Note here that, although there were no neurons present in the primary auditory cortex in the example session in *Figure 5*, such neurons were present in other sessions from the same mouse (*Figure 5—figure supplement 2d–f*), and in other side mount preparation mice in general (e.g. *Figure 3d*).

As in the dorsal mount (*Figure 4*), Rastermap sorting of neurons across this single side mount example session (*Figure 5*) was performed in order to further explore the broad range of relationships between movement/arousal measures and neural activity. As in the dorsal mount preparation, Rastermap sorting showed non-random patterns of neuronal activity that were often related to movement/arousal measures. Here, we identify and further characterize both a cluster with increased activity in relationship to bouts of walking/whisking (*Figure 5a*, top, cell group d1; see *Figure 5—figure supplement 1a, b*), and another manually selected neural ensemble whose activity appeared, by eye, to be more related to whisking than walking, per se (*Figure 5a*, top, cell group d2; see *Figure 5—figure supplement 1c, b*). Interestingly, these two groups consisted of ~300 neurons each spread across the right lateral cortex, with some neurons nearby in the same CCF areas, and others in distant, functionally distinct regions (*Figure 5d*, left, right).

Sorting by the first principal component (PC1) of activity, as in our dorsal mount preparation (*Figure 4a*, middle), revealed that a majority of imaged neurons exhibited activity related to bouts

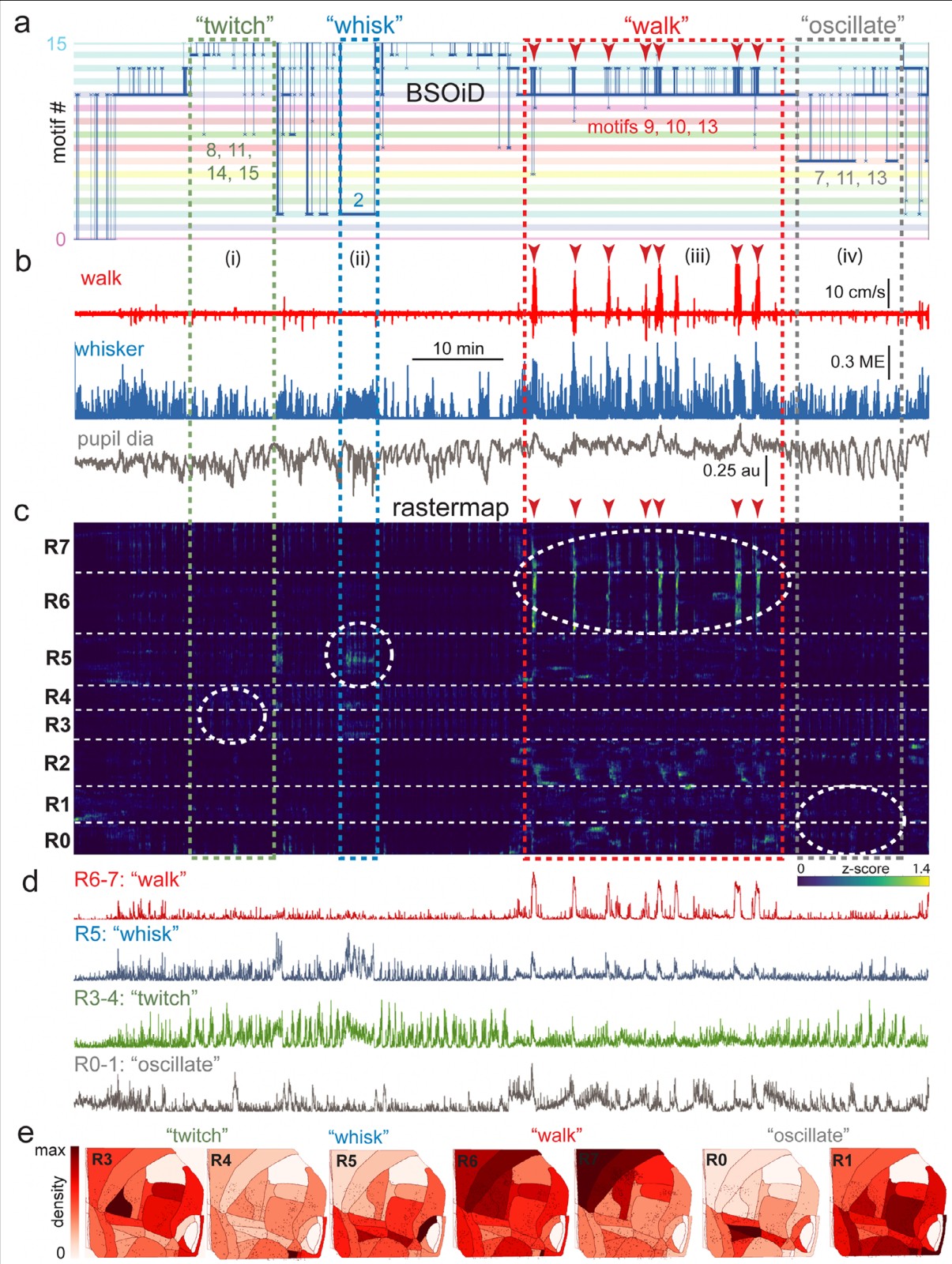

**Figure 6.** Alignment of activity sorted neural ensembles and spontaneous behavioral motifs reveals sparse, distributed encoding of arousal measures across dorsolateral cortex in the side mount preparation. (**a**) Manual identification of high-level, qualitative behaviors (twitch, green dashed box; whisk, blue dashed box; walk, red dashed box; pupil oscillate, gray dashed box), aligned to sets of raw, unfiltered behavioral motifs (B-SOiD) extracted from pose-estimates (DeepLabCut) for a single example session. Numbered B-SOiD motifs (trained on a set of four sessions from the same mouse, motif

*Figure 6 continued*

# indicated by the position of the thick blue line relative to the y-axis at each time point) are color-coded (horizontal bars) across the entire 90-min session. (**b**) High-level behaviors (dashed colored boxes, (i–iv) and BSOiD motifs from a) are vertically temporally aligned to the behavioral primitive movement and arousal measures of walk speed, whisker motion energy, and pupil diameter. Expanded alignments of high-level behaviors ii (whisk) and iii (walk) are shown in *Figure 6—figure supplement 1a, b*, Red arrowheads indicate temporal alignment positions for multiple walk bouts across BSOiD motifs (**a**), behavioral arousal primitives (**b**), and Rastermap sorted neural data (**c**). (**c**) Rastermap sorted, z-scored, rasterized, neuropil subtracted dF/F neural activity (10th percentile baseline, rolling 30 s window) from the same side mount session, temporally aligned to the behavioral data directly above. Numbers at left (R0–R7) indicate Rastermap motif numbers, selected manually 'by-eye' for this session, and the horizontal dashed white lines on the Rastermap-sorted neural data indicate the separation between neighboring Rastermap motifs. Each row in the display corresponds to a 'superneuron', or average of 50 adjacent neurons in the Rastermap sort (*Stringer et al., 2023*, bioRxiv). Colored, dashed boxes indicate alignment of high-level behaviors with neural activity epochs from the same example session. White dashed ovals indicate areas of Rastermap group-aligned active neural ensembles during periods of defined high-level behaviors. The 2p sampling rate for this session was 3.38 Hz. (**d**) Normalized mean activity traces for all neurons in each neural ensemble indicated in (**c**) (white dashed ovals). Each trace also corresponds and is aligned to the indicated Rastermap groups. (**e**) Color-coded neuron densities per CCF area corresponding to Rastermap groups shown in (**c**), shown grouped by qualitative high-level behaviors, as indicated in (**a, b**). Cross-indexing with neurobehavioral alignments (white dashed ovals) in (**c**) allows for visualization of spatial distribution of neurons active during identified high-level behaviors (**a**) consisting of defined patterns of behavioral primitive movement and arousal measures (**b**). Corresponding maximum cell density percentages in each CCF area (white to red scale bar, top right) for each normalized Rastermap (R0–7, excluding R2) heatmap color lookup table are 11.4, 31.8, 22.9, 24.1, 38.4, 33.2, and 24.0%, respectively (left to right: R3, R4, R5, R6, R7, R0, and R1). This list of cell density percentages, therefore, corresponds to that of the CCF area filled with the darkest shade of red in each Rastermap group. White CCF areas in each Rastermap group are areas where no cells of that group were found in this example session, or where the total number of cells was less than 20 and therefore the density estimate was deemed unreliable and not reported. ME = motion energy, au = arbitrary units, CCF = common coordinate framework.

The online version of this article includes the following figure supplement(s) for figure 6:

**Figure supplement 1.** Example expanded alignment of high-level behaviors and primary corresponding BSOiD motifs, arousal measures, and Rastermap sorted neural data from side mount preparation, same example session as in *Figures 5 and 6*.

of walking/whisking (*Figure 5a*, middle; walk bouts indicated by red arrowheads, whisk bouts indicated by blue arrowheads; see *Video 10* and *Video 11* ). The top fifth of these movement-activated neurons showed activity nearly completely dominated by walk-related signals (*Figure 5a*, middle, c1, and 5 c, top, (1)), while the bottom fifth showed activity negatively correlated with walking (*Figure 5a*, middle, c2, and 5 c, bottom, (2)). Sorting by the second principal component (PC2), on the other hand, appeared to show increases in activity more closely aligned to the onset of whisking movements (*Figure 5a*, bottom). Interestingly, some neurons in PC2 exhibited sustained inter-walk bout activation (*Figure 5a*, bottom).

As in our dorsal mount example session (*Figure 5—figure supplement 2a–c*), we examined the overall spatial pattern of correlations in a side mount example session, which in this case had 3897 recorded neurons (*Figure 5—figure supplement 2d–f*). Here, correlations of individual neuron activity with whisker motion energy, pupil diameter, and walk speed ranged from –0.6 to +0.6 (*Figure 5—figure supplement 2d, e*; whisk –0.4 to +0.6, pupil –0.4 to +0.4, and walk –0.2 to +0.5) with positively skewed distributions (except for pupil diameter) and modes at small positive values (*Figure 5—figure supplement 2d, e*). Strong heterogeneity of these correlations, as in the dorsal preparation example (*Figure 5—figure supplement 2a–c*), was evident at both local (*Figure 5—figure supplement 2d, e*, single session) and regional (*Figure 5—figure supplement 2f*, mean of 8 sessions from the same mouse) spatial scales.

Examining the regional variations in mean correlations between neural activity and walk speed (*Figure 5—figure supplement 2f*, bottom, in red), pupil diameter (*Figure 5—figure supplement 2f*, middle, in gray) and whisker motion energy (*Figure 6—figure supplement 1f*, top, in blue) showed that neurons with high correlations to arousal/movement tended to concentrate in M1 and somatosensory upper and lower limb (as one might expect for movement related activity) and dorsal auditory areas in this example session (dark red, gray, and dark blue, respectively), while neurons with lower correlations tended to concentrate in somatosensory barrel, nose, and mouth regions (light red, gray, and blue, respectively). As in the dorsal preparation (*Figure 5—figure supplement 2c*), the mean correlations across 8 sessions from this example side mount mouse (i.e. the grand mean for each CCF area, equal to the mean of all means for that area across all 8 individual sessions) were general statistically significantly greater than zero (see *Figure 5—figure supplement 2f*, legend).

## Behavioral video analysis and neurobehavioral alignment

From the above results, we can see that there are clear, but complex, relationships between behavior and neuronal activity across the dorsal and lateral cortex of the mouse. Principal component analysis of this activity suggests that one major factor is variations in behavioral state, as indicated by movement/arousal (see also *Musall et al., 2019*; *Steinmetz et al., 2019*; *Stringer et al., 2019*). Indeed, previous studies have demonstrated that the use of dozens or more of the top principal components of the motion energy of facial movement videos is beneficial in explaining variance in visual cortical activity (*Stringer et al., 2019*). However, beyond the first few principal components of facial movement, the precise behavioral meaning of the subsequent components are difficult to discern (*Stringer et al., 2019*).

Here, we took a different approach by using a multi-step semi-automated analysis pipeline to identify higher order behavioral motifs and to examine the relationship of these motifs to the diversity of patterns of neural activity observed in the cortex.

To test the feasibility of this approach, we first created pose estimates consisting of (x,y) positions of a set of user-labeled body-parts and joints from our left, right, and posterior mouse videos using DeepLabCut (DLC; *Mathis et al., 2018*; see *Video 1*). These pose estimate outputs were used for unsupervised machine learning extraction of behavioral motifs by an algorithm called 'Behavioral segmentation of open field in DeepLabCut' ('B-SOiD'; *Hsu and Yttri, 2021*; see *Video 10*). To identify high-level patterns of spontaneous head-fixed mouse behavior, we next performed manifold embedding ('UMAP') and cluster identification ('HDBSCAN') of DLC pose tracking outputs in BSOiD. A random-forests machine classifier was trained with these clusters to be able to predict, or identify, behavioral motifs in any test session with the same set or subset of labeled body part pose estimates (i.e. x-y positions for a certain number of body parts with the same names), either within or between mice.

In an example side mount session (same session as *Figure 5*), we identified 16 BSOiD behavioral motifs that exhibited good coarse and fine alignment to the dynamics of behavioral arousal primitives (*Figure 6a*), including walking, and manually selected discrete Rastermap groups from the same session. Here, BSOiD was trained on a randomly selected 80% subset of four

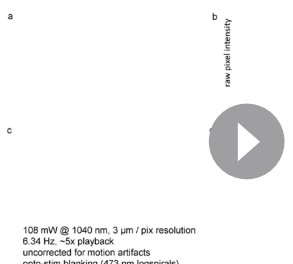

108 mW @ 1040 nm, 3 μm / pix resolution
6.34 Hz, ~5x playback
uncorrected for motion artifacts
opto-stim blanking (473 nm logspirals)

**Video 11.** Example simultaneous dual color mesoscope acquisition shows independence of activity-driven $Ca^{2+}$ transients from movement artifacts. (a), (c) Simultaneous dual color acquisition of activity-independent EYFP (top) and activity-dependent RCaMP (bottom; Thy1-RGECO). Dashed white circles indicate ROIs whose mean pixel intensities are shown in (b), (d). (b) Mean raw pixel intensity for the two ROIs shown in (a). The blue arrow indicates an example of blanking during 473 nm optogenetic stimulation (i.e. PMT is shuttered during 473 nm laser stimulation, so that section of trace is removed because it does not reflect actual fluorescence activity from the mouse cortex), and the red arrow shows an example of walking-induced movement of the imaged FOV. (d) Same as in (b), but for 3 ROIs shown in (c). Note that the magnitude of the movement artifact related transient in the $Ca^{2+}$ indicator fluorescence (i.e. the red channel, PMT2) is relatively unchanged relative to adjacent peaks in ROI1, an identified neuron, and is very small or nonexistent in other neuropil and blood vessel ROIs, and in the yellow anatomical channel (PMT1). PMT = photomultiplier tube.

https://elifesciences.org/articles/94167/figures#video11

Side mount

**Video 10.** Side mount. Same as in *Video 8*, but for an example session from a side mount mouse instead of a dorsal mount mouse. Data from a 100 s contiguous segment of single session, played at 3x original speed (3.06 Hz 2-photon acquisition, bidirectional scanning with 3-frame running average applied). Right cortex is shown (5 x 4.62 mm total at 5 μm resolution per pixel in x and y), with anterior at front (top), auditory cortex at bottom right, and midline at left edge of frame. Behavioral videography shown is taken from the right side and shows the entire front end of the mouse including torso, paws, ears and face. Time scaling is the same as in *Video 8*.

https://elifesciences.org/articles/94167/figures#video10

sessions from the same mouse across three days, including the target session itself, with minimum cumulative motif times set to 1% of total video time or ~100 s, and then used to label individual video frames from the entirety of the target session. We found that, in some cases, training on individual sessions resulted in poor identification of rare and or brief behaviors, such as walking (which occurs rarely in some sessions; not shown). Initial attempts to train BSOiD on sessions across mice failed, likely due to differences in their behavioral repertoires. However, we were able to identify a subset of repeating behavioral motifs across mice. Also, the number of motifs per session and per mouse were comparable and tended to exhibit state transitions on similar timescales.

Examination of four types of qualitatively identified behavioral epochs ('twitch', 'whisk', 'walk', and 'pupil oscillate') at a finer temporal scale showed good alignment of BSOiD motifs with fine-scale features of spontaneous head-fixed mouse behavior at the level of movement and arousal primitives (*Figure 6a and b*, *Figure 6—figure supplement 1*). For example, balancing wheel movements during 'twitch' seem to align well to flickering into BSOiD motifs 8 and 11, while flickering into motif 13 during 'walk' appears to correspond well to individual periods of high walk speed during the extended walking bout (see *Figure 6—figure supplement 1b*). The qualitative labels that best describe these behaviors ('twitch', *Figure 6a–c*; 'whisk', *Figure 6a–c* and *Figure 6—figure supplement 1*; 'walk', *Figure 6a–c* and *Figure 6—figure supplement 1b*; 'pupil oscillate', *Figure 6a–c*) did not appear to align with or be clearly explained by simple 'first-order' interaction effects of movement and arousal primitives (i.e. whisk/walk/pupil x ON/OFF), despite the fact that the above-mentioned, closer, albeit preliminary, analysis of fine-scale behavioral features revealed precise and reliable alignments. This suggests that these methods will yield novel and important insights into the organization of neurobehavioral alignment across cortex, because other factors that are currently poorly understood appear to be contributing to the alignments.

Alignment of the bulk activity of all neurons in each Rastermap group with BSOiD motif transitions for this example session (*Figure 6a–c*, colored, vertically oriented dashed boxes) showed differential alignment with the above-mentioned four high-level, manually identified qualitative behaviors ('twitch', 'whisk', 'walk', and 'pupil oscillate'). Note that, although individual BSOiD motifs could occur across multiple qualitatively identified higher level behaviors, each behavior consisted of a repeating pattern of a restricted set of motifs. For example, motif 13 in this session seems to correspond to a syllable that might best be termed 'symmetric and ready stance'. This tended to occur just before and after walking, but also during rhythmic wheel balancing movements that appear during the 'oscillate' behavior.

The occurrences of these behaviors, then, were clearly aligned with both behavioral movement/arousal primitives (*Figure 6b*) and the activity of putative, qualitatively segmented neural ensembles (*Figure 6c–d*) that displayed varied spatial distributions across defined CCF areas of the side mount preparation (*Figure 6e*).

For example, the qualitatively identified high-level behavior 'whisk', which in this case corresponds to whisking in the absence of walking (*Figure 6a and b*), with relatively elevated pupil diameter, aligns with strong activity in Rastermap group 5 (*Figure 6c–d*, white dashed oval corresponding to Rastermap group #5, mean trace in blue), which has a higher density of neurons in retrosplenial, secondary visual, somatosensory barrel, and dorsal auditory CCF areas (*Figure 6e*, R5). On the other hand, the qualitatively identified high-level behavior 'walk', which in this case refers to neurons with strong, transient activity aligned to seven individual bursts of walking during an extended walking bout, aligns with strong activity in Rastermap groups 6 and 7 (*Figure 6c–d*, white dashed oval corresponding to Rastermap groups #6 and 7, mean trace in red), which has a higher density of neurons in primary and secondary motor cortices, upper and lower limb somatosensory cortices, and primary visual cortex.

Note that the qualitative behaviors identified in *Figure 6* do not correspond directly to the Rastermap group labels used in earlier figures to point out possible alignment between neural activity and behavioral primitives. For example, the 'whisk' group in *Figure 5d*, right, corresponds to a subset of the neurons with strong neural activity aligned to the 'twitch' behavior in *Figure 6c* (R3 and R4), and the walk and whisk group in *Figure 5d*, left, corresponds to a subset of the neurons with strong neural activity aligned to the 'walk' behavior in *Figure 6c* (R6 and R7). Furthermore, these single session results are preliminary and are not intended, at this point, as a result that necessarily generalizes across sessions or across mice. Rastermap maximum neuron densities can be found in the corresponding

figure legend, but we did not apply tests of statistical significance here for these reasons. However, with ~30 CCF areas, a uniform distribution of neurons in a given Rastermap group would be ~3.3% per area. One would expect, therefore, that some of the neural density values we observed, which ranged up to ~40% for area M2 in Rastermap group 7 corresponding to the 'walk' qualitative behavior, would be highly significant if they were to be observed consistently across sessions.

We expect that, in general and based both on our preliminary results and emerging trends in existing literature, that the predictive contribution of high level behavioral motifs relative to movement and arousal primitives will be greater both in lateral cortical areas, such as A1, in non-primary CCF areas further up the cortical hierarchy, such as secondary sensory cortices and M2 (*Wang et al., 2023*; bioRxiv), and over intermediate and longer timescales of activity aligned with purposeful and/or goal oriented movements organized at higher levels of the behavioral hierarchy (*Mimica et al., 2023*).

## Discussion

### Overview and main findings

We developed novel headpost designs, surgical procedures, 3D-printed accessories, experimental protocols, and analysis pipelines that allowed us to routinely perform 2p imaging of up to ~7500 neurons simultaneously across ~25 mm$^2$ of either bilateral parieto-occipital ('dorsal mount') or unilateral temporo-parietal ('side mount') cortex in separate populations of awake, behaving mice, while simultaneously monitoring an array of global arousal state variables. Correlations between neural activity and arousal/movement were broadly, but heterogeneously, represented across both local and regional spatial scales. Preliminary analysis showed that this rich dataset could be used to segregate specific mappings between high-level, qualitatively identified behaviors (e.g. 'twitch', 'whisk', 'walk', and 'pupil oscillate'; *Figure 6a*), each consisting of a persistent pattern of robustly identifiable, high-level behavioral motifs, low-level behavioral arousal, and movement primitives (i.e. walk, whisker, and pupil; *Figure 6b*), and spatially localized neural activity clusters (i.e. 'Rastermap' groups; *Figure 6c–e*). Further analysis of such mappings will both allow for the fine dissection of patterned behaviors and neural activity, and for the development of a powerful framework for the sophisticated prediction of performance in mice engaged in a variety of multimodal sensory discrimination tasks (see *Hulsey et al., 2023*; bioRxiv, and *Wang et al., 2023*; bioRxiv).

Taken together, these findings show the strong potential of our novel methods for further elucidation of the principles of pan-cortical neurobehavioral alignment at the level of densely sampled individual neurons. For example, the spatial distributions of densities of neurons in each Rastermap (*Stringer et al., 2023*, bioRxiv) group across CCF areas were partially overlapping, yet strongly dissociable (see *Figures 4d, 5d, and 6e*). This suggested that dynamic, recurring patterns of arousal-dependent reorganization might act to enable activity mode switching between various distributed, cortical functional 'communities'. Each of these communities, then, could be specialized for different forms of activity related to various aspects of spontaneous behavior or task performance. This topic should be further explored with principled statistical techniques for recursive, top-down hierarchical community detection (*Li et al., 2020*).

In summary, we have shown here that it is feasible to monitor individual neuronal activities across broad expanses of the cerebral cortex and to perform neurobehavioral alignment of high resolution behavioral arousal state motifs and pan-cortical, activity-clustered neural ensembles in awake, behaving mice. Furthermore, our preliminary findings suggest a detailed alignment between both arousal/movement primitives and high-level behavioral motifs that appears to vary across broad regions of cortex (see *Figure 6*). These findings extend those of earlier studies that showed widespread encoding of behavioral state during spontaneous behavior in visual cortex (*Stringer et al., 2019*; imaging and Neuropixels recordings) and encoding of uninstructed movements during task performance both across dorsal cortex (*Musall et al., 2019*; widefield imaging across dorsal cortex, 2p imaging in restricted FOVs in primary visual and secondary motor cortex) and brain-wide during task performance (*Steinmetz et al., 2019*; Neuropixels recordings at low volumetric sampling density).

In particular, our findings are consistent with those of *Stringer et al., 2019* in that we observe widespread, strong correlations of cortical neural activity with the behavioral state variables of movement (facial and locomotor) and pupil diameter. However, we imaged larger areas of cortex simultaneously, including medial, anterior, and lateral cortical areas, at a higher overall imaging frequency

(~3 Hz vs. ~1 Hz). Although admittedly this imaging speed is still not fast enough to record fast cortical encoding of sensory information or decision making, these factors, taken together, may have contributed to our ability to observe enhanced heterogeneity across these non-primary cortical areas, including in neurons whose activities were strongly suppressed during periods of increased arousal/ movement (see *Figure 4c*, 2). Higher concentrations of such neurons were evident over lateral regions of cortex such as primary auditory and somatosensory nose and mouth regions, and in anterior regions including secondary motor cortex.

We also showed that high-level behaviors, manually identified and aligned to clusters of repeating motifs by UMAP embedding (BSOiD; *Hsu and Yttri, 2021*), aligned well with patterns of neural activity identified by Rastermap corresponding to neural ensembles with non-uniform spatial distributions (*Figure 6*). This suggests that encoding of behavioral state across cortex may occur at multiple levels of the behavioral hierarchy, in addition to that of 'low-level' behavioral arousal/movement primitives such as pupil diameter, whisker motion energy, and walking speed (see *Mimica et al., 2023*).

Our finding that neurons with no correlation to arousal/movement, or with a large negative correlation, are also widespread across cortex, suggests that multiple mechanisms or pathways may exist in parallel to distribute information about behavioral state across the brain. Interestingly, as evidenced by our discovery with Rastermap of neural ensembles exhibiting diverse response kinetics and polarities (*Figure 4a, top, d*, *Figure 5a, top, d*, and *Figure 6c–e*), these pathways may be activated with different temporal dynamics simultaneously across various brain areas, thus making our approach for simultaneous recording of many areas one of the keys to potentially understanding the nature of their neuronal activity. The reason for this is that correlations between behavior and neural activity across different cortical regions appear to depend on the exact time since the behavior began (*Shimaoka et al., 2018*). Thus, the distribution of behavioral state dwell times must be the same across the recording epochs corresponding to all cortical areas in order to detect distributed patterns of correlations between behavior and neural activity. In practice this would seem to require the simultaneous recording of all the individual neurons across the dorso-lateral cortex that are part of the neural ensembles identified by Rastermap, as we have done here - accurately piecing such ensembles together across multiple separate recordings might not be possible.

For this reason, directly combining and/or comparing the correlations between behavior and neural activity across regions imaged in separate sessions may not reveal the true differences in the relationship between behavior and neural activity across cortical areas, due to a 'temporal filtering effect', because the correlations between behavior and neural activity in each region appear to depend on the exact time since the behavior began (*Shimaoka et al., 2018*). In our view, this makes the simultaneous recording of multiple cortical areas essential for proper comparison of the dependence of their neural activity on arousal/movement, because only then are the distributions of behavioral state dwell times the same across the recording epochs corresponding to all cortical areas.

Additional experiments are warranted to uncover the mechanistic basis of such differences across large populations of cortical neurons associated with changes in activity level during periods of fluctuating arousal/engagement. For example, correlations between spiking-related $Ca^{2+}$ activity and movement/arousal under 2p imaging, both as shown here and in previous studies (*Musall et al., 2019*; *Stringer et al., 2019*), and Neuropixels recordings (*Steinmetz et al., 2019*; *Stringer et al., 2019*), appear not to mirror those observed with widefield voltage imaging of membrane potential (*Shimaoka et al., 2018*; especially in VIS_lm and SS_b). This suggests the existence of a complex interaction between movement/arousal related changes in membrane potential and spiking that may vary across cortical areas.

In general, our preparations allow for the observation of diverse pan-cortical neural populations that likely communicate with each other in a transient and/or sustained manner during alternating periods of low and high movement/arousal and drive different patterns of global functional connectivity. Furthermore, the apparent variability of these neurobehavioral alignments suggests that interactions between cognitive and arousal/movement encodings may also occur during spontaneous behavior, in addition to during task performance (*Musall et al., 2019*), and that the degree of this interaction may depend on the cortical area.

## Advantages and disadvantages of our approach (with potential solutions)

A main advantage of our approach is the flexibility afforded by the combination of our two surgical preparations (see Protocols II, III), in terms of the sheer number of cortical areas spanning distant regions that can be recorded simultaneously at single cell resolution under 2p imaging. The interoperability of our preparations across widefield 1p and Thorlabs mesoscope 2p imaging rigs (see Protocol IV) allows for the straight-forward development of an experimental pipeline that first surveys widespread activity at fast timescales (widefield, ~10–50 Hz) and then investigates the activity of large subregions in a serial manner at intermediate timescales (Thorlabs mesoscope, ~1–10 Hz).

A drawback of our current preparations is that they require use of a large water-immersion objective with a relatively short working distance (~2.4–3.0 mm). This limits access for simultaneous electrophysiological recordings during mesoscale imaging, even though the same preparations could be used for simultaneous imaging and electrophysiological recordings on other acquisition rigs (i.e. widefield and/or standard 2p), used in parallel to the mesoscope with the same mice.

In tandem with our headpost, surgical, behavioral, and analytic protocols, this could enable simultaneous Neuropixels, whole-cell electrophysiology, and mesoscale single-cell resolution imaging experiments. Such an approach would allow integration of neural data from multiple brain depths and areas, as well as across multiple spatiotemporal scales. Thus, these techniques would allow modification of our current methods to allow examination of faster timescale cortical dynamics than we can currently capture with 2p imaging alone, such as those related to initial encoding of sensory information and decision making.

Another limitation of our current methodologies for single large FOV imaging, as presented here, is that we imaged neurons at a single cortical depth, and at an imaging rate that did not take full advantage of the speed of the $Ca^{2+}$ indicator that we used (i.e. GCaMP6s; we estimate that imaging at ~10 Hz would sample all available information and avoid aliasing with a rise time of ~200ms and decay time of ~1.2 s; *Chen et al., 2013*). To address these limitations, we are expanding our analyses to sessions, which we have already recorded, acquired at ~5–10 Hz over 4–6 FOVs at multiple z-depths (e.g. V1, A1, M2, RSpl, and SS_m/_n cortical areas; *Figure 1—figure supplement 2e*), with resolutions ranging from 0.2 to 1 μm per pixel (*Figure 1—figure supplement 2d*).

## Future directions

Reliability of cortical area identification and alignment in our dorsal and side mount preparations is enhanced due to standardization of headposts and cranial windows, and the combined use of skull landmark and cortical vasculature alignment in a rigorous, semi-automated widefield multimodal sensory mapping protocol. This has allowed us to acquire an extensive, standardized neurobehavioral data set (~200 hours of spontaneous behavior with 30 Hz high-resolution face and body video from three cameras, ~1,000,000 total neurons at ~3–10 Hz 2p acquisition rates imaged at ~1000 x 1,000 pixels across ~250 sessions, total combined across both preparations and both FOV imaging configurations in 17 mice; see *Figure 1—figure supplement 2e*).

We are currently working on modifying these preparations to allow combined 2p imaging and whole-cell electrophysiological recordings (i.e. using a Thorlabs Bergamo II microscope) and/or widefield imaging and high-density electrophysiological recordings (i.e. Neuropixels; *Jun et al., 2017*; *Peters et al., 2021*). This will allow us to use a single preparation in which each mouse can be used to acquire data across multiple spatiotemporal scales during spontaneous and task-engaged behavioral epochs. Such combined recordings, with either Neuropixels or whole cell recording, may soon be possible with mesoscale 2p imaging, either on the Diesel 2p or the Thorlabs 2p mesoscope (*Sofroniew et al., 2016*), by use of a recently developed air-immersion objective (*Yu et al., 2022*, bioRxiv). A similar mesoscale air-immersion 2p objective is also under development at Thorlabs.

In the future, we hope to expand our analyses of this extensive dataset to include identification of joint Rastermap/BSOiD (or keypoint-MOSEQ motif "syllable"; *Weinreb et al., 2023*; bioRxiv) transitions, and to perform further precision neurobehavioral alignment, using other methodologies such as hierarchical state-space models (Lindermann, S; https://github.com/lindermanlab/ssm; *LindermanLab, 2018*), and factorial HMMs (*Ghahramani and Jordan, 1997*). These methods will allow increased accuracy over multiple behavioral timescales and the ability to predict transition or change points between different behavioral and/or neural activity states.

We will also examine changes in pan-cortical functional connectivity between communities of neurons with similar activity relationships associated with changes in behavioral state with statistical techniques such as Vector Autoregressive Union of Intersections (VA-UoI; *Balasubramanian et al., 2020*; *Ruiz et al., 2020*; *Bouchard and Pabhat Snijders, 2017*) that allow us to rigorously test for and remove putative but 'false positive' connections based on coincidental activity correlations. Together, these analysis tools, when combined with the experimental techniques demonstrated here, will enable both the identification of patterns of neurobehavioral alignment between arousal/movement and neural activity that are strongly predictive of high levels of behavioral performance, and the direct probing control of these patterns with targeted optogenetic manipulations.

## Materials and methods

### Mice

All experiments were approved by the University of Oregon Institutional Animal Care and Use Committee, under protocols 20–12 and 20–14 (IACUC IDs TR202300000012 and TR202300000014). To image cortical activity, we used both male and female, 8–50 week old, CaMKII-tTA x tetO-GCaMP6s (provided by Cris Niell, University of Oregon; JAX # 007004x024742), CaMKII-Cre x Ai148 (GCaMP6f; JAX #005359x030328), and CaMKII-Cre x Ai162 (GCaMP6s; JAX# 005359x031562) mice (Jackson Labs, Allen Institute). Neural activity was typically imageable for up to between 30 and 180 days post-windowing. For experiments with combined $Ca^{2+}$ imaging and optogenetic inhibition we used PV-Cre x Ai32 x Thy1-RGECO mice (JAX # 017320x024109 x 030528 or 030527). Habituation of each mouse to the experimental setup was performed for 2–3 days prior to 2p data acquisition during standard protocol for spontaneous behavior.

### Surgeries

See Protocols II and III, and Supplementary methods and materials.

### Behavioral videography

See Supplementary methods and materials, Overall workflow, Protocol I. See *Videos 1 and 2*.

### Widefield imaging and multimodal mapping

See Supplementary methods and materials, Overall workflow, Protocol I, and Multimodal mapping. See *Videos 3–7*.

### 2-photon imaging and ROI extraction

See Supplementary methods and materials, ScanImage 2p acquisition. 2p imaging was performed with a Spectra Physics Mai Tai HP-244 tunable femtosecond laser and the Thorlabs Multiphoton Mesoscope. We controlled imaging with ScanImage (2018–2021; Vidrio) running in Matlab, with GPU-enabled online z-axis motion correction. Behavioral synchronization was achieved by triggering 2p acquisition with the first right body camera frame clock signal. Left and posterior cameras were triggered by the same signal, but were subject to variable LabView runtime-related delays on the order of up to ~10 ms and, on rare occaision, dropped frames. Exposure times for both left and right body cameras, as well as frame-clock times for all three body cameras, were recorded in Spike2 (CED Power1401) as continuous waveforms and events, respectively, to allow for temporal alignment of recorded videos.

We used the ScanImage function SI render to combine image strips from large field-of-view (FOV) sessions, and in some cases we used custom Suite2p Matlab scripts (courtesy of Carsen Stringer) to align small FOVs from the same or different z-planes in a single session. Rigid and non-rigid motion correction were performed in Suite2p, along with region of interest (ROI) detection and classifier screening of ROIs likely to be neurons. Rigid motion correction computes the shift between a reference image and each frame using phase-correlation, and non-rigid correction divides the image into subsections and calculates the shift of each block, or subsection, separately (*Pachitariu et al., 2016*; also: https://suite2p.readthedocs.io/en/latest/registration.html).

Fluorescence intensities were calculated as changes in fluorescence divided by baseline fluorescence (dF/F), where F was calculated, for each neuron, using an ~30–60 s (100–200 frame) rolling 10th

or 15th percentile baseline of the neuropil subtracted mean ROI pixel intensity (F-0.7*Fneu). Traces of dF/F were then truncated to match the length of acquired behavioral data and resampled (python: scipy.signal.resample for upsampling, or numpy: slicing for downsampling), along with behavioral data, to 10 Hz for further analysis. 2p acquisition rates for the large FOV sessions ranged from roughly 1.6–4.5 Hz, with most sessions acquired at ~3 Hz. ROIs with a Suite2p classifier-assigned cell probability less than 0.5 were excluded from further analysis.

Behavioral data was acquired with a Power1401 in Spike2 (CED) at 5000 Hz and downsampled to 10 Hz without prior filtering, during preprocessing in Python, from real measurement rates of 100 Hz at the encoder for walk speed, and 30 Hz at the face/body cameras for pupil diameter and whisker motion energy.

### Pose-tracking analysis (DeepLabCut)

An example DeepLabCut (DLC) (https://github.com/DeepLabCut/DeepLabCut; *DeepLabCut, 2018*; *Mathis et al., 2018*) annotated video with 66 labeled points is shown in *Video 1* (bottom right). Video snippets were concatenated, cropped, truncated, and aligned to 2p frames with FFMPEG under the Linux subsystem for Windows (Ubuntu 20.04; Windows 10 Enterprise 2004) or Ubuntu 18.04 running on local network GPU clusters. In the example session shown in *Figures 5, 6* and *Figure 6—figure supplement 1*, we analyzed the right Teledyne Dalsa camera video (30 Hz, 1020x760 pixel with 2x2 binning, 1" x 1.8" sensor, 90 min video, ffmpeg crop compression to 525.12 MB), which was natively aligned to the 2p acquisition by a common trigger with frameclock and exposure times recorded by a CED 1401 data acquisition device in Spike2 software (Cambridge Electronic Design), running 650 K iterations (i.e. below the default of 1.03 million, to avoid potential overfitting) with the resnet_50 model in DLC (see labeled video, *Video 2*). Some movies had noticeable sporadic motion blur (in particular of whiskers during whisking), small areas of overexposure, and brief periods of paw occlusion or mislabeled points during walking, grooming, and other brief periods of high motor activity. These difficulties could be remedied in the future by increasing sampling frame rate (although this generates larger file sizes), as well as further optimizing IR illumination. For the preliminary analysis contained here, we performed multiple reruns of DLC to achieve an acceptable quality standard, but we did not rigorously employ tracking, exclusion, or smoothing of DLC outputs.

### Behavioral motif analysis (BSOiD)

Pose-tracking data outputs from DLC in the form of comma-separated value files were loaded into B-SOiD (https://github.com/YttriLab/B-SOID; *Hsu et al., 2021*; *Hsu and Yttri, 2021*) along with concatenated, cropped, and spatially downsampled (FFMPEG) right-camera videos (~20–90 min,~100 MB to 1 GB per video). B-SOiD uniform manifold approximation and projection (UMAP) embedding training was performed on between 1 and 4 sessions from the same mouse. Between 20 and 60 labeling space dimensions were typically assigned per session, with between 75 and 98% of features confidently assigned to between ~2 and 15 motifs per session. Minimum bout time was set at 900 ms, and minimum cumulative motif time per session was set at ~1–5 min.

Pixel-wise analysis of cumulative motion energy showed that the motifs identified by B-SOiD differed in terms of body-region localized basic movement patterns in a manner that was consistent with human-annotated categorization labels. Thresholding of cumulative motion energy by z-score further confirmed this and made the following kinematic patterns evident: (i) In asymmetric pose motifs, a clear region of low movement (dark pixels) is visible in the chest region between the left and right shoulders that is either absent or not as prominent in high-arousal and symmetric pose motifs; (ii) Enhanced nose and mouth movement was associated with high arousal walking but could also occur independently of walking; and (iii) Increased whisker movement was associated with high arousal walking but could also occur independently of walking.

### Protocol I
Overall workflow

1. Headpost implantation in mice between 8 and 12 weeks postnatal (male preferred due to larger skull size). 60 minutes under isoflurane. (90% survival)
2. Recovery for 3–7 days. (90% survival)

3. Multimodal mapping with widefield 1p Ca²⁺ imaging on day 2 or 3 post-surgery. (100% survival)
4. Window implantation surgery. 60-90 minutes under isoflurane. (75% survival)
5. Recovery for 7–10 days. (90% survival)
6. Habituation to head-fixation and wheel (2–3 days). (100% survival)
7. Spontaneous behavior mesoscope imaging sessions (90 min each). At least three with [5000x660 µm] x 7 FOVs at ~3 Hz, and at least three with [660x660 µm] x 4 FOVs at ~10 Hz. Interleaved with passive auditory and passive visual stimulation sessions. Up to 3 sessions per day, for a total of 6–9 days of imaging (2–3 weeks). (100% survival)
8. Mice trained in the behavioral tasks (~3–6 weeks). (67% success)
9. Mice imaged on mesoscope while performing multimodal behavior (3–5 sessions over 1–2 weeks). (90% success)
10. Mice imaged on mesoscope, with targeted optogenetic inhibition of cortical subregions, while performing multimodal behavior (3–5 sessions over 1–2 weeks). (90% success)

Total time (t0=surgery 1): 15 weeks (~3.5 months)
Age range of mouse: 8–27 weeks postnatal
Overall survival rate to stage 10: 0.9 * 0.9 * 0.75 * 0.9=55%
Overall success rate to stage 10: 0.9 * 0.9 * 0.75 * 0.9 * 0.67 * 0.9 * 0.9=30%
Estimated initial sample size of mice needed to get N=6 for spontaneous and passive: 10
Estimated initial sample size of mice needed to get N=6 for behavior: 20

*Note: Perform additional widefield multimodal mapping sessions and vasculature to Allen Common Coordinate Framework (CCF) coalignment and registration approximately once every 4 weeks or as needed.

## Protocol II
### Headpost surgery: 'Side mount'

1. Weigh the mouse and prepare 1 mL ringers, 6.0 mg/kg Meloxicam SR, and 0.5 mg/kg Buprenorphine SR.
2. Anesthetize the mouse with 2–4% isoflurane in the induction chamber.
3. Mount mouse in stereotax and rotate 22.5° to the left. Reduce isoflurane to 1.4% at vaporizer, set oxygen flow rate to ~1.5 L per minute.
4. Apply ophthalmic ointment, adjust mouse position on heating blanket, place temperature probe in rectum, and stabilize temperature at 35.5°C. Re-adjust head and neck position until breathing is regular and not jerky.
5. Inject Meloxicam SR and Buprenorphine SR subcutaneously.
6. Shave dorsal and right temporoparietal surface of head and remove residual hair with Nair and surgical spears. Sterilize area three times with alternating cottonswab-applied betadine solution and 70% ethanol wipes. Be careful to clean residual Nair away from eyes with Ringers.
7. Use sharp-tipped surgical scissors to cut skin along the sagittal line from just behind lambda to the middle of the olfactory bulb. Next, cut along the parasagittal arc exposing 1–2 mm of skull over the left cortex. Make a final parasagittal cut deep over the right side of the skull to just below the auditory cortex, then follow closely around the right eye to join the initial cut over the right anterior snout.
8. Clip and retract skin around the opening with 4–6 thin 'bulldog' clamps. Use #3 forceps tips, cotton swabs, non-woven sponges, and angled broad edge of #23 or #11 scalpel blade to remove periosteum, clean surface of skull, and score expected headpost contact surface.
9. Make a parasagittal cut separating temporal muscle from right parietal ridge, then use blunt back edge of 90 degree angled hook and broad edge of scalpel to tease muscle away from ridge along its entire length. While holding muscle with #3 forceps, use fine straight spring scissors to cut away the entire muscle from posterior to anterior along the dorsal surface of the zygomatic process, passing closely behind the eye while taking care to avoid rupturing ophthalmic artery. Move 2 bulldog clamps from ventral (right) skin flap to residual edge of muscle so that entire flat lower section of temporal bone is exposed down to corner interface with zygomatic process. If a bleed erupts, place ringers-soaked Vetspon onto soft-tissue pocket and leave for at least 30 s, then remove carefully with #3 forceps. Inject subcutaneous Ringers as necessary to supplement for lost blood.

10. Use broad-side of 1.4 mm diameter drill bit to round-off entire length of parietal suture, pick off stray bone shards with #3 forceps, and clean the surface of the skull. If the mouse will be used primarily as widefield preparation, smooth surface additionally with rotary polishing tips.
11. Clean and score contact surface of headpost, then place over surface of skull in intended attachment position. Medial edge of headpost perimeter should be 1–2 mm left of and parallel to midline. Back edge of headpost should be posterior to lambdoid sutures, front edge of headpost should be over olfactory bulbs, the right eye should be close-in and centered in the eye-loop, and ventral (right) edge should be tight and low over and PAST zygomatic process. Carefully note areas where additional skull surface needs to be exposed and cleaned in order to attach headpost. Also note where the zygomatic process crosses the opening of the headpost (important for the upcoming headpost modification step).
12. Use the broad edge of scalpel blade to retract soft tissue and periosteum where needed to make room for headpost attachment, reposition bulldog clamps, and clean and dry skull surface with cotton swabs and non-woven sponges.
13. Place the headpost ventral (right) edge upwards (contact surface facing you) in a small metal crinkle dish. Add UV curable dental cement to the ventral edge, allow gravity to pull it downwards to form a 'curtain', UV cure, and repeat until the edge of dental cement runs along the position of the zygomatic process observed in step 9 (above).
14. Re-check headpost positioning on skull as in step 10 (above), then re-dry skull and headpost contact surfaces. Apply a thin coating of UV curable dental cement (Applicap, 3 M) with the spatula along the entire contact surface perimeter of the headpost and place firmly onto the skull. While applying pressure with one hand, use the other to irradiate dental cement with pulsing ultraviolet (UV) light for ~30 s. Forceps can also be used to secure headpost in position after initial contact is formed. Be careful to keep cement away from eyes.
15. Apply remaining dental cement (or deploy 2nd capsule as needed) around the interior perimeter of the headpost, filling all gaps with the skull. Use the spatula to remove excess cement from the skull, then UV cure as in step 13 (above).
16. Use a 1.4 mm diameter drill-bit to remove dental cement along the ventral edge of preparation until a smooth continuous bridge is formed between the zygomatic process, the dental cement, and the headpost.
17. Apply a thin layer of Zap cyanoacrylate ('super-glue') to the entire surface of the skull and UV cure/let-dry for 15–30 min. If the mouse is to be used primarily for wide-field 1p imaging, apply Norland UV curable glue after 1–3 days. In either case, cover Zap with Kwik-SIL (WPI), making sure to include an extended 'tab' over the arm of the headpost to enable easy removal.
18. Remove bull-dog clamps, dry skin and push upward around the exterior perimeter of the headpost. Attach skin to the headpost with Vetbond super-glue. Inject 1 mL ringers subcutaneously, un-rotate and remove mouse from stereotax. Return the mouse to their home cage in a heated incubator unit for at least 24 hr. Inject 1 mL ringers subcutaneously every 24 hr until postoperative weight stabilizes. Total time: 50-70 min.

## Protocol III
Cranial window surgery: 'Side mount'

1. Surgery to be performed between 3 and 7 days after headpost implantation, after the initial round of widefield multimodal mapping.
2. Inject 10 mg/kg Dexamethasone and 10 mg/kg Baytril subcutaneously on the day before surgery and also on the day of surgery, ~2 hr prior to placing the mouse on the stereotax.
3. Re-weigh the mouse and prepare 0.5 mg/kg Buprenorphine, 17.5 µL of 25% mannitol heated to 30 ° C, and 1 mL ringers.
4. Place mouse in induction box under 2–4% isoflurane until fully anesthetized and then transfer to stereotax with head rotated 22.5 degrees to the left using adjustable support arm (*Figure 1—figure supplement 1b*, bottom) mounted on breadboard clamped to the base of the stereotax. Decrease isoflurane to 1.5%. Use one or two 82 degree countersunk 2/56 x 1 ⁄ 4 '' screws (McMaster-Carr, MS51959-3D) to affix headpost to support the arm so that the skull does not move during drilling. Make sure that tightening screws does not move headpost relative to skull by iterative micropositioning of head and support arm.
5. Apply ophthalmic ointment and insert a rectal temperature probe. Inject Buprenorphine SR and mannitol subcutaneously. Do NOT inject Meloxicam SR, as N-SAIDs are contraindicated

for coadministration with Dexamethasone. Place non-woven sponge under rear paws and remove Kwik-SIL from skull/headpost. Sterilize the skull with betadine and 70% ethanol wipes.

6. Clean 3 custom cranial windows (9 mm radius bend for targeted imaging of ventral-temporal areas and/or imaging of multiple small fields-of-view (FOVs), or 10 mm radius bend for large FOV imaging) by placing them first in a plastic weigh-dish full of 70% ethanol, then rinsing them in ringers and submerging them in ringers in a second plastic weigh-dish.

7. Dry one window by dabbing on sterile surgical drape, then place on skull preparation, centered inside the headpost perimeter. Use a fine point permanent pen to mark the perimeter of the window on the skull. Remove and re-clean the window.

8. Prepare 20–30 mL of ice-cold Ringer's solution. Increase oxygen flow-rate to 1.7 L/min and lower isoflurane to 0.3% to maintain regular breathing and decrease intracranial pressure.

9. Use a 0.7 mm diameter drill bit to mark corner positions of the window based on pen markings, then to lightly trace the window perimeter. Gently expand the traced perimeter as much as necessary to fit the entire window without removing too much dental cement, as this can destabilize the headpost.

10. Switch to a 1.4 mm diameter drill bit and remove the next ~80% of skull thickness by tracing slowly and continuously around the perimeter of the craniotomy. Alternate with a 0.9 mm diameter bit as necessary. Continue until vasculature is clearly visible through the skull.

11. Switch to 0.5 mm diameter drill bit and thin skull around perimeter until skull cap is sparsely connected, then switch to manual tools to complete craniotomy. Use 90 and 45 degree micropoints (Fine Science Tools) in a coordinated manner to separate the skull cap around the entire perimeter. Test for separation by pushing down on the skull cap at each location. When bleeds occur, apply large volumes of ice cold ringers across the surface and wick out with non-woven sponge – only use Vetspon when absolutely necessary to stop large bleeds, as formation of clots below the skull cap will significantly impair local window clarity.

12. Prepare one window for placement by carefully drying with the tip of a KimWipe or Wek-Cel / Sugi-spear and placing in the correct orientation on the surface of the surgical drape. Attach a left-side stereotaxic microinjection arm (Kopf) and position then rotate-out a 3D printed window stabilizer attached to the end of an injection needle so that it can be quickly repositioned as necessary once the window is in-place. Increase $O_2$ flow rate from 1.25 to 2.0 L/min, and decrease isoflurane concentration from 1.5 to 1.0% for the remainder of the surgery. This may help to decrease intracranial pressure and to elevate respiration during the final, critical steps of the window implantation.

13. Remove the skull cap by placing the 90 degree micropoint at the rostral limit and the 45 degree micropoint at the caudal limit just right of the sagittal sinus, then lifting the skull up from the caudal limit until the sagittal sinus is pulled away from the dura. Re-lower the skull to remove tension on the vasculature, then re-lift skull slightly higher than the first lift to tease the skull vasculature off the skull so that it remains on the dura. Repeat these steps until the skull is free of the dura and vasculature. Use the rostral micropoint to prevent the skull from digging into the brain as the caudal edge is raised, then lift together until the skull is cleared.

14. Perfuse surface of brain with ice-cold ringers for ~30 s or until all bleeds have stably ceased, using fresh non-woven sponges to flow and wick liquid off opposite side of preparation. Use Vetspon sparingly and only as necessary while continuously perfusing. If a significant blood clot occludes a large area, attempt to remove it with #5 forceps but be extremely careful not to damage dura.

15. Place window on surface of brain with two pairs of #3 forceps. Position 3D printed window stabilizer near center of window at ~22.5 degree angle to the right and adjust until even pressure is achieved across the entire window-brain interface. Iteratively lower and raise stabilizer by small amounts with stereotax until visual clarity and contrast of blood vessels is maximized and breathing stabilizes. Rinse with ice-cold ringers and then thoroughly dry with fine-rolled KimWipe corners and/or sugi spears.

16. Apply Flow-It ALC around the perimeter of the window then UV cure. Make sure that there are no bubbles or gaps. Preemptively apply UV light in areas where there might be a risk of Flow-It invading the preparation under the window.

17. (optional) Apply Loctite 4305 around the outer edge of Flow-It, and also at the interface between dental cement and skull, being very careful not to contaminate the surface of the window. UV cure once more.

18. Remove window stabilizer slowly to minimize window rebound. Clean the window with ringers, dry, then apply KwikCast to cover the window, leaving a tab for easy removal over the headpost arm.

19. Subcutaneously inject 1 mL ringers, remove mouse from stereotax, and place in heated recovery unit for at least 24 h. Subcutaneously inject Meloxicam SR on Day 1 post-surgery, along with additional ringers as needed for post-operative weight stabilization. Total time: 60-90 min.

## Protocol IV
### Mounting a headpost and window implanted mouse onto Mesoscope or Widefield rigs

1. If mounting a dorsal mount preparation mouse, attach two horizontally aligned or 'orthogonal' support arms to the breadtable on either side of the running wheel (see *Figure 1a* and *Figure 1—figure supplement 1c*, top).
2. If mounting a side mount preparation mouse, attach one 22.5 degree-angled headpost attachment on the left side of the running wheel (see *Figure 1b and c*, bottom).
3. Insert and fully tighten, and then remove countersunk test screws into the support arm to test the threading. If mounting a dorsal mount preparation mouse, affix a 'test' headpost to both support arms to ensure that their 3D alignment is correct, before attempting with the mouse.
4. Place the mouse centered on the running wheel and hold the middle of its tail with your right hand.
5. If mounting a dorsal mount preparation mouse, use your left hand to place the left wing of the headpost into the support arm screw slot (see *Figure 1—figure supplement 1b, c, top, and e*). Hold the wing in the slot with your thumb, and use your right hand to affix the right wing to its support arm with a single screw.
6. Use a second screw to affix the left wing, then tighten both sides fully in step.
7. If mounting a side mount preparation mouse, use your left hand to place the left wing of the headpost in the support arm slot. Then, use your left thumb to hold the tip of the wing in the slot while using your right hand to insert and tighten a screw in the hole proximal to the mouse's head. Then, use your right hand to hold the middle of the mouse's tail and your left hand to insert and tighten a screw in the hole distal to the mouse's head.
8. Fully tighten both screws in step.
9. If mounting on the mesoscope, place a flat 3D-printed light-shield (wok) fitted to the headpost that you are using onto the preparation, and hold in place bilaterally with 3-pronged lab clamps attached to support arms with quick-ties. With UV and IR illumination on, use LabView NI MAX software to align cameras and ensure that the mouse's face and eyes/pupils are fully visible on both the left and right sides.
10. Mix 1:1 black and white 170 FAST CURE Sylgard in a plastic weigh boat. While holding the flood light fiber in your left hand, use the wooden back-end of a cotton swab to carefully drip Sylgard into the interface between the 3D printed plastic (black PLA) light-shield (wok) and the titanium headpost. Make sure that there are no gaps, and that you do not contaminate the cranial window. Wait 5–6 min for curing to complete, then add water to full meniscus height to test for leaks before proceeding with imaging.
11. If mounting on the widefield microscope for 1p imaging, attach the 3D-printed light blocking cone (left half for multi-modal, full circumference for behavioral experiment) by pressing it onto the edge of the headpost and using an inverted 3-pronged lab clamp attached to the imaging lens with velcro to grip and secure the top edge of the cone before proceeding with imaging.

## Protocol V
### Allen CCF, widefield, vasculature, and 2-photon co-alignment and registration (see *Figures 2 and 3*, *Figure 1—figure supplement 2*, Videos 3–7)

1. Acquire co-aligned blood vessel, skull, and multimodal widefield reference images.
2. Create blood vessel and skull images by loading 30 s (1500 frames at 50 Hz) widefield movies into Fiji, auto-adjusting brightness and contrast, and creating standard deviation z-stack projections.
3. Create stimulus triggered mean dF/F (here $F_0$ is equal to the global 10th percentile of fluorescence intensity) images for each 5 min (15000 frames at 50 Hz) sensory stimulation movie by averaging the 1 s baseline-subtracted $Ca^{2+}$ fluorescence response in a 1 s window following

each stimulus onset using custom Matlab code (SensoryMapping_Vickers_Jun2520.m). Adjust stimulus timing signal detection threshold for each modality as needed.

4. For each multimodal mapping session, load skull, blood vessel, and multimodal dF/F images into Fiji and make sure that image size and resolution are the same for all images before proceeding.

5. Create a master overlay image by starting with the blood vessel image. Then proceed through each overlay image with the following substeps: (i) Threshold image so that the saturated area is contiguous in the target area and has an outline that matches its shape in the Allen CCF. (ii) Create a mask. (iii) Create selection. (iv) Use the magic wand tool to select the region of interest. (v) Select master blood vessel image. (vi) Press shift +e to place selection outline as overlay. (vii) Select Image/overlay/flatten and save new image.

6. It may be necessary to perform step 5 (above) for the skull image by manually selecting bregma and lambda with Fiji circle drawing tool.

7. Open custom Matlab code for CCF alignment and check to confirm that it's 'on path' (align_recording_to_allen_Vickers_affine_Jan0120.m, or align_recording_to_allen_Vickers_pwl.m; adapted code from Shreya Saxena and Matt Kaufman, personal communication, 2018).

8. Navigate to the folder containing outputs of steps 5 and 6.

9. If aligning a 'Crystal Skull' preparation or an 'A1/V1/M2' preparation with fewer than 6 alignment points, use code for 'affine' transformation. If aligning an 'A1/V1/M2' preparation with more than or equal to 6 alignment points, use code for 'pwl' (piecewise linear) transformation.

10. Run 'computeAllenDorsalMap.m' to create 'allenDorsalMap.mat', and create a string array called 'alignareas' containing a list of CCF areas whose centers you will designate based on your master overlay image. Each entry will be in the form 'Cortical_hemisphere Area_name'; for example: 'R VISp1', 'R AUDp1', or 'R SSp-bfd1'. Copy these files into your working directory.

11. Load your image by double-clicking its name in the 'Current folder' window.

12. Convert your image to grayscale and auto-adjust brightness and contrast by typing 'im0=mat2gray(imageName)', followed by im0=imadjust(im0).

13. Load 'alignareas', 'areanames', and 'dorsalMapScaled' by double-clicking on 'preprocessed_allenDorsalMap.mat' and 'alignareas' in the 'Current Folder' window.

14. Run the 'align_recording…..m' code by typing "tform = align_recording_to_allen_Vickers_affine(im0,alignareas,true)

15. Select points on the master image as requested.

16. Program will generate output images with CCF overlaid on the master image with reference points shown as red 'x', user-selected points as blue 'o'. For affine transformation, use '….inverse.png' output so see CCF overlay on original coordinate system.

17. Run custom Python code (https://github.com/vickerse1/mesoscope_spontaneous/blob/main/python_code/mesoscope_preprocess_MMM_creation.ipynb; *Vickers, 2024b*) to warp and align mesoscope meanImage with cellMap onto master overlay image with CCF based on user-identified common vasculature intersections.

18. For each Suite2p-identified cell in the cellMap, assign a coded CCF area name (i.e. a dictionary with a unique number and color identifier for each area).

## Acknowledgements

We thank Luca Mazzucato for helpful comments on the manuscript and Paul Steffan, Lawrence Scatena, Julian McAdams, and Daniel Hulsey for technical assistance. We thank Jack Waters for contributing CCF outlines and masks for the side mount rotated view cortical map. We thank Kazi Rafizullah, John Boosinger, and Eowyn Boosinger for helping conceive of, design, and build the mesoscope resonant scanner noise reduction shield. We also thank Elliott Abe, David Wyrick, Shreya Saxena, Matthew Kaufman, Alexander Hsu, Caleb Weinreb, Jens Tillmann, and Carsen Stringer for assistance with coding and setting up preliminary data analysis for both neural and behavioral data, and the teams at Vidrio ScanImage and Thorlabs for technical assistance with software and hardware related to imaging acquisition, respectively. This work was supported by NIH grants R35NS097287 (DAM) and R01NS118461 (DAM).

# Additional information

## Funding

| Funder | Grant reference number | Author |
|---|---|---|
| National Institutes of Health | R35NS097287 | David A McCormick |
| National Institutes of Health | R01NS118461 | David A McCormick |

The funders had no role in study design, data collection and interpretation, or the decision to submit the work for publication.

## Author contributions

Evan D Vickers, Conceptualization, Resources, Data curation, Software, Formal analysis, Validation, Investigation, Visualization, Methodology, Writing - original draft, Project administration, Writing – review and editing; David A McCormick, Conceptualization, Supervision, Funding acquisition, Project administration, Writing – review and editing

## Author ORCIDs

Evan D Vickers http://orcid.org/0000-0002-7053-4740
David A McCormick https://orcid.org/0000-0002-9803-8335

## Ethics

All experiments were approved by the University of Oregon Institutional Animal Care and Use Committee under protocols 20-12 and 20-14 (IACUC IDs TR202300000012 and TR202300000014) and performed in strict accordance with the recommendations in the Guide for the Care and Use of Laboratory Animals of the National Institutes of Health. All surgery was performed under isoflurane anesthesia, and every effort was made to minimize suffering.

Reviewer #1 (Public Review): https://doi.org/10.7554/eLife.94167.3.sa1
Reviewer #2 (Public Review): https://doi.org/10.7554/eLife.94167.3.sa2
Reviewer #3 (Public Review): https://doi.org/10.7554/eLife.94167.3.sa3
Author response https://doi.org/10.7554/eLife.94167.3.sa4

# Additional files

## Supplementary files

• MDAR checklist

## Data availability

Data related to all main and supplementary figures have been deposited on FigShare. All related code, supplementary figures and movies, and design files have been deposited on GitHub (copy archived at *Vickers, 2024a*).

The following dataset was generated:

| Author(s) | Year | Dataset title | Dataset URL | Database and Identifier |
|---|---|---|---|---|
| Vickers ED, McCormick DA | 2024 | Datasets supporting "Pan-cortical 2-photon mesoscopic imaging and neurobehavioral alignment in awake, behaving mice" | https://plus.figshare.com/collections/Datasets_supporting_Pan-cortical_2-photon_mesoscopic_imaging_and_neurobehavioral_alignment_in_awake_behaving_mice_/7052513 | figshare, 10.25452/figshare.plus.c.7052513 |

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

## Appendix 1

## Supplementary methods and materials
Overall workflow, Protocol I

**Appendix 1—table 1.** Approximate costs–single behavior rig.

| Product description | Manufacturer | Quantity | Unit price | Extended price ($USD) |
|---|---|---|---|---|
| 25 Watt power supply | Tucker Davis | 1 | 325 | $325.00 |
| Electrostatic loudspeakers | Tucker Davis | 4 | 195 | 780.00 |
| Electrostatic loudspeaker drivers | Tucker Davis | 2 | 650 | 1,300.00 |
| PC14461 Sound card | National Instruments | 1 | 4,805.00 | 4,805.00 |
| SCB 68 with Connector Cable | National Instruments | 1 | 478.00 | 478.00 |
| PCIe 6321 Multifunction I/O | National instruments | 1 | 690.00 | 690.00 |
| 810 nm narrow bandpass filter | Midwestern Optical Systems | 1 | 141 | 141 |
| Genie Nano M2050 Mono | Teledyne Dalsa | 1 | 878.43 | 878.43 |
| NE-500 Programmable OEM Syringe Pump | New Era Pumps | 7 | 495 | 3,465.00 |
| US OEM Starter kit | New Era Pumps | 7 | 25 | 175 |
| C-Mount 55 mm Telecentric Fixed Focus Lens COTEC55 | computar | 4 | 327.52 | 1,310.08 |
| Assorted Optomechanical Posts and breadboard (approximate) | Thorlabs | varied | N/A | 488.88 |
| LCD Monitor with Blanking Apparatus (Includes Arduino Micro-Controller) (approx.) | Amazon/Arduino | varied | N/A | 120.00 |
| Rotary Encoder | KAMAN AUTOMATION INC | 1 | 217 | 217.00 |
| Custom Cylindrical Treadmill | Public Missiles LTD | 1 | 84 | 84.00 |
| Lick Detection Unit | Custom Build | 1 | 1000 | 1000.00 |
| Computer for Behavior (Intel i5 or better, 32 GB RAM or better) (approx.) | User Preference | 2 | 700 | 1400.00 |
| Assorted Electrical Components, Misc. Hardware, Adapters, etc. | Varied | varied | N/A | 700.00 |
| Power 1401 | CED | 1 | 5697.60 | 5697.60 |
| | | | Total | 24,054.99 |

## Design software

autodesk.com: Inventor and Fusion
emachineshop.com

## 3D printed titanium headposts, shot-peening finish

i.materialise.com/

sculpteo.com/
~$35-60per headpost

## 3D printed light deflectors and accessories

makergear.com
black PLA, 0.35 mm nozzle

## Kopf stereotax

926-B mouse nose/tooth bar assembly
907 mouse anesthesia mask

## Widefield imaging

Redshirt Imaging, DaVinci SciMeasure & Turbo-SM64
PCO.edge 5.5 M-AIR-CLHS-PCO, & CamWare 4.0
Tamron SP 90 mm f/2.8 Di Macro 1:1 VC USD Lens for Nikon F

## Headpost surgery, Protocol II

Mouse electric trimmer combo kit with detailer (CL9990-1201, Kent Scientific)
Nair sensitive hair removal cream (https://www.amazon.com)
PurSwab 3" small pointed ESD foam swabs (https://www.amazon.com)
Vetoquinol Nutri Cal (4.25 oz Paste; https://www.amazon.com)
Puralube Vet Ophthalmic Ointment (https://www.amazon.com)
3 M Vet Bond (https://www.amazon.com)
Scalpel blades - #23 (10023–00; Fine Science Tools)
Kwik-Cast sealant (World Precision Instruments)
Kwik-SIL sealant (World Precision Instruments)
Pacer Technology (Zap) Slo-Zap (Thick) Adhesives, 2 oz (https://www.amazon.com)
Micro-bulldog clamp for mice (INS600119-2, Kent Scientific)
RelyXUniCem Aplicap Refill A1 20/Bx 3 M ESPE Products (036090–3789981; Henry Schein DBA Butler Animal Health)
Disposable aluminum crinkle dishes with tabs, 8 mL (12577–081; VWR)
3 M Aplicap Applier Activator/Applier Set, 37160
HP 1RF-009 Round Stainless Steel Burs Pk/10 [1RF-009-HP (All4Dentist)]
Small homeothermic blanket system with control unit (Q-21090, Harvard Apparatus)
Gelfoam for cessation of bleeding (NC1061303, Fisher Scientific)
Applicator cotton tipped his non-sterile, 3 inch, wood handle (Henry Schein)
Dumont #3 forceps (11293–00; Fine Science Tools)
Angled 80 deg long probe (10140–03; Fine Science Tools)
Fine scissors, tungsten carbide, straight (14568–09; Fine Science Tools)
Vannas spring scissors – 2.5 mm (15000–08; Fine Science Tools)

## Cranial window surgery, Protocol III

https://www.labmaker.org/, "Crystal Skull – One Million Neurons" (Tony Kim, Yanping Zhang and Mark Schnitzer; https://www.labmaker.org/products/crystal-skull?_pos=1&_sid=84c-16cded&_ss=r)
TLC International custom cutting, Phoenix-600, 0.21 mm Schott D263T Glass (9849 North 21 Avenue, Phoenix, AZ; A1/V1/M2)
GlasWerk Inc, custom 9–12 mm radius glass bending (29710 Avenida de las banderas, Rancho Santa Margarita, CA; A1/V1/M2)
Loctite 4305 LT cure ad 1oz bottle (LT303389, Krayden)

Dynarex non-woven sponge, N/S 4Ply (amazon.com)
Osada EXL-M40 brushless micromotor system (EXL-M40; Dorado dental supply)
Flow It ALC Flowable Syringe WO Value Pack 6/Pk (726240, Henry Schein)
Micro-point, angled 90 degree long (10065–15; Fine Science Tools)
Micro-point, angled 45 degree long (10066–15; Fine Science Tools)

### Rig mounting, Protocol IV

Mil Spec St Steel Phillips Flat Head Screws, 82 degree (96877 A18, McMaster-CARR)
Dow Corning Sylgard 170 Fast Cure Silicone Encapsulant Black 210 mL (Ellsworth Adhesives)
Posts (Thorlabs; TR6-P5)
Ball joint (Thorlabs; SL20)
Post holder (Thorlabs; PH2E)
Clamp fork (Thorlabs; CF125)
Platform (Thorlabs, MB4)
Motorized linear stage, 25 mm range, 104 mm/s, 48 V (X-LSM025B-E03-KX14A; zaber.com)

### Multimodal mapping

Eyoyo 15.6" inch Gaming Monitor 1920x1080 HDR Display Second (newegg.com)
E-650 Piezo Amplifier for Multilayer bending actuators, 18 W
PICMA multilayer piezo bending actuator, 2000 μm travel range, 45 mm × 11.00 mm×0.55 mm, stranded wires (PL140.11; physik instrumente.store)
Tucker Davis ES1 Free Field Electrostatic Speaker
Tucker Davis ED1 Electrostatic Speaker Driver
Oben BD-0 mini ball head (amazon.com)
Locking ball and socket mount, ¼"–20 threaded (TRB2; Thorlabs)
Mounted LED 470 nm (760 mW, 1000 mA) (M470L4, Thorlabs)
Dichroic (T495lpxr, chroma.com)
Excitation filter (ET470/40 x; chroma.com)
Emission filter (ET525/50 m; chroma.com)
Kinematic fluorescence filter cube (DFM1, Thorlabs)

### ScanImage 2p acquisition

Spectra Physics Mai Tai HP-244 tunable femtosecond laser
https://www.spectra-physics.com/en/f/mai-tai-ultrafast-laser
Thorlabs Mesoscope
https://www.thorlabs.com/newgrouppage9.cfm?objectgroup_id=10646
ScanImage 2018–2021 (Vidrio Technologies)
https://vidriotechnologies.com/scanimage/
http://scanimage.vidriotechnologies.com/display/SIH/ScanImage+Home

### Computing resources

EVGA GeForce RTX 3080 Ti FTW3 Ultra Gaming, 12G-P5-3967-KR, 12 GB GDDR6X, iCX3 Technology, ARGB LED, Metal Backplate (B0922N253, amazon.com)
Tesla V100 PCIe 32 GB GPUs
GitHub repository: https://github.com/vickerse1/mesoscope_spontaneous; *Vickers, 2024a*.
FigShare +site: https://doi.org/10.25452/figshare.plus.c.7052513

