## [Editor Report · eLife assessment]

This **important** paper presents a thoroughly detailed methodology for mesoscale-imaging of extensive areas of the cortex, either from a top or lateral perspective, in behaving mice. The examples of scientific results to be derived with this method offer promising and stimulating insights. Overall, the method and results presented are **convincing** and will be of interest to neuroscientists focused on cortical processing in rodents and beyond.

---

## [Referee Report · Reviewer #1 (Public Review)]

Summary:

The authors introduce two preparations for observing large-scale cortical activity in mice during behavior. Alongside, they present intriguing preliminary findings utilizing these methods. This paper is poised to be an invaluable resource for researchers engaged in extensive cortical recording in behaving mice.

Strengths:

Comprehensive methodological detailing:

The paper excels in providing an exceptionally detailed description of the methods used. This meticulous documentation includes a step-by-step workflow, complemented by thorough workflow, protocols and list of materials in the supplementary materials.

Minimal of movement artifacts:

A notable strength of this study is the remarkably low movement artifacts, with specific strategies outlined to attain this outcome.

Insightful preliminary data and analysis:

The preliminary data unveiled in the study reveal interesting heterogeneity in the relationships between neural activity and detailed behavioral features, particularly notable in the lateral cortex. This aspect of the findings is intriguing and suggests avenues for further exploration.

Weaknesses:

Clarification about the extent of the method in title:

The title of the paper, using the term "pan-cortical", may inadvertently suggest that both the top and lateral view preparations are utilized in the same set of mice, while the authors employ either the dorsal view (which offers limited access to the lateral ventral regions) or the lateral view (which restricts access to the opposite side of the cortex).

Despite the authors not identifying qualitative effects, tilting the mouse's head could potentially influence behavioral outcomes in certain paradigms.

---

## [Referee Report · Reviewer #2 (Public Review)]

Summary:

The authors present a comprehensive technical overview of the challenging acquisition of large-scale cortical activity, including surgical procedures and custom 3D-printed headbar designs to obtain neural activity from large parts of the dorsal or lateral neocortex. They then describe technical adjustments for stable head fixation, light shielding, and noise insulation in a 2-photon mesoscope and provide a workflow for multisensory mapping and alignment of the obtained large-scale neural data sets in the Allen CCF framework. Lastly, they show different analytical approaches to relate single-cell activity from various cortical areas to spontaneous activity by using visualization and clustering tools, such as Rastermap, PCA-based cell sorting, and B-SOID behavioral motif detection.

The study contains a lot of useful technical information that should be of interest to the field. It tackles a timely problem that an increasing number of labs will be facing as recent technical advances allow the activity measurement of an increasing number of neurons across multiple areas in awake mice. Since the acquisition of cortical data with a large field of view in awake animals poses unique experimental challenges, the provided information could be very helpful to promote standard workflows for data acquisition and analysis and push the field forward.

Strengths:

The proposed methodology is technically sound and the authors provide convincing data to suggest that they successfully solved various challenging problems, such as motion artifacts of large imaging preparations or high-frequency noise emissions, during 2-photon imaging. Overall, the authors achieved their goal of demonstrating a comprehensive approach for imaging neural data across many cortical areas and providing several examples that demonstrate the validity of their methods and recapitulate and further extend some recent findings in the field. A particular focus of the results is to emphasize the need for imaging large population activity across cortical areas to identify cross-area information processing during active behaviors.

Weaknesses:

The manuscript contains a lot of technical details and might be challenging for readers without previous experimental experience. However, the different paragraphs illuminate a large range of technical aspects and challenges of large-scale functional imaging. Therefore, the work should be a valuable source of solutions for a diverse audience.

---

## [Referee Report · Reviewer #3 (Public Review)]

Summary

In their manuscript, Vickers and McCormick have demonstrated the potential of leveraging mesoscale two-photon calcium imaging data to unravel complex behavioural motifs in mice. Particularly commendable is their dedication in providing detailed surgical preparations and corresponding design files, a contribution that will greatly benefit the broader neuroscience community as a whole. The quality of the data is high and examples are available to the community. More importantly, the authors have acquired activity-clustered neural ensembles at an unprecedented spatial scale to further correlate with high level behaviour motifs identified by B-SOiD. Such an advancement marks a significant contribution to the field. While the manuscript is comprehensive and the analytical strategy proposed is promising, some technical aspects warrant further clarification. Overall, the authors have presented an invaluable and innovative approach, effectively laying a solid foundation for future research in correlating large scale neural ensembles with behavioural. The implementation of a custom sound insulator for the scanner is a great idea and should be something implemented by others.

This is a methods paper, but there is no large diagram (in the main figures) that shows how all the parts are connected, communicating and triggering between each other. This is described in the methods and now supplemental figure, but a visual representation would greatly benefit the readers looking to implement something similar as a main figure but I guess they can find it in the methods. No stats for the results shown in Figure 6e, it would be useful to know which of these neural densities for all areas show a clear statistical significance across all the behaviors. While I understand that this is a methods paper, it seems like the authors are aware of the literature surrounding large neuronal recordings during mouse behavior. Indeed, in line 178-179 the authors mention how a significant portion of the variance in neural activity can be attributed to changes in "arousal or self-directed movement even during spontaneous behavior." Why then did the authors not make an attempt at a simple linear model that tries to predict the activity of their many thousands of neurons by employing the multitude of regressors at their disposal (pupil, saccades, stimuli, movements, facial changes, etc). These models are straightforward to implement, and indeed it would benefit this work if the model extracts information on par with what it's known from the literature. We also realize such a model could be done in the future.

---

## [Author Response]

The following is the authors’ response to the original reviews.

**eLife assessment**
This valuable paper presents a thoroughly detailed methodology for mesoscale-imaging of extensive areas of the cortex, either from a top or lateral perspective, in behaving mice. While the examples of scientific results to be derived with this method are in the preliminary stages, they offer promising and stimulating insights. Overall, the method and results presented are convincing and will be of interest to neuroscientists focused on cortical processing in rodents.

Authors’ Response: We thank the reviewers for the helpful and constructive comments. They have helped us plan for significant improvements to our manuscript. Our preliminary response and plans for revision are indicated below.

**Public Reviews:**

**Reviewer #1 (Public Review):**
Summary:The authors introduce two preparations for observing large-scale cortical activity in mice during behavior. Alongside this, they present intriguing preliminary findings utilizing these methods. This paper is poised to be an invaluable resource for researchers engaged in extensive cortical recording in behaving mice.Strengths:-Comprehensive methodological detailing:The paper excels in providing an exceptionally detailed description of the methods used. This meticulous documentation includes a step-by-step workflow, complemented by thorough workflow, protocols, and a list of materials in the supplementary materials.-Minimal movement artifacts:A notable strength of this study is the remarkably low movement artifacts. To further underscore this achievement, a more robust quantification across all subjects, coupled with benchmarking against established tools (such as those from suite2p), would be beneficial.

Authors’ Response: This is a good suggestion. We have records of the fast-z correction applied by the ScanImage on microscope during acquisition, so we have supplied the online fast-z motion correction .csv files for two example sessions on our GitHub page as supplementary files:

https://github.com/vickerse1/mesoscope_spontaneous/tree/main/online_fast_z_correction

These files correspond to Figure S3b (2367_200214_E210_1) and to Figures 5 and 6 (3056_200924_E235_1). These are now also referenced in the main text. See lines ~595, pg 18 and lines ~762, pg 24.

We have also made minor revisions to the main text of the manuscript with clear descriptions of methods that we have found important for the minimization of movement artifacts, such as fully tightening all mounting devices, implanting the cranial window with proper, evenly applied pressure across its entire extent, and mounting the mouse so that it is not too close or far from the surface of the running wheel. See Line ~309, pg 10.

Insightful preliminary data and analysis:The preliminary data unveiled in the study reveal interesting heterogeneity in the relationships between neural activity and detailed behavioral features, particularly notable in the lateral cortex. This aspect of the findings is intriguing and suggests avenues for further exploration.Weaknesses:-Clarification about the extent of the method in the title and text:The title of the paper, using the term "pan-cortical," along with certain phrases in the text, may inadvertently suggest that both the top and lateral view preparations are utilized in the same set of mice. To avoid confusion, it should be explicitly stated that the authors employ either the dorsal view (which offers limited access to the lateral ventral regions) or the lateral view (which restricts access to the opposite side of the cortex). For instance, in line 545, the phrase "lateral cortex with our dorsal and side mount preparations" should be revised to "lateral cortex with our dorsal or side mount preparations" for greater clarity.

Authors’ Response: We have opted to not change the title of the paper, because we feel that adding the qualifier, “in two preparations,” would add unnecessary complexity. In addition, while the dorsal mount preparation allows for imaging of bilateral dorsal cortex, the side mount preparation does indeed allow for imaging of both dorsal and lateral cortex across the right hemisphere (a bit of contralateral dorsal cortex is also imageable), and the design can be easily “flipped” across a mirror-plane to allow for imaging of left dorsal and lateral cortex. Taken together, we do show preparations that allow for pan-cortical 2-photon imaging.

We do agree that imprecise reference to the two preparations can sometimes lead to confusion. Therefore, we made several small revisions to the manuscript, including at ~line 545, to make it clearer that we used two imaging preparations to generate our combined 2-photon mesoscope dataset, and that each of those two preparations had both benefits and limitations.

-Comparison with existing methods:A more detailed contrast between this method and other published techniques would add value to the paper. Specifically, the lateral view appears somewhat narrower than that described in Esmaeili et al., 2021; a discussion of this comparison would be useful.

Authors’ Response: The preparation by Esmaeili et al. 2021 has some similarities to, but also differences from, our preparation. Our preliminary reading is that their through-the-skull field of view is approximately the same as our through-the-skull field of view that exists between our first (headpost implantation) and second (window implantation) surgeries for our side mount preparation, although our preparation appears to include more anterior areas both near to and on the contralateral side of the midline. We have compared these preparations more thoroughly in the revised manuscript. (See lines ~278.)

Furthermore, the number of neurons analyzed seems modest compared to recent papers (50k) - elaborating on this aspect could provide important context for the readers.

Authors’ response: With respect to the “modest” number of neurons analyzed (between 2000 and 8000 neurons per session for our dorsal and side mount preparations with medians near 4500; See Fig. S2e) we would like to point out that factors such as use of dual-plane imaging or multiple imaging planes, different mouse lines, use of different duration recording sessions (see our Fig S2c), use of different imaging speeds and resolutions (see our Fig S2d), use of different Suite2p run-time parameters, and inclusion of areas with blood vessels and different neuron cell densities, may all impact the count of total analyzed neurons per session. We now mention these various factors and have made clear that we were not, for the purposes of this paper, trying to maximize neuron count at the expense of other factors such as imaging speed and total spatial FOV extent.

We refer to these issues now briefly in the main text. (See ~line 93, pg 3).

-Discussion of methodological limitations:The limitations inherent to the method, such as the potential behavioral effects of tilting the mouse's head, are not thoroughly examined. A more comprehensive discussion of these limitations would enhance the paper's balance and depth.

Authors’ Response: Our mice readily adapted to the 22.5 degree head tilt and learned to perform 2-alternative forced choice (2-AFC) auditory and visual tasks in this configuration (Hulsey et al, 2024; Cell Reports). The advantages and limitations of such a rotation of the mouse, and possible ways to alleviate these limitations, as detailed in the following paragraphs, are now discussed more thoroughly in the revised manuscript at ~line 235, pg. 7.

One can look at Supplementary Movie 1 for examples of the relatively similar behavior between the dorsal mount (not rotated) and side mount (rotated) preparations. We do not have behavioral data from mice that were placed in both configurations. Our preliminary comparisons across mice indicates that side and dorsal mount mice show similar behavioral variability. We have added brief additional mention of these considerations on ~lines 235-250, pg 7.

It was in general important to make sure that the distance between the wheel and all four limbs was similar for both preparations. In particular, careful attention must be paid to the positioning of the front limbs in the side mount mice so that they are not too high off the wheel. This can be accomplished by a slight forward angling of the left support arm for side mount mice.

Although it is possible to image the side mount preparation in the same optical configuration that we do without rotating the mouse, by rotating the objective 20 degrees to the right of vertical, we found that the last 2-3 degrees of missing rotation (our preparation is rotated 22.5 degrees left, which is more than the full available 20 degrees rotation of the Thorlabs mesoscope objective), along with several other factors, made this undesirable. First, it was very difficult to image auditory areas without the additional flexibility to rotate the objective more laterally. Second, it was difficult or impossible to attach the horizontal light shield and to establish a water meniscus with the objective fully rotated. One could use ultrasound gel instead (which we found to be, to some degree, optically inferior to water), but without the horizontal light shield, light from the UV and IR LEDs can reach the PMTs via the objective and contaminate the image or cause tripping of the PMT. Third, imaging the right pupil and face of the mouse is difficult under these conditions because the camera would need the same optical access angle as the 2-photon objective, or would need to be moved downward toward the air table and rotated up at an angle of 20 degrees, in which case its view would be blocked by the running wheel and other objects mounted on the air table.

-Preliminary nature of results:The results are at a preliminary stage; for example, the B-soid analysis is based on a single mouse, and the validation data are derived from the training data set.

Authors’ Response: In this methods paper, we have chosen to supply proof of principle examples, without a complete analysis of animal-to-animal variance.

The B-SOiD analysis that we show in Figure 6 is based on a model trained on 80% of the data from four sessions taken from the same mouse, and then tested on all of a single session from that mouse. Initial attempts to train across sessions from different mice were unsuccessful, probably due to differences in behavioral repertoires across mice. However, we have performed extensive tests with B-SOiD and are confident that these sorts of results are reproducible across mice, although we are not prepared to publish these results at this time.

We now clarify these points in the main text at ~line 865, pg 27.

An additional comparison of the results of B-SOiD trained on different numbers of sessions to that of keypoint-MOSEQ (Weinreb et al, 2023, bioRxiv) trained on ~20 sessions can now be found as supplementary material on our GitHub site:

https://github.com/vickerse1/mesoscope_spontaneous/blob/main/Figure_SZZ_BSOID_MOSEQ_align.pdf

The discrepancy between the maps in Figures 5e and 6e might indicate that a significant portion of the map represents noise. An analysis of variability across mice and a method to assign significance to these maps would be beneficial.

Authors’ Response: After re-examination of the original analysis output files, we have indeed discovered that some of the Rastermap neuron density maps in Figure 6e were incorrectly aligned with their respective qualitative behaviors due to a discrepancy in file numbering between the images in 6e and the ensembles identified in 6c (each time that Rastermap is run on the same data, at least with the older version available at the time of creation of these figures, the order of the ensembles on the y-axis changes and thus the numbering of the ensembles would change even though the neuron identities within each group stayed the same for a given set of parameters).

This unfortunate panel alignment / graphical display error present in the original reviewed preprint has been fixed in the current, updated figure (i.e. twitch corresponds to Rastermap groups 2 and 3, whisk to group 6, walk to groups 5 and 4, and oscillate to groups 0 and 1), and in the main text at ~line 925, pg 29. We have also changed the figure legend, which also contained accurate but misaligned information, for Figure 6e to reflect this correction.

One can now see that, because the data from both figures is from the same session in the same mouse, as you correctly point out, Fig 5d left (walk and whisk) corresponds roughly to Fig 6e group R7, “walk”, and that Fig 5d right (whisk) corresponds roughly to Fig 6e group R4, “twitch”.

We have double-checked the identity of other CCF map displays of Rastermap neuron density and of mean correlations between neural activity and behavioral primitives in all other figures, and we found no other such alignment or mis-labeling errors.

We have also added a caveat in the main text at ~lines 925-940, pg. 30, pointing out the preliminary nature of these findings, which are shown here as an example of the viability of the methods. Analysis of the variability of Rastermap alignments across sessions is beyond the scope of the current paper, although it is an issue that we hope to address in upcoming analysis papers.

-Analysis details:More comprehensive details on the analysis would be beneficial for replicability and deeper understanding. For instance, the statement "Rigid and non-rigid motion correction were performed in Suite2p" could be expanded with a brief explanation of the underlying principles, such as phase correlation, to provide readers with a better grasp of the methodologies employed.

Authors’ Response: We added a brief explanation of Suite2p motion correction at ~line 136, pg 4. We have also added additional details concerning CCF / MMM alignment and other analysis issues. In general we cite other papers where possible to avoid repeating details of analysis methods that are already published.

**Reviewer #2 (Public Review):**
Summary:The authors present a comprehensive technical overview of the challenging acquisition of large-scale cortical activity, including surgical procedures and custom 3D-printed headbar designs to obtain neural activity from large parts of the dorsal or lateral neocortex. They then describe technical adjustments for stable head fixation, light shielding, and noise insulation in a 2-photon mesoscope and provide a workflow for multisensory mapping and alignment of the obtained large-scale neural data sets in the Allen CCF framework. Lastly, they show different analytical approaches to relate single-cell activity from various cortical areas to spontaneous activity by using visualization and clustering tools, such as Rastermap, PCA-based cell sorting, and B-SOID behavioral motif detection.

Authors’ Response: Thank you for this excellent summary of the scope of our paper.

The study contains a lot of useful technical information that should be of interest to the field. It tackles a timely problem that an increasing number of labs will be facing as recent technical advances allow the activity measurement of an increasing number of neurons across multiple areas in awake mice. Since the acquisition of cortical data with a large field of view in awake animals poses unique experimental challenges, the provided information could be very helpful to promote standard workflows for data acquisition and analysis and push the field forward.

Authors’ Response: We very much support the idea that our work here will contribute to the development of standard workflows across the field including those for multiple approaches to large-scale neural recordings.

Strengths:The proposed methodology is technically sound and the authors provide convincing data to suggest that they successfully solved various problems, such as motion artifacts or high-frequency noise emissions, during 2-photon imaging. Overall, the authors achieved their goal of demonstrating a comprehensive approach for the imaging of neural data across many cortical areas and providing several examples that demonstrate the validity of their methods and recapitulate and further extend some recent findings in the field.Weaknesses:Most of the descriptions are quite focused on a specific acquisition system, the Thorlabs Mesoscope, and the manuscript is in part highly technical making it harder to understand the motivation and reasoning behind some of the proposed implementations. A revised version would benefit from a more general description of common problems and the thought process behind the proposed solutions to broaden the impact of the work and make it more accessible for labs that do not have access to a Thorlabs mesoscope. A better introduction of some of the specific issues would also promote the development of other solutions in labs that are just starting to use similar tools.

Authors’ Response: We have edited the motivations behind the study to clarify the general problems that are being addressed. However, as the 2-photon imaging component of these experiments were performed on a Thorlabs mesoscope, the imaging details necessarily deal specifically with this system.

We briefly compare the methods and results from our Thorlabs system to that of Diesel-2p, another comparable system, based on what we have been able to glean from the literature on its strengths and weaknesses. See ~lines 206-213, pg 6.

**Reviewer #3 (Public Review):**
SummaryIn their manuscript, Vickers and McCormick have demonstrated the potential of leveraging mesoscale two-photon calcium imaging data to unravel complex behavioural motifs in mice. Particularly commendable is their dedication to providing detailed surgical preparations and corresponding design files, a contribution that will greatly benefit the broader neuroscience community as a whole. The quality of the data is high, but it is not clear whether this is available to the community, some datasets should be deposited. More importantly, the authors have acquired activity-clustered neural ensembles at an unprecedented spatial scale to further correlate with high-level behaviour motifs identified by B-SOiD. Such an advancement marks a significant contribution to the field. While the manuscript is comprehensive and the analytical strategy proposed is promising, some technical aspects warrant further clarification. Overall, the authors have presented an invaluable and innovative approach, effectively laying a solid foundation for future research in correlating large-scale neural ensembles with behaviour. The implementation of a custom sound insulator for the scanner is a great idea and should be something implemented by others.

Authors’ Response: Thank you for the kind words.

We have made ~500 GB of raw data and preliminary analysis files publicly available on FigShare+ for the example sessions shown in Figures 2, 3, 4, 5, 6, S3, and S6. We ask to be cited and given due credit for any fair use of this data.

The data is located here: https://doi.org/10.25452/figshare.plus.c.7052513

We intend to release a complete data set to the public as a Dandiset on the DANDI archive in conjunction with in-depth analysis papers that are currently in preparation.

This is a methods paper, but there is no large diagram that shows how all the parts are connected, communicating, and triggering each other. This is described in the methods, but a visual representation would greatly benefit the readers looking to implement something similar.

Authors’ Response: This is an excellent suggestion. We have included a workflow diagram in the revised manuscript, in the form of a 3-part figure, for the methods (a), data collection (b and c), and analysis (d). This supplementary figure is now located on the GitHub page at the following link:

https://github.com/vickerse1/mesoscope_spontaneous/blob/main/pancortical_workflow_diagrams.pdf

We now reference this figure on ~lines 190-192, pg 6 of the main text, near the beginning of the Results section.

The authors should cite sources for the claims stated in lines 449-453 and cite the claim of the mouse's hearing threshold mentioned in lines 463.

Authors’ Response: For the claim stated in lines 449-453:

“The unattenuated or native high-frequency background noise generated by the resonant scanner causes stress to both mice and experimenters, and can prevent mice from achieving maximum performance in auditory mapping, spontaneous activity sessions, auditory stimulus detection, and auditory discrimination sessions/tasks”

,we can provide the following references: (i) for mice: Sadananda et al, 2008 (“Playback of 22-kHz and 50-kHz ultrasonic vocalizations induces differential c-fos expression in rat brain”, Neuroscience Letters, Vol 435, Issue 1, p 17-23), and (ii) for humans: Fletcher et al, 2018 (“Effects of very high-frequency sound and ultrasound on humans. Part I: Adverse symptoms after exposure to audible very-high frequency sound”, J Acoust Soc A, 144, 2511-2520). We will include these references in the revised paper.

For the claim stated on line 463:

“i.e. below the mouse hearing threshold at 12.5 kHz of roughly 15 dB”

,we can provide the following reference: Zheng et al, 1999 (“Assessment of hearing in 80 inbred strains of mice by ABR threshold analyses”, Vol 130, Issues 1-2, p 94-107).

We have included these two new references in the new, revised version of our paper. Thank you for identifying these citation omissions.

No stats for the results shown in Figure 6e, it would be useful to know which of these neural densities for all areas show a clear statistical significance across all the behaviors.

Authors’ Response: It would be useful if we could provide a statistic similar to what we provide for Fig. S6c and f, in which for each CCF area we compare the observed mean correlation values to a null of 0, or, in this case, the population densities of each Rastermap group within each CCF area to a null value equal to the total number of CCF areas divided by the total number of recorded neurons for that group (i.e. a Rastermap group with 500 neurons evenly distributed across ~30 CCF areas would contain ~17 neurons, or ~3.3% density, per CCF area.) Our current figure legend states the maximums of the scale bar look-up values (reds) for each group, which range from ~8% to 32%.

However, because the data in panel 6e are from a single session and are being provided as an example of our methods and not for the purpose of claiming a specific result at this point, we choose not to report statistics. It is worth pointing out, perhaps, that Rastermap group densities for a given CCF area close to 3.3% are likely not different from chance, and those closer to ~40%, which is our highest density (for area M2 in Rastermap group 7, which corresponds to the qualitative behavior “walk”), are most likely not due to chance. Without analysis of multiple sessions from the same mouse we believe that making a clear statement of significance for this likelihood would be premature.

We now clarify this decision and related considerations in the main text at ~line 920, pg 29.

While I understand that this is a methods paper, it seems like the authors are aware of the literature surrounding large neuronal recordings during mouse behavior. Indeed, in lines 178-179, the authors mention how a significant portion of the variance in neural activity can be attributed to changes in "arousal or self-directed movement even during spontaneous behavior." Why then did the authors not make an attempt at a simple linear model that tries to predict the activity of their many thousands of neurons by employing the multitude of regressors at their disposal (pupil, saccades, stimuli, movements, facial changes, etc). These models are straightforward to implement, and indeed it would benefit this work if the model extracts information on par with what is known from the literature.

Authors’ Response: This is an excellent suggestion, but beyond the scope of the current methods paper. We are following up with an in depth analysis of neural activity and corresponding behavior across the cortex during spontaneous and trained behaviors, but this analysis goes well beyond the scope of the present manuscript.

Here, we prefer to present examples of the types of results that can be expected to be obtained using our methods, and how these results compare with those obtained by others in the field.

Specific strengths and weaknesses with areas to improve:The paper should include an overall cartoon diagram that indicates how the various modules are linked together for the sampling of both behaviour and mesoscale GCAMP. This is a methods paper, but there is no large diagram that shows how all the parts are connected, communicating, and triggering each other.

Authors’ Response: This is an excellent suggestion. We have included a workflow diagram in the revised manuscript, in the form of a 3-part figure, for the methods (a), data collection (b and c), and analysis (c). This supplementary figure is now located on the GitHub page at the following link:

https://github.com/vickerse1/mesoscope_spontaneous/blob/main/pancortical_workflow_diagrams.pdf

The paper contains many important results regarding correlations between behaviour and activity motifs on both the cellular and regional scales. There is a lot of data and it is difficult to draw out new concepts. It might be useful for readers to have an overall figure discussing various results and how they are linked to pupil movement and brain activity. A simple linear model that tries to predict the activity of their many thousands of neurons by employing the multitude of regressors at their disposal (pupil, saccades, stimuli, movements, facial changes, etc) may help in this regard.

Authors’ Response: This is an excellent suggestion, but beyond the scope of the present methods paper. Such an analysis is a significant undertaking with such large and heterogeneous datasets, and we provide proof-of-principle data here so that the reader can understand the type of data that one can expect to obtain using our methods. We will provide a more complete analysis of data obtained using our methodology in the near future in another manuscript.

Previously, widefield imaging methods have been employed to describe regional activity motifs that correlate with known intracortical projections. Within the authors' data it would be interesting to perhaps describe how these two different methods are interrelated -they do collect both datasets. Surprisingly, such macroscale patterns are not immediately obvious from the authors' data. Some of this may be related to the scaling of correlation patterns or other factors. Perhaps there still isn't enough data to readily see these and it is too sparse.

Authors’ Response: Unfortunately, we are unable to directly compare 1-photon widefield GCaMP6s activity with mesoscope 2-photon GCaMP6s activity. During widefield data acquisition, animals were stimulated with visual, auditory, or somatosensory stimuli (i.e. “passive sensory stimulation”), while 2-photon mesoscope data collection occurred during spontaneous changes in behavioral state, without sensory stimulation. The suggested comparison is, indeed, an interesting project for the future.

In lines 71-71, the authors described some disadvantages of one-photon widefield imaging including the inability to achieve single-cell resolution. However, this is not true. In recent years, the combination of better surgical preparations, camera sensors, and genetically encoded calcium indicators has enabled the acquisition of single-cell data even using one-photon widefield imaging methods. These methods include miniscopes (Cai et al., 2016), multi-camera arrays (Hope et al., 2023), and spinning disks (Xie et al., 2023).Cai, Denise J., et al. "A shared neural ensemble links distinct contextual memories encoded close in time." Nature 534.7605 (2016): 115-118.Hope, James, et al. "Brain-wide neural recordings in mice navigating physical spaces enabled by a cranial exoskeleton." bioRxiv (2023).Xie, Hao, et al. "Multifocal fluorescence video-rate imaging of centimetre-wide arbitrarily shaped brain surfaces at micrometric resolution." Nature Biomedical Engineering (2023): 1-14.

Authors’ Response: We have corrected these statements and incorporated these and other relevant references. There are advantages and disadvantages to each chosen technique, such as ease of use, field of view, accuracy, and speed. We will reference the papers you mention without an extensive literature review, but we would like to emphasize the following points:

Even the best one-photon imaging techniques typically have ~10-20 micrometer resolution in xy (we image at 5 micrometer resolution for our large FOV configuration, but the xy point-spread function for the Thorlabs mesoscope is 0.61 x 0.61 micrometers in xy with 970 nm excitation) and undefined z-resolution (4.25 micrometers for Thorlabs mesoscope). A coarser resolution increases the likelihood that activity related fluorescence from neighboring cells may contaminate the fluorescence observed from imaged neurons. Reducing the FOV and using sparse expression of the indicator lessens this overlap problem.

We do appreciate these recent advances, however, particularly for use in cases where more rapid imaging is desired over a large field of view (CCD acquisition can be much faster than that of standard 2-photon galvo-galvo or even galvo-resonant scanning, as the Thorlabs mesoscope uses). This being said, there are few currently available genetically encoded Ca2+ sensors that are able to measure fluctuations faster than ~10 Hz, which is a speed achievable on the Thorlabs 2-photon mesoscope with our techniques using the “small, multiple FOV” method (Fig. S2d, e).

We have further clarified our discussion of these issues in the main text at ~lines 76-80, pg 2.

The authors' claim of achieving optical clarity for up to 150 days post-surgery with their modified crystal skull approach is significantly longer than the 8 weeks (approximately 56 days) reported in the original study by Kim et al. (2016). Since surgical preparations are an integral part of the manuscript, it may be helpful to provide more details to address the feasibility and reliability of the preparation in chronic studies. A series of images documenting the progression optical quality of the window would offer valuable insight.

Authors’ Response: As you suggest, we now include brief supplementary material demonstrating the changes in the window preparation that we observed over the prolonged time periods of our study, for both the dorsal and side mount preparations. The following link to this material is now referenced at ~line 287, pg 9, and at the end of Fig S1:

https://github.com/vickerse1/mesoscope_spontaneous/blob/main/window_preparation_stability.pdf

We have also included brief additional details in the main text that we found were useful for facilitating long term use of these preparations. These are located at ~line 287-290, pg 9.

**Recommendations for the authors:**

**Reviewer #1 (Recommendations For The Authors):**
(1) Sharing raw data and code:I strongly encourage sharing some of the raw data from your experiments and all the code used for data analysis (e.g. in a github repository). This would help the reader evaluate data quality, and reproduce your results.

Authors’ Response: We have made ~500 GB of raw data and preliminary analysis files publicly available on FigShare+ for the example sessions shown in Figures 2, 3, 4, 5, 6, S3, and S6. We ask to be cited and given due credit for any fair use of this data.

We intend to release a complete data set to the public as a Dandiset on the DANDI archive in conjunction with second and third in-depth analysis papers that are currently in preparation.

The data is located here: https://doi.org/10.25452/figshare.plus.c.7052513

We intend to release a complete data set to the public as a Dandiset on the DANDI archive in conjunction with second and third in-depth analysis papers that are currently in preparation.

Our existing GitHub repository, already referenced in the paper, is located here:

https://github.com/vickerse1/mesoscope_spontaneous

We have added an additional reference in the main text to the existence of these publicly available resources, including the appropriate links, located at ~lines 190-200, pg 6.

(2) Use of proprietary software:The reliance on proprietary tools like LabView and Matlab could be a limitation for some researchers, given the associated costs and accessibility issues. If possible, consider incorporating or suggesting alternatives that are open-source, to make your methodology more accessible to a broader range of researchers, including those with limited resources.

Authors’ Response: We are reluctant to recommend open source software that we have not thoroughly tested ourselves. However, we will mention, when appropriate, possible options for the reader to consider.

Although LabView is proprietary and can be difficult to code, it is particularly useful when used in combination with National Instruments hardware. ScanImage in use with the Thorlabs mesoscope uses National Instruments hardware, and it is convenient to maintain hardware standards across the integrated rig/experimental system. Labview is also useful because it comes with a huge library of device drivers that makes addition of new hardware from basically any source very convenient.

That being said, there are open source alternatives that could conceivably be used to replace parts of our system. One example is AutoPilot (author: Jonny Saunders), for control of behavioral data acquisition: https://open-neuroscience.com/post/autopilot/.

We are not aware of an alternative to Matlab for control of ScanImage, which is the supported control software for the ThorLabs 2-photon mesoscope.

Most of our processing and analysis code (see GitHub page: https://github.com/vickerse1/mesoscope_spontaneous) is in Python, but some of the code that we currently use remains in Matlab form. Certainly, this could be re-written as Python code. However, we feel like this is outside the scope of the current paper. We have provided commenting to all code in an attempt to aid users in translating it to other languages, if they so desire.

(3) Quantifying the effect of tilted head:To address the potential impact of tilting the mouse's head on your findings, a quantitative analysis of any systematic differences in the behavior (e.g. Bsoid motifs) could be illuminating.

Authors’ Response: We have performed DeepLabCut analysis of all sessions from both preparations, across several iterations with different parameters, to extract pose estimates, and we have also performed BSOiD of these sessions. We did not find any obvious qualitative differences in the number of behavioral motifs identified, the dwell times of these motifs, and similar issues, relating to the issue of tilting of the mouse’s head in the side mount preparation. We also did not find any obvious differences in the relative frequencies of high level qualitative behaviors, such as the ones referred to in Fig. 6, between the two preparations.

Our mice readily adapted to the 22.5 degree head tilt and learned to perform 2-alternative forced choice (2-AFC) auditory and visual tasks in this configuration (Hulsey et al, 2024; Cell Reports). The advantages and limitations of such a rotation of the mouse, and possible ways to alleviate these limitations, as detailed in the following paragraphs, are now discussed more thoroughly in the revised manuscript. (See ~line 235, pg. 7)

One can look at Supplementary Movie 1 for examples of the relatively similar behavior between the dorsal mount (not rotated) and side mount (rotated) preparations. We do not have behavioral data from mice that were placed in both configurations. Our preliminary comparisons across mice indicates that side and dorsal mount mice show similar behavioral variability. We have added brief additional mention of these considerations on ~lines 235-250, pg 7.

It was in general important to make sure that the distance between the wheel and all four limbs was similar for both preparations. In particular, careful attention must be paid to the positioning of the front limbs in the side mount mice so that they are not too high off the wheel. This can be accomplished by a slight forward angling of the left support arm for side mount mice.

Although it would in principle be nearly possible to image the side mount preparation in the same optical configuration that we do without rotating the mouse, by rotating the objective 20 degrees to the right of vertical, we found that the last 2-3 degrees of missing rotation (our preparation is rotated 22.5 degrees left, which is more than the full available 20 degrees rotation of the Thorlabs mesoscope objective), along with several other factors, made this undesirable. First, it was very difficult to image auditory areas without the additional flexibility to rotate the objective more laterally. Second, it was difficult or impossible to attach the horizontal light shield and to establish a water meniscus with the objective fully rotated. One could use gel instead (which we found to be optically inferior to water), but without the horizontal light shield, the UV and IR LEDs can reach the PMTs via the objective and contaminate the image or cause tripping of the PMT. Third, imaging the right pupil and face of the mouse is difficult to impossible under these conditions because the camera would need the same optical access angle as the objective, or would need to be moved down toward the air table and rotated up 20 degrees, in which case its view would be blocked by the running wheel and other objects mounted on the air table.

(4) Clarification in the discussion section:The paragraph titled "Advantages and disadvantages of our approach" seems to diverge into discussing future directions, rather than focusing on the intended topic. I suggest revisiting this section to ensure that it accurately reflects the strengths and limitations of your approach.

Authors’ Response: We agree with the reviewer that this section included several potential next steps or solutions for each advantage and disadvantage, which the reviewer refers to as “future directions” and are thus arguably beyond the scope of this section. Therefore we have retitled this section as, “Advantages and disadvantages of our approach (with potential solutions):”.

Although we believe this to be a logical organization, and we already include a section focused purely on future directions in the Discussion section, we have refocused each paragraph of the advantages/disadvantages subsection to concentrate on the advantages and disadvantages per se. In addition, we have made minor changes to the “future directions” section to make it more succinct and practical. These changes can be found at lines ~1016-1077, pg 33-34.

**Reviewer #2 (Recommendations For The Authors):**
Below are some more detailed points that will hopefully help to further improve the quality and scope of the manuscript.While it is certainly favorable for many questions to measure large-scale activity from many brain regions, the introduction appears to suggest that this is a prerequisite to understanding multimodal decision-making. This is based on the argument that combining multiple recordings with movement indicators will 'necessarily obscure the true spatial correlation structures'. However, I don't understand why this is the case or what is meant by 'true spatial correlation structures'. Aren't there many earlier studies that provided important insights from individual cortical areas? It would be helpful to improve the writing to make this argument clearer.

Authors’ Response: The reviewer makes an excellent point and we have re-worded the manuscript appropriately, to reflect the following clarifications. These changes can be found at ~lines 58-71, pg. 2.

We believe you are referring to the following passage from the introduction:

“Furthermore, the arousal dependence of membrane potential across cortical areas has been shown to be diverse and predictable by a temporally filtered readout of pupil diameter and walking speed (Shimoaka et al, 2018). This makes simultaneous recording of multiple cortical areas essential for comparison of the dependence of their neural activity on arousal/movement, because combining multiple recording sessions with pupil dilations and walking bouts of different durations will necessarily obscure the true spatial correlation structures.”

Here, we do not mean to imply that earlier studies of individual cortical areas are of no value. This argument is provided as an example, of which there are others, of the idea that, for sequences or distributed encoding schemes that simultaneously span many cortical areas that are too far apart to be simultaneously imaged under conventional 2-photon imaging, or are too sparse to be discovered with 1-photon widefield imaging, there are some advantages of our new methods over conventional imaging methods that will allow for truly novel scientific analyses and insights.

The general idea of the present example, based on the findings of Shimoaka et al, 2018, is that it is not possible to directly combine and/or compare the correlations between behavior and neural activity across regions that were imaged in separate sessions, because the correlations between behavior and neural activity in each region appear to depend on the exact time since the behavior began (Shimoaka et al, 2018), in a manner that differs across regions. So, for example, if one were to record from visual cortex in one session with mostly brief walk bouts, and then from somatosensory cortex in a second session with mostly long walk bouts, any inferred difference between the encoding of walk speed in neural activity between the two areas would run the risk of being contaminated by the “temporal filtering” effect shown in Shimoaka et al, 2018. However, this would not be the case in our recordings, because the distribution of behavior durations corresponding to our recorded neural activity across areas will be exactly the same, because they were recorded simultaneously.

The text describes different timescales of neural activity but is an imaging rate of 3 Hz fast enough to be seen as operating at the temporal dynamics of the behavior? It appears to me that the sampling rate will impose a hard limit on the speed of correlations that can be observed across regions. While this might be appropriate for relatively slow behaviors and spontaneous fluctuations in arousal, sensory processing and decision formation likely operate on faster time scales below 100ms which would even be problematic at 10 Hz which is proposed as the ideal imaging speed in the manuscript.

Authors’ Response: Imaging rate is always a concern and the limitations of this have been discussed in other manuscripts. We will remind the reader of these limitations, which must always be kept in mind when interpreting fluorescence based neural activity data.

Previous studies imaging on a comparable yet more limited spatial scale (Stringer et al, 2019) used an imaging speed of ~1 Hz. With this in view, our work represents an advance both in spatial extent of imaged cortex and in imaging speed. Specifically, we believe that ~1 Hz imaging may be sufficient to capture flip/flop type transitions between low and high arousal states that persist in general for seconds to tens of seconds, and that ~3-5 Hz imaging likely provides additional information about encoding of spontaneous movements and behavioral syllables/motifs.

Indeed, even 10 Hz imaging would not be fast enough to capture the detailed dynamics of sensory processing and decision formation, although these speeds are likely sufficient to capture “stable” encodings of sensory representations and decisions that must be maintained during a task, for example with delayed match-to-sample tasks.

In general we are further developing our preparations to allow us to perform simultaneous widefield imaging and Neuropixels recordings, and to perform simultaneous 1.2 x 1.2 mm 2-photon imaging and visually guided patch clamp recordings.

Both of these techniques will allow us to combine information across both the slow and fast timescales that you refer to in your question.

We have clarified these points in the Introduction and Discussion sections, at ~lines ~93-105, pg 3, and ~lines 979-983, pg 31 and ~lines 1039-1045, pg 33, respectively.

The dorsal mount is very close to the crystal skull paper and it was ultimately not clear to me if there are still important differences aside from the headbar design that a reader should be aware of. If they exist, it would be helpful to make these distinctions a bit clearer. Also, the sea shell implants from Ghanbari et al in 2019 would be an important additional reference here.

Authors’ Response: We have added brief references to these issues in our revised manuscript at ~lines 89-97, pg 3:

Although our dorsal mount preparation is based on the “crystal skull paper” (Kim et al, 2016), which we reference, the addition of a novel 3-D printable titanium headpost, support arms, light shields, and modifications to the surgical protocols and CCF alignment represent significant advances that made this preparation useable for pan-cortical imaging using the Thorlabs mesoscope. In fact, we were in direct communication with Cris Niell, a UO professor and co-author on the original Kim et al, 2016 paper, during the initial development of our preparation, and he and members of his lab consulted with us in an ongoing manner to learn from our successful headpost and other hardware developments. Furthermore, all of our innovations for data acquisition, imaging, and analysis apply equally to both our dorsal mount and side mount preparations.

Thank you for mentioning the Ghanbari et al, 2019 paper on the transparent polymer skull method, “See Shells.” We were in fact not aware of this study. However, it should be noted that their preparation seems to, like the crystal skull preparation and our dorsal mount preparation, be limited to bilateral dorsal cortex and not to include, as does our cranial window side mount preparation and the through-the-skull widefield preparation of Esmaeili et al, 2021, a fuller range of lateral cortical areas, including primary auditory cortex.

When using the lateral mount, rotating the objective, rather than the animal, appears to be preferable to reduce the stress on the animal. I also worry that the rather severe head tilt could be an issue when training animals in more complex behaviors and would introduce an asymmetry between the hemispheres due to the tilted body position. Is there a strong reason why the authors used water instead of an imaging gel to resolve the issue with the meniscus?

Authors’ Response: Our mice readily adapted to the 22.5 degree head tilt and learned to perform 2-alternative forced choice (2-AFC) auditory and visual tasks in this situation (Hulsey et al, 2024; Cell Reports). The advantages and limitations of such a rotation of the mouse, and possible ways to alleviate these limitations, as detailed in the following paragraphs, are now discussed more thoroughly in the revised manuscript. (See ~line 235, pg. 7)

One can look at Supplementary Movie 1 for examples of the relatively similar behavior between the dorsal mount (not rotated) and side mount (rotated) preparations. We do not have behavioral data from mice that were placed in both configurations. Our preliminary comparisons across mice indicates that side and dorsal mount mice show similar behavioral variability. We have added brief additional mention of these considerations on ~lines 235-250, pg 7.

It was in general important to make sure that the distance between the wheel and all four limbs was similar for both preparations. In particular, careful attention must be paid to the positioning of the front limbs in the side mount mice so that they are not too high off the wheel. This can be accomplished by a slight forward angling of the left support arm for side mount mice.

Although it would in principle be nearly possible to image the side mount preparation in the same optical configuration that we do without rotating the mouse, by rotating the objective 20 degrees to the right of vertical, we found that the last 2-3 degrees of missing rotation (our preparation is rotated 22.5 degrees left, which is more than the full available 20 degrees rotation of the objective), along with several other factors, made this undesirable. First, it was very difficult to image auditory areas without the additional flexibility to rotate the objective more laterally. Second, it was difficult or impossible to attach the horizontal light shield and to establish a water meniscus with the objective fully rotated. One could use gel instead (which we found to be optically inferior to water), but without the horizontal light shield, the UV and IR LEDs can reach the PMTs via the objective and contaminate the image or cause tripping of the PMT. Third, imaging the right pupil and face of the mouse is difficult to impossible under these conditions because the camera would need the same optical access angle as the objective, or would need to be moved down toward the air table and rotated up 20 degrees, in which case its view would be blocked by the running wheel and other objects mounted on the air table.

In parts, the description of the methods is very specific to the Thorlabs mesoscope which makes it harder to understand the general design choices and challenges for readers that are unfamiliar with that system. Since the Mesoscope is very expensive and therefore unavailable to many labs in the field, I think it would increase the reach of the manuscript to adjust the writing to be less specific for that system but instead provide general guidance that could also be helpful for other systems. For example (but not exclusively) lines 231-234 or lines 371 and below are very Thorlabs-specific.

Authors’ Response: We have revised the manuscript so that it is more generally applicable to mesoscopic methods.

We will make revisions as you suggest where possible, although we have limited experience with the other imaging systems that we believe you are referring to. However, please note that we already mentioned at least one other comparable system in the original eLife reviewed pre-print (Diesel 2p, line 209; Yu and Smith, 2021).

Here are a couple of examples of how we have broadened our description:

(1) On lines ~231-234, pg 7, we write:

“However, if needed, the objective of the Thorlabs mesoscope may be rotated laterally up to +20 degrees for direct access to more ventral cortical areas, for example if one wants to use a smaller, flat cortical window that requires the objective to be positioned orthogonally to the target region.”

Here have modified this to indicate that one may in general rotate their objective lens if their system allows it. Some systems, such as the Thorlabs Bergamo microscope and the Sutter MOM system, allow more than 20 degrees of rotation.

(2) On line ~371, pg 11, we write:

“This technique required several modifications of the auxiliary light-paths of the Thorlabs mesoscope”

Here, we have changed the writing to be more general such as “may require…of one’s microscope.”

Thank you for these valuable suggestions.

Lines 287-299: Could the authors quantify the variation in imaging depth, for example by quantifying to which extent the imaging depth has to be adjusted to obtain the position of the cortical surface across cortical areas? Given that curvature is a significant challenge in this preparation this would be useful information and could either show that this issue is largely resolved or to what extent it might still be a concern for the interpretation of the obtained results. How large were the required nominal corrections across imaging sites?

Authors’ Response: This information was provided previously (lines 297-299):

“In cases where we imaged multiple small ROIs, nominal imaging depth was adjusted in an attempt to maintain a constant relative cortical layer depth (i.e. depth below the pial surface; ~200 micrometer offset due to brain curvature over 2.5 mm of mediolateral distance, symmetric across the center axis of the window).”

This statement is based on a qualitative assessment of cortical depth based on neuron size and shape, the density of neurons in a given volume of cortex, the size and shape of blood vessels, and known cortical layer depths across regions. A ground-truth measurement of this depth error is beyond the scope of the present study. However, we do specify the type of glass, thickness, and curvature that we use, and the field curvature characterization of the Thorlabs mesoscope is given in Fig. 6 of the Sofroniew et al, 2016 eLife paper.

In addition, we have provided some documentation of online fast-z correction parameters on our GitHub page at:

https://github.com/vickerse1/mesoscope_spontaneous/tree/main/online_fast_z_correction

,and some additional relevant documentation can be found in our publicly available data repository on FigShare+ at: https://doi.org/10.25452/figshare.plus.c.7052513

Given the size of the implant and the subsequent work attachments, I wonder to which extent the field of view of the animal is obstructed. Did the authors perform receptive field mapping or some other technique that can estimate the size of the animals' remaining field of view?

Authors’ Response: The left eye is pointed down ~22.5 degrees, but we position the mouse near the left edge of the wheel to minimize the degree to which this limits their field of view. One may view our Fig. 1 and Suppl Movies 1 and 6 to see that the eyes on the left and right sides are unobstructed by the headpost, light shields, and support arms. However, other components of the experimental setup, such as the speaker, cameras, etc. can restrict a few small portions of the visual field, depending on their exact positioning.

The facts that mice responded to left side visual stimuli in preliminary recordings during our multimodal 2-AFC task, and that the unobstructed left and right camera views, along with pupillometry recordings, showed that a significant portion of the mouse’s field of view, from either side, remains intact in our preparation.

We have clarified these points in the text at ~lines 344-346, pg. 11.

Line 361: What does movie S7 show in this context? The movie seems to emphasize that the observed calcium dynamics are not driven by movement dynamics but it is not clear to me how this relates to the stimulation of PV neurons. The neural dynamics in the example cell are also not very clear. It would be helpful if this paragraph would contain some introduction/motivation for the optogenetic stimulation as it comes a bit out of the blue.

Authors’ Response: This result was presented for two reasons.

First, we showed it as a control for movement artifacts, since inhibition of neural activity enhances the relative prominence of non-activity dependent fluorescence that is used to examine the amplitude of movement-related changes in non-activity dependent fluorescence (e.g. movement artifacts). We have included a reference to this point at ~lines 587-588, pg 18.

Second, we showed it as a demonstration of how one may combine optogenetics with imaging in mesoscopic 2-P imaging. References to this point were already present in the original version of the manuscript (the eLife “ reviewed preprint”).

Lines 362-370: This paragraph and some of the following text are quite technical and would benefit from a better description and motivation of the general workflow. I have trouble following what exactly is done here. Are the authors using an online method to identify the CCF location of the 2p imaging based on the vessel pattern? Why is it important to do this during the experiment? Wouldn't it be sufficient to identify the areas of interest based on the vessel pattern beforehand and then adjust the 2p acquisition accordingly? Why are they using a dial, shutter, and foot pedal and how does this relate to the working distance of the objective? Does the 'standardized cortical map' refer to the Allen common coordinate framework?

Authors’ Response: We have revised this section to make it more clear.

Currently, the general introduction to this section appears in lines 349-361. Starting in line 362, we currently present the technical considerations needed to implement the overall goals stated in that first paragraph of this section.

In general we use a post-hoc analysis step to confirm the location of neurons recorded with 2-photon imaging. We use “online” juxtaposition of the multimodal map image with overlaid CCF with the 2-photon image by opening these two images next to each other on the ScanImage computer and matching the vasculature patterns “by eye”. We have made this more clear in the text so that the interested reader can more readily implement our methods.

By use of the phrase “standardized cortical map” in this context, we meant to point out that we had not decided a priori to use the Allen CCF v3.0 when we started working on these issues.

Does Fig. 2c show an example of the online alignment between widefield and 2p data? I was confused here since the use of suite2p suggests that this was done post-recording. I generally didn't understand why the user needed to switch back and forth between the two modes. Doesn't the 2p image show the vessels already? Also, why was an additional motorized dichroic to switch between widefield and 2p view needed? Isn't this the standard in most microscopes (including the Thorlabs scopes)?

Authors’ Response: We have explained this methodology more clearly in the revised manuscript, both at ~lines 485-500, pg 15-16, and ~lines 534-540, pg 17.

The motorized dichroic we used replaced the motorized mirror that comes with the Thorlabs mesoscope. We switched to a dichroic to allow for near-simultaneous optogenetic stimulation with 470 nm blue light and 2-photon imaging, so that we would not have to move the mirror back and forth during live data acquisition (it takes a few seconds and makes an audible noise that we wanted to avoid).

Figure 2c shows an overview of our two step “offline” alignment process. The image at the right in the bottom row labeled “2” is a map of recorded neurons from suite2p, determined post-hoc or after imaging. In Fig. 2d we show what the CCF map looks like when it’s overlaid on the neurons from a single suite2p session, using our alignment techniques. Indeed, this image is created post-hoc and not during imaging. In practice, “online” during imaging, we would have the image at left in the bottom row of Fig. 2c (i.e. the multimodal map image overlaid onto an image of the vasculature also acquired on the widefield rig, with the 22.5 degree rotated CCF map aligned to it based on the location of sensory responses) rotated 90 degrees to the left and flipped over a horizontal mirror plane so that its alignment matches that of the “online” 2-photon acquisition image and is zoomed to the same scale factor. Then, we would navigate based on vasculature patterns “by-eye” to the desired CCF areas, and confirm our successful 2-photon targeting of predetermined regions with our post-hoc analysis.

Why is the widefield imaging done through the skull under anesthesia? Would it not be easier to image through the final window when mice have recovered? Is the mapping needed for accurate window placement?

Authors’ Response: The headpost and window surgeries are done 3-7 days apart to increase success rate and modularize the workflow. Multimodal mapping by widefield imaging is done through the skull between these two surgeries for two major reasons. First, to make efficient use of the time between surgeries. Second, to allow us to compare the multimodal maps to skull landmarks, such as bregma and lambda, for improved alignment to the CCF.

Anesthesia was applied to prevent state changes and movements of the mouse, which can produce large, undesired effects on neural responses in primary sensory cortices in the context of these mapping experiments. We sometimes re-imaged multimodal maps on the widefield microscope through the window, roughly every 30-60 days or whenever/if significant changes in vasculature pattern became apparent.

We have clarified these points in the main text at ~lines 510-522, pg 20-21, and we added a link to our new supplementary material documenting the changes observed in the window preparation over time:

https://github.com/vickerse1/mesoscope_spontaneous/blob/main/window_preparation_stability.pdf

Thank you for these questions.

Lines 445 and below: Reducing the noise from resonant scanners is also very relevant for many other 2p experiments so it would be helpful to provide more general guidance on how to resolve this problem. Is the provided solution only applicable to the Thorlabs mesoscope? How hard would it be to adjust the authors' noise shield to other microscopes? I generally did not find many additional details on the Github repo and think readers would benefit from a more general explanation here.

Authors’ Response: Our revised Github repository has been modified to include more details, including both diagrams and text descriptions of the sound baffle, respectively:

https://github.com/vickerse1/mesoscope_spontaneous/blob/main/resonant_scanner_baffle/closed_cell_honeycomb_baffle_for_noise_reduction_on_resonant_scanner_devices.pdf

https://github.com/vickerse1/mesoscope_spontaneous/blob/main/resonant_scanner_baffle/closed_cell_honeycomb_baffle_methodology_summary.pdf

However, we can not presently disclose our confidential provisional patent application. Complete design information will likely be available in early 2025 when our full utility patent application is filed.

With respect to your question, yes, this technique is adaptable to any resonant scanner, or, for that matter, any complicated 3D surface that emits sound. We first 3D scan the surface, and then we reverse engineer a solid that fully encapsulates the surface and can be easily assembled in parts with bolts and interior foam that allow for a tight fit, in order to nearly completely block all emitted sound.

It is this adaptability that has prompted us to apply for a full patent, as we believe this technique will be quite valuable as it may apply to a potentially large number of applications, starting with 2-photon resonant scanners but possibly moving on to other devices that emit unwanted sound.

Does line 458 suggest that the authors had to perform a 3D scan of the components to create the noise reduction shield? If so, how was this done? I don't understand the connection between 3D scanning and printing that is mentioned in lines 464-466.

Authors’ Response: We do not want to release full details of the methodology until the full utility patent application has been submitted. However, we have now included a simplified text description of the process on our GitHub page and included a corresponding link in the main text:

https://github.com/vickerse1/mesoscope_spontaneous/blob/main/resonant_scanner_baffle/closed_cell_honeycomb_baffle_methodology_summary.pdf

We also clarified in the main text, at the location that you indicate, why the 3D scanning is a critical part of our novel 3D-design, printing, and assembly protocol.

Lines 468 and below: Why is it important to align single-cell data to cortical areas 'directly on the 2-photon microscope'? Is this different from the alignment discussed in the paragraph above? Why not focus on data interpretation after data acquisition? I understand the need to align neural data to cortical areas in general, I'm just confused about the 'on the fly' aspect here and why it seems to be broken out into two separate paragraphs. It seems as if the text in line 485 and below could also be placed earlier in the text to improve clarity.

Authors’ Response: Here by “such mapping is not routinely possible directly on the 2-photon mesoscope” what we mean is that it is not possible to do multimodal mapping directly on the mesoscope - it needs to be done on the widefield imaging rig (a separate microscope). Then, the CCF is mapped onto the widefield multimodal map, which is overlaid on an image of the vasculature (and sometimes also the skull) that was also acquired on the widefield imaging rig, and the vasculature is used as a sort of Rosetta Stone to co-align the 2-photon image to the multimodal map and then, by a sort of commutative property of alignment, to the CCF, so that each individual neuron in the 2-photon image can be assigned a unique CCF area name and numerical identifier for subsequent analysis.

We have clarified this in the text, thank you.

The Python code for aligning the widefield and 2-photon vessel images would also be of great value for regular 2p users. It would strongly improve the impact of the paper if the repository were better documented and the code would be equally applicable for alignment of imaging data with smaller cranial windows.

Authors’ Response: All of the code for multimodal map, CCF, and 2-photon image alignment is, in fact, already present on the GitHub page. We have made some minor improvements to the documentation, and readers are more than welcome to contact us for additional help.

Specifically, the alignment you refer to starts in cell #32 of the meso_pre_proc_1.ipynb notebook. In general the notebooks are meant to be run sequentially, starting with cell #1 of meso_pre_proc_1, then going to the next cell etc…, then moving to meso_pre_proc_2, etc… The purpose of each cell is labeled at the top of the cell in a comment.

We now include a cleaned, abridged version of the meso_pre_proc_1.pynb notebook that contains only the steps needed for alignment, and included a direct link to this notebook in the main text:

https://github.com/vickerse1/mesoscope_spontaneous/blob/main/python_code/mesoscope_preprocess_MMM_creation.ipynb

Rotated CCF maps are in the CCF map rotation folder, in subfolders corresponding to the angle of rotation.

Multimodal map creation involves use of the SensoryMapping_Vickers_Jun2520.m script in the Matlab folder.

We updated the main text to clarify these points and included direct links to scripts relevant to each processing step.

Figure 4a: I found it hard to see much of the structure in the Rastermap projection with the viridis colormap - perhaps also because of a red-green color vision impairment. Correspondingly, I had trouble seeing some of the structure that is described in the text or clearer differences between the neuron sortings to PC1 and PC2. Is the point of these panels to show that both PCs identify movement-aligned dynamics or is the argument that they isolate different movement-related response patterns? Using a grayscale colormap as used by Stringer et al might help to see more of the many fine details in the data.

Authors’ Response: In Fig. 4a the viridis color range is from blue to green to yellow, as indicated in the horizontal scale bar at bottom right. There is no red color in these Rastermap projections, or in any others in this paper. Furthermore, the expanded Rastermap insets in Figs. S4 and S5 provide additional detailed information that may not be clear in Fig 4a and Fig 5a.

We prefer, therefore, not to change these colormaps, which we use throughout the paper.

We have provided grayscale png versions of all figures on our GitHub page:

https://github.com/vickerse1/mesoscope_spontaneous/tree/main/grayscale_figures

In Fig 4a the point of showing both the PC1 and PC2 panels is to demonstrate that they appear to correspond to different aspects of movement (PC1 more to transient walking, both ON and OFF, and PC2 to whisking and sustained ON walk/whisk), and to exhibit differential ability to identify neurons with positive and negative correlations to arousal (PC1 finds both, both PC2 seems to find only the ON neurons).

We now clarify this in the text at ~lines 696-710, pg 22.

I find panel 6a a bit too hard to read because the identification and interpretation of the different motifs in the different qualitative episodes is challenging. For example, the text mentions flickering into motif 13 during walk but the majority of that sequence appears to be shaped by what I believe to be motif 11. Motif 11 also occurs prominently in the oscillate state and the unnamed sequence on the left. Is this meaningful or is the emphasis here on times of change between behavioral motifs? The concept of motif flickering should be better explained here.

Authors’ Response: Here motif 13 corresponds to a syllable that might best be termed “symmetric and ready stance”. This tends to occur just before and after walking, but also during rhythmic wheel balancing movements that appear during the “oscillate” behavior.

The intent of Fig. 6a is to show that each qualitatively identified behavior (twitch, whisk, walk, and oscillate) corresponds to a period during which a subset of BSOiD motifs flicker back and forth, and that the identity of motifs in this subset differs across the identified qualitative behaviors. This is not to say that a particular motif occurs only during a single identified qualitative behavior. Admittedly, the identification of these qualitative behaviors is a bit arbitrary - future versions of BSOiD (e.g. ASOiD) in fact combine supervised (i.e. arbitrary, top down) and unsupervised (i.e. algorithmic, objective, bottom-up) methods of behavior segmentation in attempt to more reliably identify and label behaviors.

Flickering appears to be a property of motif transitions in raw BSOiD outputs that have not been temporally smoothed. If one watches the raw video, it seems that this may in fact be an accurate reflection of the manner in which behaviors unfold through time. Each behavior could be thought of, to use terminology from MOSEQ (B Datta), as a series of syllables strung together to make a phrase or sentence. Syllables can repeat over either fast or slow timescales, and may be shared across distinct words and sentences although the order and frequency of their recurrence will likely differ.

We have clarified these points in the main text at ~lines 917-923, pg 29, and we added motif 13 to the list of motifs for the qualitative behavior labeled “oscillate” in Fig. 6a.

Lines 997-998: I don't understand this argument. Why does the existence of different temporal dynamics make imaging multiple areas 'one of the keys to potentially understanding the nature of their neuronal activity'?

Authors’ Response: We believe this may be an important point, that comparisons of neurobehavioral alignment across cortical areas cannot be performed by pooling sessions that contain different distributions of dwell times for different behaviors, if in fact that dependence of neural activity on behavior depends on the exact elapsed time since the beginning of the current behavioral “bout”. Again, other reasons that imaging many areas simultaneously would provide a unique advantage over imaging smaller areas one at a time and attempting to pool data across sessions would include the identification of sequences or neural ensembles that span many areas across large distances, or the understanding of distributed coding of behavior (an issue we explore in an upcoming paper).

We have clarified these points at the location in the Discussion that you have identified. Thank you for your questions and suggestions.

MinorLine 41: What is the difference between decision, choice, and response periods?

Authors’ Response: This now reads “...temporal separation of periods during which cortical activity is dominated by activity related to stimulus representation, choice/decision, maintenance of choice, and response or implementation of that choice.”

Line 202: What does ambulatory mean in this context?

Authors’ Response: Here we mean that the mice are able to walk freely on the wheel. In fact they do not actually move through space, so we have changed this to read “able to walk freely on a wheel, as shown in Figs. 1a and 1b”.

Is there a reason why 4 mounting posts were used for the dorsal mount but only 1 post was sufficient for the lateral mount?

Authors’ Response: Here, we assume you mean 2 posts for the side mount and 4 posts for the dorsal mount.

In general our idea was to use as many posts as possible to provide maximum stability of the preparations and minimize movement artifacts during 2-photon imaging. However, the design of the side mount headpost precluded the straight-forward or easy addition of a right oriented, second arm to its lateral/ventral rim - this would have blocked access of both the 2-photon objective and the right face camera. In the dorsal mount, the symmetrical headpost arms are positioned further back (i.e. posterior), so that the left and right face cameras are not obscured.

When we created the side mount preparation, we discovered that the 2 vertical 1” support posts were sufficient to provide adequate stability of the preparation and minimize 2-photon imaging movement artifacts. The side mount used two attachment screws on the left side of the headpost, instead of the one screw per side used in the dorsal mount preparation.

We have included these points/clarifications in the main text at ~lines 217-230, pg 7.

Figure S1g appears to be mislabeled.

Authors’ Response: Yes, on the figure itself that panel was mislabeled as “f” in the original eLife reviewed preprint. We have changed this to read “g”.

Line 349 and below: Why is the method called pseudo-widefield imaging?

Authors’ Response: On the mesoscope, broad spectrum fluorescent light is passed through a series of excitation and emission filters that, based on a series of tests that we performed, allow both reflected blue light and epifluorescence emitted (i.e. Stokes-shifted) green light to reach the CCD camera for detection. Furthermore, the CCD camera (Thorlabs) has a much smaller detector chip than that of the other widefield cameras that we use (RedShirt Imaging and PCO), and we use it to image at an acquisition speed of around 10 Hz maximum, instead of ~30-50 Hz, which is our normal widefield imaging acquisition speed (it also has a slower readout than what we would consider to be a standard or “real” 1-photon widefield imaging camera).

For these 3 reasons we refer to this as “pseudo-widefield” imaging. We would not use this for sensory activity mapping on the mesoscope - we primarily use it for mapping cortical vasculature and navigating based on our multimodal map to CCF alignment, although it is actually “contaminated” with some GCaMP6s activity during these uses.

We have briefly clarified this in the text.

Figures 4d & e: Do the colors show mean correlations per area? Please add labels and units to the colorbars as done in panel 4a.

Authors’ Response: For both Figs 4 and 5, we have added the requested labels and units to each scale bar, and have relabeled panels d to say “Rastermap CCF area cell densities”, and panels e to say “mean CCF area corrs w/ neural activity.”

Thank you for catching these omissions/mislabelings.

Line 715: what is superneuron averaging?

Authors’ Response: This refers to the fact that when Rastermap displays more than ~1000 neurons it averages the activity of each group of adjacent 50 neurons in the sorting to create a single display row, to avoid exceeding the pixel limitations of the display. Each single row representing the average activity of 50 neurons is called a “superneuron” (Stringer et al, 2023; bioRxiv).

We have modified the text to clarify this point.

Line 740: it would be good to mention what exactly the CCF density distribution quantifies.

Authors’ Response: In each CCF area, a certain percentage of neurons belongs to each Rastermap group. The CCF density distribution is the set of these percentages, or densities, across all CCF areas in the dorsal or side mount preparation being imaged in a particular session. We have clarified this in the text.

Line 745: what does 'within each CCF' mean? Does this refer to different areas?

Authors’ Response: The corrected version of this sentence now reads: “Next, we compared, across all CCF areas, the proportion of neurons within each CCF area that exhibited large positive correlations with walking speed and whisker motion energy.”

How were different Rastermap groups identified? Were they selected by hand?

Authors’ Response: Yes, in Figs. 4, 5, and 6, we selected the identified Rastermap groups “by hand”, based on qualitative similarity of their activity patterns. At the time, there was no available algorithmic or principled means by which to split the Rastermap sort. The current, newer version of Rastermap (Stringer et al, 2023) seems to allow for algorithmic discretization of embedding groups (we have not tested this yet), but it was not available at the time that we performed these preliminary analyses.

In terms of “correctness” of such discretization or group identification, we intend to address this issue in a more principled manner in upcoming publications. For the purposes of this first paper, we decided that manual identification of groups was sufficient to display the capabilities and outcomes of our methods.

We clarify this point briefly at several locations in the revised manuscript, throughout the latter part of the Results section.

**Reviewer #3 (Recommendations For The Authors):**
In "supplementary figures, protocols, methods, and materials", Figure S1 g is mislabeled as Figure f.

Authors’ Response: Yes, on the figure itself this panel was mislabeled as “f” in the original reviewed preprint. We have changed this to read “g”.

In S1 g, the success rate of the surgical procedure seems quite low. Less than 50% of the mice could be imaged under two-photon. Can the authors elaborate on the criteria and difficulties related to their preparations?

Authors’ Response: We will elaborate on the difficulties that sometimes hinder success in our preparations in the revised manuscript.

The success rate indicated to the point of “Spontaneous 2-P imaging (window) reads 13/20, which is 65%, not 50%. The drop to 9/20 by the time one gets to the left edge of “Behavioral Training” indicates that some mice do not master the task.

Protocol I contains details of the different ways in which mice either die or become unsuitable or “unsuccessful” at each step. These surgeries are rather challenging - they require proper instruction and experience. With the current protocol, our survival rate for the window surgery alone is as high as 75-100%. Some mice can be lost at headpost implantation, in particular if they are low weight or if too much muscle is removed over the auditory areas. Finally, some mice survive windowing but the imageable area of the window might be too small to perform the desired experiment.

We have added a paragraph detailing this issue in the main text at ~lines 287-320, pg 9.

In both Suppl_Movie_S1_dorsal_mount and Suppl_Movie_S1_side_mount provided (Movie S1), the behaviour video quality seems to be unoptimized which will impact the precision of Deeplabcut. As evident, there were multiple instances of mislabeled key points (paws are switched, large jumps of key points, etc) in the videos.Many tracked points are in areas of the image that are over-exposed.Despite using a high-speed camera, motion blur is obvious.Occlusions of one paw by the other paws moving out of frame.As Deeplabcut accuracy is key to higher-level motifs generated by BSOi-D, can the authors provide an example of tracking by exclusion/ smoothing of mislabeled points (possibly by the median filtering provided by Deeplabcut), this may help readers address such errors.

Authors’ Response: We agree that we would want to carefully rerun and carefully curate the outputs of DeepLabCut before making any strong claims about behavioral identification. As the aim of this paper was to establish our methods, we did not feel that this degree of rigor was required at this point.

It is inevitable that there will be some motion blur and small areas of over-exposure, respectively, when imaging whiskers, which can contain movement components up to ~150 Hz, and when imaging a large area of the mouse, which has planes facing various aspects. For example, perfect orthogonal illumination of both the center of the eye and the surface of the whisker pad on the snout would require two separate infrared light sources. In this case, use of a single LED results in overexposure of areas orthogonal to the direction of the light and underexposure of other aspects, while use of multiple LEDs would partially fix this problem, but still lead to variability in summated light intensity at different locations on the face. We have done our best to deal with these limitations.

We now briefly point out these limitations in the methods text at ~lines 155-160, pg 5.

In addition, we have provided additional raw and processed movies and data related to DeepLabCut and BSOiD behavioral analysis in our FigShare+ repository, which is located at:

https://doi.org/10.25452/figshare.plus.c.7052513

In lines 153-154, the authors mentioned that the Deeplabcut model was trained for 650k iterations. In our experience (100-400k), this seems excessive and may result in the model overfitting, yielding incorrect results in unseen data. Echoing point 4, can the authors show the accuracy of their Deeplabut model (training set, validation set, errors, etc).

Authors’ Response: Our behavioral analysis is preliminary and is included here as an example of our methods, and not to make claims about any specific result. Therefore we believe that the level of detail that you request in our DeepLabCut analysis is beyond the scope of the current paper. However, we would like to point out that we performed many iterations of DeepLabCut runs, across many mice in both preparations, before converging on these preliminary results. We believe that these results are stable and robust.

We believe that 650k iterations is within the reasonable range suggested by DLC, and that 1 million iterations is given as a reasonable upper bound. This seems to be supported by the literature for example, see Willmore et al, 2022 (“Behavioral and dopaminergic signatures of resilience”, Nature, 124:611, 124-132). Here, in a paper focused squarely on behavioral analysis, DLC training was run with 1.3 million iterations with default parameters.

We now note, on ~lines 153-154, pg 5, that we used 650K iterations, a number significantly less than the default of 1.03 million, to avoid overfitting.

In lines 140-141, the authors mentioned the use of slicing to downsample their data. Have any precautions, such as a low pass filter, been taken to avoid aliasing?

Authors’ Response: Most of the 2-photon data we present was acquired at ~3 Hz and upsampled to 10 Hz. Most of the behavioral data was downsampled from 5000 Hz to 10 Hz by slicing, as stated. We did not apply any low-pass filter to the behavioral data before sampling. The behavioral variables have heterogeneous real sampling/measurement rates - for example, pupil diameter and whisker motion energy are sampled at 30 Hz, and walk speed is sampled at 100 Hz. In addition, the 2-photon acquisition rate varied across sessions.

These facts made principled, standardized low-pass filtering difficult to implement. We chose rather to use a common resampling rate of 10 Hz in an unbiased manner. This downsampled 10 Hz rate is also used by B-SOiD to find transitions between behavioral motifs (Hsu and Yttri, 2021).

We do not think that aliasing is a major factor because the real rate of change of our Ca2+ indicator fluorescence and behavioral variables was, with the possible exception of whisker motion energy, likely at or below 10 Hz.

We now include a brief statement to this effect in the methods text at ~lines 142-146, pg. 4.

Line 288-299, the authors have made considerable effort to compensate for the curvature of the brain which is particularly important when imaging the whole dorsal cortex. Can the authors provide performance metrics and related details on how well the combination of online curvature field correction (ScanImage) and fast-z "sawtooth"/"step" (Sofroniew, 2016)?

Authors’ Response: We did not perform additional “ground-truth” experiments that would allow us to make definitive statements concerning field curvature, as was done in the initial eLife Thorlabs mesoscope paper (Sofroniew et al, 2016).

We estimate that we experience ~200 micrometers of depth offset across 2.5 mm - for example, if the objective is orthogonal to our 10 mm radius bend window and centered at the apex of its convexity, a small ROI located at the lateral edge of the side mount preparation would need to be positioned around 200 micrometers below that of an equivalent ROI placed near the apex in order to image neurons at the same cortical layer/depth, and would be at close to the same depth as an ROI placed at or near the midline, at the medial edge of the window. We determined this by examining the geometry of our cranial windows, and by comparing z-depth information from adjacent sessions in the same mouse, the first of which used a large FOV and the second of which used multiple small FOVs optimized so that they sampled from the same cortical layers across areas.

We have included this brief explanation in the main text at ~lines 300-311, pg 9.

In lines 513-515, the authors mentioned that the vasculature pattern can change over the course of the experiment which then requires to re-perform the realignment procedure. How stable is the vasculature pattern? Would laser speckle contrast yield more reliable results?

Authors’ Response: In general the changes in vasculature we observed were minimal but involved the following: (i) sometimes a vessel was displaced or moved during the window surgery, (ii) sometimes a vessel, in particular the sagittal sinus, enlarged or increased its apparent diameter over time if it is not properly pressured by the cranial window, and (iii) sometimes an area experiencing window pressure that is too low could, over time, show outgrowth of fine vascular endings. The most common of these was (i), and (iii) was perhaps the least common. In general the vasculature was quite stable.

We have added this brief discussion of potential vasculature changes after cranial window surgery to the main text at ~lines 286-293, pg 9.

We already mentioned, in the main text of the original eLife reviewed preprint, that we re-imaged the multimodal map (MMM) every 30-60 days or whenever changes in vasculature are observed, in order to maintain a high accuracy of CCF alignment over time. See ~lines 507-511, pg 16.

We are not very familiar with laser speckle contrast, and it seems like a technique that could conceivably improve the fine-grained accuracy of our MMM-CCF alignment in some instances. We will try this in the future, but for now it seems like our alignments are largely constrained by several large blood vessels present in any given FOV, and so it is unclear how we would incorporate such fine-grained modifications without applying local non-rigid manipulations of our images.

In lines 588-598, the authors mentioned that the occasional use of online fast-z corrections yielded no difference. However, it seems that the combination of the online fast-z correction yielded "cleaner" raster maps (Figure S3)?

Authors’ Response: The Rastermaps in Fig S3a and b are qualitatively similar. We do not believe that any systematic difference exists between their clustering or alignments, and we did not observe any such differences in other sessions that either used or didn’t use online fast-z motion correction.

We now provide raw data and analysis files corresponding to the sessions shown in Fig S3 (and other data-containing figures) on FigShare+ at:

https://doi.org/10.25452/figshare.plus.c.7052513

Ideally, the datasets contained in the paper should be available on an open repository for others to examine. I could not find a clear statement about data availability. Please include a linked repo or state why this is not possible.

Authors’ Response: We have made ~500 GB of raw data and preliminary analysis files publicly available on FigShare+ for the example sessions shown in Figures 2, 3, 4, 5, 6, S3, and S6. We ask to be cited and given due credit for any fair use of this data.

The data is located here:

Vickers, Evan; A. McCormick, David (2024). Pan-cortical 2-photon mesoscopic imaging and neurobehavioral alignment in awake, behaving mice. Figshare+. Collection:

https://doi.org/10.25452/figshare.plus.c.7052513

We intend to release a complete data set to the public as a Dandiset on the DANDI archive in conjunction with second and third in-depth analysis papers that are currently in preparation.